# DecompNet: Enhancing Time Series Forecasting Models with Implicit Decomposition

**Donghao Luo, Xue Wang**

Department of Precision Instrument, Tsinghua University, Beijing 100084, China

`ldh21@mails.tsinghua.edu.cn, wangxue@mail.tsinghua.edu.cn`

## Abstract

In this paper, we pioneer the idea of implicit decomposition. And based on this idea, we propose a powerful decomposition-based enhancement framework, namely **DecompNet**. Our method converts the time series decomposition into an implicit process, where it can give a time series model the decomposition-related knowledge during inference, even though this model does not actually decompose the input time series. Thus, our DecompNet can enable a model to inherit the performance promotion brought by time series decomposition but will not introduce any additional inference costs, successfully enhancing the model performance while enjoying better efficiency. Experimentally, our DecompNet exhibits promising enhancement capability and compelling framework generality. Especially, it can also enhance the performance of the latest and state-of-the-art models, greatly pushing the performance limit of time series forecasting. Through comprehensive comparisons, DecompNet also shows excellent performance and efficiency superiority, making the decomposition-based enhancement framework surpass the well-recognized normalization-based frameworks *for the first time*. Code is available at this repository: `https://github.com/luodhhh/DecompNet`.

## 1 Introduction

Time series forecasting is widely used in various real-world scenarios (e.g., transportation management [4], industrial monitoring [35, 25, 26], energy planning [36, 29] and weather forecasting [40]), helping the society make better decisions. Because of the immense practical value, various time series models are developed in recent years, bringing great prosperity to time series forecasting [39, 28, 1, 24, 27].

Apart from the proposal of time series models, designing model-agnostic enhancement frameworks is another hot research topic. Enhancement frameworks can assist the existing time series models to handle the non-stationary issue in time series data, effectively improving their performance. And recently, the research of enhancement frameworks is still dominated by normalization-based methods [11, 20, 9, 43].

On the other hand, time series decomposition is also a powerful tool to handle the non-stationary issue [10, 34], because it can unravel the entangled temporal patterns in raw time series and highlight the stationary components, making it easier to capture the temporal dependency. **However, there is still no popularized enhancement framework based on time series decomposition.** This can be attributed to the following reasons: (i) *Less framework generality.* Previous decomposition-based methods are only designed as specific time series models, not meeting the needs of model-agnostic frameworks. (ii) *Severe efficiency issues.* As shown in Figure 1 (a.1), previous decomposition-based methods need to explicitly decompose the time series into seasonal and trend components and maintain two expert models (namely seasonal model and trend model) for inference. This process will multiple the computation and inevitably introduce additional inference costs. Considering the

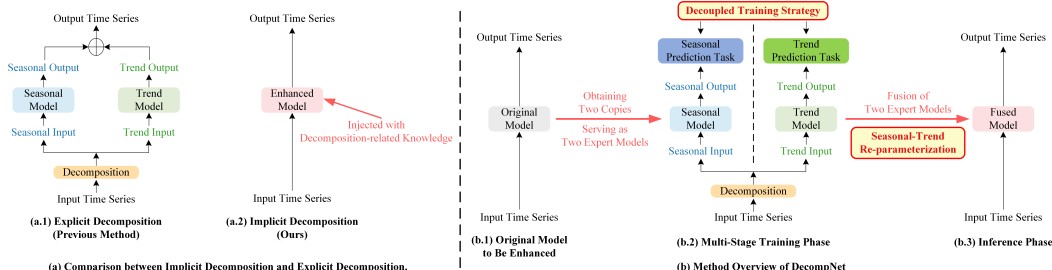

Figure 1: (a) Comparison between implicit decomposition and explicit decomposition. (b) Method overview of our DecompNet.

budget on compute and latency, these previous explicit decomposition methods are too costly to be an ideal enhancement framework.

Targeted to these issues, we propose the idea of **implicit decomposition**, in which we aim to give a time series model the decomposition-related knowledge during inference, even though it does not actually decompose the input time series. As shown in Figure 1 (a.2), our idea brings following advantages, meeting the needs of a powerful enhancement framework: (i) *Efficiency.* We only deploy one enhanced model for inference and no more need to explicitly decompose the input time series, *successfully solving the efficiency issue in previous explicit decomposition methods.* (ii) *Performance.* Although directly handling the undecomposed input time series, the enhanced model can still utilize the already-injected decomposition-related knowledge to *enhance the performance.* (iii) *Framework generality.* We can inject the decomposition-related knowledge into any time series models, regardless of the model-specific macro designs, nicely meeting the needs of model-agnostic frameworks.

Based on this idea, we propose a powerful decomposition-based enhancement framework named **DecompNet**. Technically, as shown in Figure 1 (b), when given a time series model to be enhanced, we first pretrain two copies of it on the decomposed seasonal and trend data, helping them grasp the decomposition-related knowledge and making them to be the seasonal and trend expert models. Then we fuse these two expert models back as one single model, merging all of their decomposition-related knowledge into the fused model. In this way, we make it to inject the decomposition-related knowledge into a single model (i.e., the fused model). And this model can be seen as an enhanced version of the original given model. During inference, since the decomposition-related knowledge has been injected into the fused model, it can utilize the decomposition-related knowledge to enhance the performance without actually decomposing the input time series, enjoying great performance improvement while not introducing any additional inference costs. Experimentally, when compared with other enhancement frameworks and explicit decomposition methods, our method shows **comprehensive superiority** in performance, efficiency, enhancement capability and framework generality. Especially, our method can also enhance the performance of the latest and state-of-the-art models, greatly pushing the performance limit of time series forecasting. **Our contributions are:**

- We pioneer the idea of implicit decomposition and provide guidance to upgrade the time series decomposition into a model-agnostic enhancement framework. Based on this idea, we propose a powerful decomposition-based enhancement framework called DecompNet, which features two key designs: Seasonal-Trend Re-parameterization and decoupled training strategy.

- DecompNet shows promising enhancement capability, framework generality and performance superiority, making the decomposition-based enhancement frameworks surpass the well-recognized normalization-based frameworks *for the first time*.

- Our method can enhance the model performance without introducing any additional inference costs, enjoying totally the same inference costs as the original model. This *inference-cost-free* property demonstrates the efficiency superiority of our method.

- As a brand new idea, our implicit decomposition can bring some fresh perspectives and provide a better solution to the classic research topic of time series decomposition.

## 2 Related Work

### 2.1 Time Series Decomposition

Time series decomposition is an important technique in time series community. In the latest decomposition-based methods, the mainstream procedure is to decompose the time series into seasonal and trend components and process them with two expert models respectively. In terms of specific implementations, most methods adopt the classic moving average to decompose the time series [45, 46]. As an improvement, [37] adopts a multi-kernel decomposition and [44] proposes a learnable decomposition. Besides, [38] proposes to decompose the time series in frequency domain. However, these explicit decomposition methods are mainly designed as specific time series models and cannot serve as model-agnostic enhancement frameworks. To fill this gap, we pioneer the idea of implicit decomposition, providing guidance to upgrade the time series decomposition into a model-agnostic enhancement framework. Please refer to Appendix F.1, F.2, F.3 for more discussion.

### 2.2 Time Series Enhancement Framework

Time series enhancement framework can assist time series models to solve the non-stationary issue in time series data and improve their performance. The mainstream enhancement frameworks can be categorized as normalization-based and prior-based ones. In detail, normalization-based frameworks can alleviate the non-stationary issue from the perspective of statistical measures [11, 9, 20, 43]. And prior-based frameworks can also improve the model performance, provided that the periodicity prior of a dataset is known in advance [15, 16]. In this paper, we establish a new category of decomposition-based enhancement framework and propose DecompNet as its representative, which can greatly improve the model performance and bring completely no additional inference costs.

It's worth noting that, previous studies generally consider normalization-based enhancement frameworks as better solutions for the non-stationary issue [9, 20, 43]. And *we are the first* to propose a decomposition-based enhancement framework that achieves better performance than normalization-based ones, which is a great breakthrough in this direction. More discussion is in Appendix F.4.

### 2.3 Structural Re-parameterization

Structural re-parameterization is a methodology to convert model structures via fusing model parameters. For example, [6, 8, 7, 17, 23] adopt structural re-parameterization to simplify the structure of multi-branch convolution networks, which is achieved by fusing the multi-branch part as one single branch, successfully solving their efficiency issues while maintaining their performance. M2PT [47] further extends its application to fusing two individual neural networks, which is achieved by fusing each of the layers at corresponding positions of two models. As a variant of structural re-parameterization specially designed for implicit decomposition, we propose Seasonal-Trend Re-parameterization in this paper, which brings efficiency superiority to our method.

## 3 DecompNet

### 3.1 Preliminaries

**Time Series Forecasting** Given a length-$I$ multivariate time series with $M$ variates as input, time series forecasting aims to predict the length-$T$ future series ($\widehat{\mathbf{Y}} \in \mathbb{R}^{T \times M}$) based on this length-$I$ input series ($\mathbf{X} \in \mathbb{R}^{I \times M}$). To bring better forecasting performance, we propose a model-agnostic enhancement framework, namely DecompNet, to assist existing time series forecasters, which can effectively improve their performance and greatly push the performance limit of time series forecasting.

**Time Series Decomposition** Time series decomposition is mainly carried out based on moving average. In detail, we first smooth out the raw time series and obtain the trend component. Then we obtain the seasonal component by subtracting the estimated trend component from the raw time series. For a length-$I$ input time series $\mathbf{X} \in \mathbb{R}^{I \times M}$, the decomposition process can be formulated as:

$$\mathbf{X}_t = \mathrm{AvgPool}(\mathrm{Padding}(\mathbf{X})) \tag{1}$$
$$\mathbf{X}_s = \mathbf{X} - \mathbf{X}_t \tag{2}$$

where $\mathbf{X}_s, \mathbf{X}_t \in \mathbb{R}^{I \times M}$ denote the seasonal and the trend components, respectively. We adopt the $\mathrm{AvgPool}(\cdot)$ for moving average and set the moving average window size as 25 by default. And we adopt the $\mathrm{Padding}(\cdot)$ operation to keep the series length unchanged, which is employed using terminal values. For simplicity, we use $\mathbf{X}_s, \mathbf{X}_t = \mathrm{SeriesDecomp}(\mathbf{X})$ to summarize above equations.

## 3.2 Framework Overview

**Figure 1 (b) shows the overall workflow of DecompNet**. During training, the main purpose of our method is to give a model the decomposition-related knowledge and enhance its performance. And we propose a "first-decouple-then-fuse" multi-stage training process to achieve this goal.

When given a time series model to be enhanced, we first pretrain two copies of it on the decomposed seasonal and trend data, helping them grasp the decomposition-related knowledge and making them to be the seasonal and trend expert models. During this pretraining stage for two expert models, we propose a **decoupled training strategy** to help them better grasp the decomposition-related knowledge, which will be introduced in Section 3.3.

Then we fuse these two pretrained expert models back as one single model, merging all of their decomposition-related knowledge into the fused model. In this way, we make it to inject the decomposition-related knowledge into a single model (i.e., the fused model). And this fused model can be seen as an enhanced version of the original given model. And the key design of this fusion stage is **Seasonal-Trend Re-parameterization**, a variant of structural re-parameterization specially designed for our implicit decomposition, which will be introduced in Section 3.4.

During inference, we only deploy this single fused model for inference and no more need to explicitly decompose the input time series. Since the decomposition-related knowledge has been injected into the fused model, it can utilize the decomposition-related knowledge to enhance the performance without actually decomposing the input time series. Thus, our method can enhance the model performance without introducing any additional inference costs, enjoying totally the same inference costs as the original model.

## 3.3 Decoupled Training Strategy

We propose a decoupled training strategy to better pretrain the seasonal expert model and trend expert model. As shown in Figure 2 (a), our strategy decouples the whole-series prediction task into a seasonal prediction task and a trend prediction task respectively, which can provide more direct and clearer supervision signals for two expert models.

Given $\mathbf{X} \in \mathbb{R}^{I \times M}$ as the input time series, we first decompose it into the seasonal and trend inputs $\mathbf{X}_s, \mathbf{X}_t \in \mathbb{R}^{I \times M}$. Then they are passed into the two expert models to predict the seasonal and trend outputs respectively:

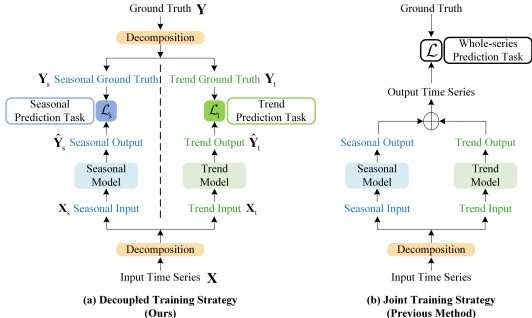

Figure 2: Illustration of our decoupled training strategy. We also provide a figure of previous joint training strategy for comparison.

$$\mathbf{X}_s, \mathbf{X}_t = \text{SeriesDecomp}(\mathbf{X}) \tag{3}$$

$$\widehat{\mathbf{Y}}_s = \text{Model}_s(\mathbf{X}_s) \tag{4}$$

$$\widehat{\mathbf{Y}}_t = \text{Model}_t(\mathbf{X}_t) \tag{5}$$

where $\text{Model}_s$ and $\text{Model}_t$ are the seasonal model and trend model respectively. And $\widehat{\mathbf{Y}}_s, \widehat{\mathbf{Y}}_t \in \mathbb{R}^{T \times M}$ denote the predicted seasonal output and trend output.

Then in terms of loss calculation, we also decompose the ground truth $\mathbf{Y}$ to obtain the seasonal ground truth and trend ground truth, denoted as $\mathbf{Y}_s, \mathbf{Y}_t \in \mathbb{R}^{T \times M}$. Then we can calculate losses for the seasonal part and trend part separately, which are $\mathcal{L}_s$ and $\mathcal{L}_t$. In this way, our strategy can provide separate supervision signals for two expert models, making two models trained in a decoupled manner:

$$\mathbf{Y}_s, \mathbf{Y}_t = \text{SeriesDecomp}(\mathbf{Y}) \tag{6}$$

$$\mathcal{L}_s = \|\widehat{\mathbf{Y}}_s - \mathbf{Y}_s\|_2^2 \tag{7}$$

$$\mathcal{L}_t = \|\widehat{\mathbf{Y}}_t - \mathbf{Y}_t\|_2^2 \tag{8}$$

**Advantages of Decoupled Training Strategy**   The goal of decomposition-based methods is to train two expert models specialized in the trend prediction task and seasonal prediction task, respectively. By decoupling the training objective from the previous whole-series prediction task into a

seasonal prediction task and a trend prediction task, our strategy can provide more direct and clearer supervision signals for two expert models, thus bringing better training results and fully unleashing the performance potential of decomposition-based methods. And our strategy can also contribute to faster convergence, since it unravels $\mathbf{Y}$ into more predictable components and makes the training process easier. Please refer to Appendix D for details.

## 3.4 Seasonal-Trend Re-parameterization

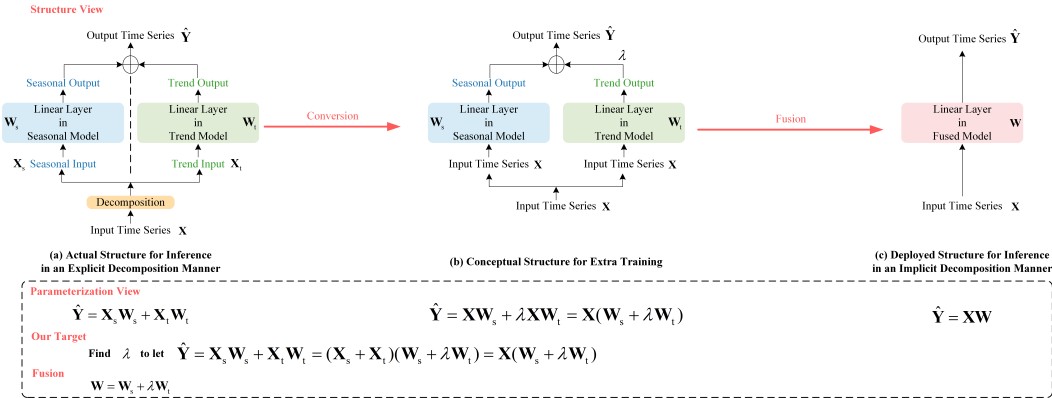

Figure 3: Illustration of Seasonal-Trend Re-parameterization. (a) The actual structure of two pretrained expert models, which infers in an explicit decomposition manner and suffers from heavier inference costs. (b) The conceptual structure for extra training of $\lambda$. (c) The final deployed structure. It infers in an implicit decomposition manner, thus solving the efficiency issue.

After above decoupled pretraining, if we directly deploy the two pretrained expert models for inference in an explicit decomposition manner (Figure 3 (a)), it will bring heavier inference costs. To solve this efficiency issue, we propose Seasonal-Trend Re-parameterization to fuse these two expert models and convert the decomposition process into an implicit manner.

**Fusion of linear layers** As shown in Figure 3, we start from the fusion of two one-layer expert models to introduce our Seasonal-Trend Re-parameterization. Let $\mathbf{X}_s, \mathbf{X}_t$ be the seasonal and trend inputs decomposed from input series $\mathbf{X}$. And let $\mathbf{W}_s$ and $\mathbf{W}_t$ be the weight matrices for the linear layers in seasonal model and trend model, respectively. If we directly deploy these two expert models for inference in an explicit decomposition manner (Figure 3 (a)), we have:

$$\widehat{\mathbf{Y}} = \mathbf{X}_s\mathbf{W}_s + \mathbf{X}_t\mathbf{W}_t \tag{9}$$

Here, we use the matrix multiplication between an input data and a weight matrix to indicate the forward process in a linear layer. And we omit the bias term for brevity.

We observe that $\mathbf{X} = \mathbf{X}_s + \mathbf{X}_t$ (Equation 2). So the data part is naturally mergeable. And if the weight matrix part can also be merged, we can convert above explicit decomposition into an implicit manner (Figure 3 (c)). Inspired by [47], we propose Seasonal-Trend Re-parameterization to achieve this goal, where we intent to find a **learnable fusion factor** $\lambda$ to fuse the two pretrained weight matrices $\mathbf{W}_s, \mathbf{W}_t$ as a single matrix $\mathbf{W}$, letting $\mathbf{W} = \mathbf{W}_s + \lambda\mathbf{W}_t$ and making following target hold:

$$\begin{aligned}
\widehat{\mathbf{Y}} &= \mathbf{X}_s\mathbf{W}_s + \mathbf{X}_t\mathbf{W}_t \\
&= (\mathbf{X}_s + \mathbf{X}_t)(\mathbf{W}_s + \lambda\mathbf{W}_t) \\
&= \mathbf{X}(\mathbf{W}_s + \lambda\mathbf{W}_t) \\
&= \mathbf{XW}
\end{aligned} \tag{10}$$

From above process, we make it to convert the explicit decomposition into an implicit process, in which we can directly process the undecomposed raw time series $\mathbf{X}$ and only need to deploy one linear layer with $\mathbf{W}$ as the weight matrix for inference (Figure 3 (c)). This linear layer is the fusion of the two linear layers from seasonal and trend models, so it can inherit their pretrained decomposition-related knowledge to enhance its performance.

To make our target described in Equation 10 hold, **we should introduce an extra training process to learn the optimal fusion factor** $\lambda$, as suggested by [47]. In detail, since we have $\widehat{\mathbf{Y}} = \mathbf{X}(\mathbf{W}_s + \lambda \mathbf{W}_t) = \mathbf{X}\mathbf{W}_s + \lambda \mathbf{X}\mathbf{W}_t$ according to Equation 10, we can convert the actual structure of two expert models (Figure 3 (a)) into a conceptual structure for this extra training process (Figure 3 (b)), where both expert models turn to directly receive the raw time series $\mathbf{X}$ as inputs, and the output is their weighted sum scaled by $\lambda$. **In this stage, we let $\mathbf{W}_s$ and $\mathbf{W}_t$ continue to be trained along with** $\lambda$. Since $\mathbf{W}_s$ and $\mathbf{W}_t$ are already well pretrained by decomposition-related knowledge, it helps to make this extra training process converge quickly.

After above extra training, we merge the parameters by computing $\mathbf{W} = \mathbf{W}_s + \lambda \mathbf{W}_t$ and only save this fused model parameterized by $\mathbf{W}$. Thus we only need to deploy one model for inference and no longer explicitly decompose the input series, making our method not introduce any additional inference costs, successfully solving the efficiency issue.

Note that the fusion logic of convolution layer is similar to linear layer [42], since the forward process in a convolution layer can be indicated by the convolution operation between an input data and a weight matrix, which is similar to the matrix multiplication operation in the forward process in a linear layer. So we can realize the fusion of convolution layers in a similar way we fuse linear layers.

**Fusion of complicated models** Based on some foundational researches on structural re-parameterization [47], we can achieve the fusion of two individual neural networks by fusing each of their layers at corresponding positions.

And specifically for time series community, most of the complicated models are built upon linear layers. For example, the embedding layer, the prediction head and the backbone in MLP-based models are all made up of linear layers. Similarly, the backbone in convolution-based models are made up of convolution layers. For more complicated Transformer-based models, their learnable parameters are also within linear layers (e.g., QKV and output projections in the attention block and those in the FFN block). Thus, by applying Seasonal-Trend Re-parameterization to fuse each corresponding linear or convolution layer, we can realize the fusion of two complicated expert models. And in this fusion process, $\lambda$ factors are learned independently for each layer.

In this paper, our fusion mechanism relies on strict layer-wise alignment and the prerequisite for our fusion process is that the two expert models must have the same model architecture. And this prerequisite definitely holds true in this paper, since the two expert models in our fusion process are two copies of the same original model, ensuring that they definitely have the same architecture. It will be our future work to further study on how to handle the architectural mismatches during fusion. Please see Appendix Q for more details.

## 4 Experiments

In this section, we conduct extensive experiments to evaluate the effectiveness of our DecompNet. In particular, we investigate the enhancement capability of DecompNet by integrating it with various advanced time series backbone models in Section 4.1, compare our DecompNet with other enhancement frameworks in Section 4.2 and compare our idea of implicit decomposition with the previous idea of explicit decomposition in Section 4.3. See Appendix A and C for details of experimental setups. Please also refer to Appendix S for visualization showcases and forecasting examples.

### 4.1 Main Results

**Setups** We conduct long-term forecasting experiments on 8 popular real-world benchmarks, including Weather [40], Traffic [33], ECL [36], Solar-Energy [29] and 4 ETT datasets [48]. Following the previous settings, we set prediction lengths as $\{96, 192, 336, 720\}$ and fix the input length as 384. We calculate the mean squared error (MSE) and mean absolute error (MAE) of multivariate time series forecasting as metrics. In all experiments, we set the moving average window size as 25 for time series decomposition during the decoupled pretraining stage. And we provide the parameter sensitivity study in Section 4.5. Our method is trained with the L2 loss, using the ADAM optimizer with an initial learning rate in $\{10^{-3}, 5 \times 10^{-4}, 10^{-4}\}$. The default training process is 30 epochs with proper early stopping (i.e., we set the maximum epoch as 30 and set the early stop patience as 3). More datasets details and implementation details are in Appendix B and C.

**Backbone Models** DecompNet is a model-agnostic enhancement framework and can be applied to any time series forecasters. To better prove our enhancement capability, we adopt following

representative forecasters as strong backbone models, including the Transformer-based models: PatchTST [30], iTransformer [18]; the MLP-based models: RLinear, RMLP [14] and the Convolution-based model: ModernTCN [23]. The selection criterion of backbone models are in Appendix C.

**Results** As shown in Table 1, DecompNet consistently enhances the performance of all backbone models by a large margin. For instance, on Solar, ETTh1, and Traffic datasets, the average MSE performance improvements are rather substantial: 8.0%, 6.4% and 6.1% respectively. Note that these performance promotions are brought by time series decomposition. It proves that our DecompNet can really inherit the performance promotion obtained from time series decomposition even though not actually decomposing the input time series, verifying the feasibility and soundness of our idea of implicit decomposition.

It is worth noting that the selected backbone models have already achieved the previous state-of-the-art performance by themselves. And our framework can still further improve their performance, which can truly reflect our strong enhancement capability. Meanwhile, the backbone models cover the mainstream types of time series forecasters and our DecompNet brings consistent performance enhancement to them, which can firmly prove our framework generality. Moreover, by improving the performance of these state-of-the-art models, this paper greatly pushes the performance limit of time series forecasting, setting a new standard for future researches.

Table 1: Performance promotion obtained by our DecompNet framework in long-term forecasting tasks. We adopt five mainstream state-of-the-art time series forecasters as backbone models. A lower MSE or MAE indicates a better performance and the best results are in **bold**. Results are averaged from four prediction lengths $T \in \{96, 192, 336, 720\}$. See Table 12 in Appendix R.1 for full results. And we also report the full error bar in Appendix I, which can verify that the improvement in our main result is meaningful and is not due to chance.

| Models | PatchTST | | iTransformer | | RLinear | | RMLP | | ModernTCN | |
|---|---|---|---|---|---|---|---|---|---|---|
| Settings | Original [30] | +DecompNet (Ours) | Original [18] | +DecompNet (Ours) | Original [14] | +DecompNet (Ours) | Original [14] | +DecompNet (Ours) | Original [23] | +DecompNet (Ours) |
| Metric | MSE MAE | MSE MAE | MSE MAE | MSE MAE | MSE MAE | MSE MAE | MSE MAE | MSE MAE | MSE MAE | MSE MAE |
| ETTh1 | 0.418 0.433 | **0.403 0.421** | 0.461 0.464 | **0.417 0.435** | 0.415 0.428 | **0.410 0.422** | 0.482 0.469 | **0.423 0.430** | 0.403 0.419 | **0.387 0.411** |
| ETTh2 | 0.349 0.391 | **0.346 0.387** | 0.377 0.412 | **0.344 0.389** | 0.354 0.397 | **0.338 0.383** | 0.364 0.401 | **0.347 0.390** | 0.328 0.382 | **0.312 0.373** |
| ETTm1 | 0.355 0.385 | **0.345 0.380** | 0.372 0.399 | **0.358 0.385** | 0.364 0.381 | **0.359 0.377** | 0.371 0.394 | **0.355 0.383** | 0.365 0.386 | **0.354 0.379** |
| ETTm2 | 0.262 0.319 | **0.257 0.314** | 0.274 0.332 | **0.262 0.321** | 0.259 0.316 | **0.255 0.312** | 0.271 0.326 | **0.259 0.318** | 0.261 0.320 | **0.254 0.314** |
| Weather | 0.227 0.264 | **0.224 0.261** | 0.235 0.272 | **0.229 0.268** | 0.246 0.281 | **0.235 0.275** | 0.244 0.279 | **0.239 0.275** | 0.233 0.270 | **0.231 0.266** |
| Solar-Energy | 0.218 0.317 | **0.199 0.257** | 0.212 0.272 | **0.204 0.259** | 0.256 0.323 | **0.227 0.274** | 0.231 0.294 | **0.214 0.249** | 0.218 0.306 | **0.200 0.258** |
| ECL | 0.163 **0.260** | **0.161 0.260** | 0.172 0.271 | **0.168 0.264** | 0.178 0.275 | **0.168 0.259** | 0.172 0.265 | **0.165 0.258** | 0.168 0.264 | **0.159 0.258** |
| Traffic | 0.410 0.288 | **0.387 0.268** | 0.428 0.320 | **0.395 0.280** | 0.453 0.314 | **0.425 0.286** | 0.436 0.308 | **0.413 0.285** | 0.417 0.293 | **0.394 0.275** |

## 4.2 Compared with Other Enhancement Frameworks

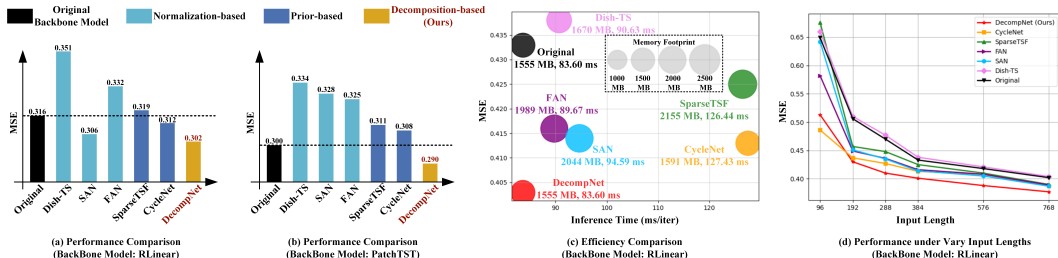

Figure 4: Comparison of our DecompNet with other enhancement frameworks. (a) and (b) Performance comparison. A lower MSE (a shorter bar) means better performance. Results are averaged from 8 datasets. See Appendix R.3 for full results. (c) Efficiency comparison. We conduct experiments in Traffic dataset under input-384-predict-96 setting. A smaller memory usage and a lower inference time indicate better efficiency. (d) Performance under different input lengths. We conduct experiments in Traffic dataset and fix the prediction length as 96.

**Setups** We compare our DecompNet with other time series enhancement frameworks, including the popular normalization-based frameworks: Dish-TS [9], SAN [20], FAN [43] and the latest prior-based frameworks: SparseTSF [16], CycleNet [15]. And we adopt RLinear [14] and PatchTST [30] as backbone models for this comparison.

**Performance Comparison**    Figure 4 (a) and (b) show the excellent performance superiority of DecompNet. Concretely, our DecompNet achieves the best performance among existing enhancement frameworks and brings the best enhancement effects to all backbone models. In detail, DecompNet is *the first* decomposition-based method that outperforms the well-recognized normalization-based methods, which is a great breakthrough in this direction. And DecompNet also surpasses the latest prior-based methods. For instance, when using PatchTST as the backbone model, DecompNet achieves the best performance with an average MSE of 0.290, obtaining an average of 6.3% MSE reduction than the latest prior-based methods in all datasets. Besides, as an advantage over these prior-based methods, our method can perform well without the need of any prior information about the dataset, thus being suitable for a wider range of application scenarios.

**Framework Generality Comparison**    Although normalization-based enhancement frameworks can work well on the simple backbone model like RLinear [14], they fail to enhance the performance of more advanced backbone model like PatchTST [30]. As shown in Figure 4 (b), combining normalization-based frameworks with PatchTST brings negative effects, making their performance even worse than the original model's. By contrast, our DecompNet can bring consistent performance promotion to both kinds of backbone models, showing better framework generality. And we'd like to highlight that our method is currently *the only one* that can improve the performance of more advanced backbone models, which is a unique advantage that sets us apart from other competitors.

According to previous studies [30, 23], the more advanced backbone models can learn to handle the statistical informaition by themselves. As a result, the normaliztion-based enhancement framework, which also works from the perspective of statistical informaition, is not necessary for these advanced backbone models. By contrast, our results indicate that existing backbone models still fail to obtain the decomposition-related knowledge on their owns. Therefore, the external assistance from our decomposition-based enhancement framework is still needed.

**Efficiency Comparison**    We comprehensively compare the forecasting performance and inference costs (i.e., inference speed and memory usage) of the selected enhancement frameworks. And the results are shown in Figure 4 (c). When bringing the best enhancement effect to the backbone model, our method has totally the same inference speed and memory usage as the original model, proving that our method can greatly improve the forecasting performance and bring completely no additional inference costs. Our efficiency superiority can be credited to the idea of implicit decomposition. Since we don't need to explicitly decompose the input series for inference, it can greatly improve our efficiency. As a comparison, other enhancement frameworks need to explicitly extract the periodic, statistical or frequency information of the input series for inference, which inevitably brings more memory usage and inference time, making them less efficient.

**Robustness to Vary Input Length**    As shown in Figure 4 (d), all enhancement frameworks gain continuous performance improvement with the increasing input length, validating their effectiveness in extracting useful information from longer history. Overall, our method achieves the best performance under most input lengths, demonstrating our great adaptability to vary input lengths and highlighting our performance superiority.

**Conclusion**    In conclusion, DecompNet achieves the best in both performance and efficiency. And our method also enjoys a wider range of applicability since it doesn't require any prior information of the dataset. These results validate that our DecompNet is an ideal enhancement framework to solve the non-stationary issue in time series data. Besides, the success of DecompNet makes the decomposition-based enhancement frameworks surpass the well-recognized normalization-based methods *for the first time*. This breakthrough reveals the great potential of decomposition-based methods, which can encourage further studies in this direction.

### 4.3    Compared with Explicit Decomposition

**Setups**    Our DecompNet is designed based on the idea of implicit decomposition, which is a brand new idea to apply decomposition in time series forecasting. And our implicit decomposition holds very different opinions from the previous idea of explicit decomposition. To provide an adequate comparison of these two design ideas, we compare our DecompNet with some latest and representative explicit decomposition methods, i.e., MICN [37] and Leddam [44]. Please refer to Appendix F.3 for a detailed introduction of their model structures and data processing pipelines.

**Overall Comparison**    As shown in Figure 5 (a), our implicit decomposition outperforms the previous explicit decomposition by a large margin in all datasets, indicating that our implicit decomposition is a better solution to apply decomposition in time series forecasting.

**Component-wise Analysis** To clearly demonstrate where our performance superiority comes from, we also gradually modify these explicit decomposition methods with our implicit decomposition designs. The results are in Figure 5 (b). And we have some interesting findings that challenge the commonly held belief in previous decomposition-based methods, which are as follows:

**(1) What contributes to better performance?** Previous explicit decomposition pays more attention on data processing, while our implicit decomposition focuses more on model training. To provide a comparison, we remove the sophisticated data processing in previous explicit decomposition methods and apply our decoupled training strategy to them. We observe great performance improvement after this modification (14.7% promotion in MICN and 6.3% in

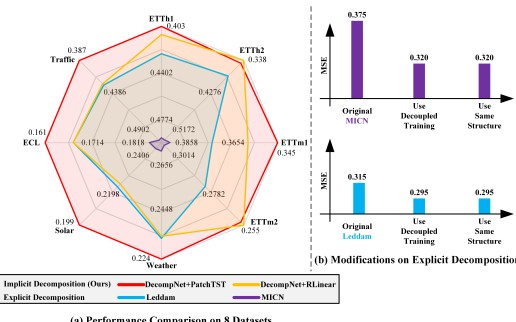

Figure 5: (a) Performance (average MSE results) comparison with explicit decomposition methods on 8 datasets. (b) Modifications on explicit decomposition methods with our implicit decomposition designs. Results are further averaged from 8 datasets. Full results are in Appendix R.2.

Leddam), which can prove the importance of model training, highlighting that a better training strategy is the key to fully unleash the performance potential of decomposition-based methods, not the more sophisticated data processing.

**(2) Do the two expert models have to use different model structures?** The two expert models in our implicit decomposition have the same model structure, since they are actually two copies of the same original model. But previous explicit decomposition advocates to use different model structures in two expert models and consider it as a prerequisite to handle the different nature in seasonal and trend components. We also re-examine this advocacy and surprisingly find that *using the same model structure does not result in performance degradation*. We attribute this finding to our decoupled training strategy, since it can better train the two expert models and fully realize the strong representation capabilities within neural networks, making them able to capture the meaningful seasonal and trend representations even with same model structure. This finding verifies the feasibility to use same model structure in two expert models, paving the way to improve the efficiency by our Seasonal-Trend Re-parameterization.

**Conclusion** In this section, we conduct comprehensive comparisons and prove that our implicit decomposition is a better idea to apply decomposition in time series forecasting. Meanwhile, we re-examine many well-established practices in existing decomposition-based methods and reveal that they are not optimal choices, making the research on time series decomposition still an open question. We hope these new findings can prompt people to rethink this classic research topic and design more innovative methods for time series decomposition.

## 4.4 Ablation Study

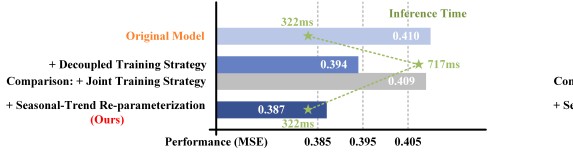 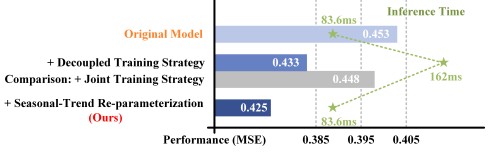

(a) Ablation results in Traffic dataset using PatchTST as backbone model  (b) Ablation results in Traffic dataset using RLinear as backbone model

Figure 6: Ablation study. A lower MSE and inference time (a shorter blue bar and a green star closer to the vertical axis) indicate better performance and efficiency. More results are in Appendix G.

To validate the effectiveness of our designs, we start from an original model and gradually enhance it by adding our designs step by step. And we provide the trajectory going from the original model to its enhanced version in Figure 6.

**Decoupled Training Strategy** We first obtain two copies of the original model and train them to be the expert models for seasonal and trend prediction tasks. And we directly deploy these two expert models for inference in an explicit decomposition manner without fusion. We observe great

performance improvement when using our decoupled training strategy, while using joint training strategy results in less performance promotion. This result proves that our decoupled training strategy is a better strategy to fully unleash the performance potential of decomposition-based methods.

**Seasonal-Trend Re-parameterization**   Secondly, the fusion by our Seasonal-Trend Re-parameterization can greatly improve the efficiency. And it causes no performance degradation, indicating that our Seasonal-Trend Re-parameterization can assist the fused model to successfully inherit the decomposition-related knowledge. Meanwhile, we observe that Seasonal-Trend Re-parameterization can bring further performance promotion in some datasets, since the trainable fusion factor $\lambda$ helps to fuse the seasonal and trend knowledge in a more appropriate proportion than the direct summation in explicit decomposition.

After equipped with all our designs, the final model can be seen as the enhanced version of the original model, enjoying better performance and having totally the same inference costs (e.g., in traffic dataset, it can boost the performance from 0.410 to 0.387 under the same inference time 322ms), which verifies the necessity and effectiveness of our decomposition-based enhancement framework.

### 4.5   Parameter Sensitivity

Moving average window size is the only tunable parameter for our framework, which is used in time series decomposition during our decoupled pretraining stage. We perform its parameter sensitivity study in Figure 7. The results show that our method is robust to the choice of moving average window size. And using the default value of 25 can provide ideal performance across various datasets.

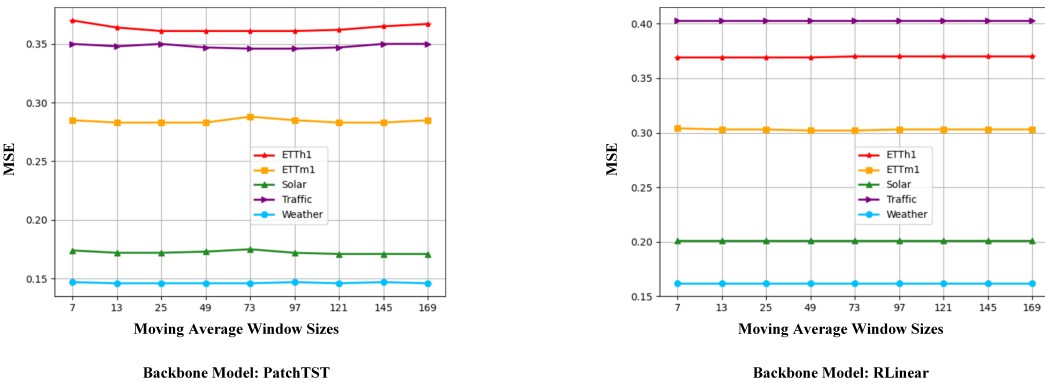

Figure 7: Parameter sensitivity. We conduct experiments under input-384-predict-96 settings, adopting PatchTST [30] and RLinear [14] as backbone models.

### 4.6   Training Efficiency

In addition to the excellent inference efficiency, our method also enjoys nice training efficiency. As stated in Section 3.3 and 3.4, the decoupled pretraining process for two expert models and the extra training process for fusion can converge quickly, **making the total training cost of our method comparable to the original model**. Please refer to Appendix E.1 for detailed experimental evidence.

## 5   Conclusion and Future Work

In this paper, we pioneer the idea of implicit decomposition and propose a powerful decomposition-based enhancement framework named **DecompNet**. Our method can enable a time series model to inherit the decomposition-related knowledge, even though this model is not actually decomposing the input time series. Therefore, our DecompNet can greatly enhance the forecasting performance of various time series models and bring no additional inference costs. Experimentally, DecompNet demonstrates excellent enhancement capability and framework generality, greatly pushing the performance limit of time series forecasting. Meanwhile, DecompNet also shows compelling performance and efficiency superiority, making the decomposition-based enhancement framework surpass the well-recognized normalization-based frameworks *for the first time*. We hope our exploration on implicit decomposition can provide some fresh perspectives and facilitate the future researches in both time series decomposition and time series enhancement frameworks.

## Acknowledgment

This work was supported by the Hebei Innovation Plan (20540301D).

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

# A  Details of Experimental Setup

**Datasets**  We conduct long-term forecasting experiments on 8 popular real-world benchmarks, including Weather [40], Traffic [33], ECL [36], Solar-Energy [29] and 4 ETT datasets [48]. More details of these datasets are provided in Appendix B.

**Backbone Models**  DecompNet is a model-agnostic enhancement framework and can be applied to any time series forecasters. To better prove our enhancement capability, we adopt following representative forecasters as strong backbone models, including the Transformer-based models: PatchTST [30], iTransformer [18]; the MLP-based models: RLinear, RMLP [14] and the Convolution-based model: ModernTCN [23]. The selection criterion for above backbone models are introduced in Appendix C.

**Comprehensive Comparison**  To better prove our comprehensive superiority, we compare our DecompNet with other time series enhancement frameworks in Section 4.2, including the popular normalization-based frameworks: Dish-TS [9], SAN [20], FAN [43] and the latest prior-based frameworks: SparseTSF [16], CycleNet [15].

And we also compare our idea of implicit decomposition with the previous idea of explicit decomposition in Section 4.3, including the latest and representative explicit decomposition methods: MICN [37] and Leddam [44].

More information of these methods adopted for comparisons are provided in Appendix F.3 and F.4.

**Evaluation**  We conduct long-term forecasting experiments for evaluation. Following the previous settings, we set prediction lengths as $\{96, 192, 336, 720\}$ and calculate the mean squared error (MSE) and mean absolute error (MAE) of multivariate time series forecasting as metrics.

Meanwhile, to truly reflect the enhancement effects, we should ensure that each backbone model has achieved its best performance. And according to previous findings [45, 14, 30, 21, 23, 22], most of the time series models achieve their best performance when the input length is longer than 300. Thus we fix the input length as 384 to truly unleash the potential of backbone models and obtain strong baselines, which can make our results more persuasive. More implementation details are in Appendix C.

# B  Datasets

Table 2: Detailed descriptions of multivariate datasets. The Dataset Size denotes the total number of time points in (Train, Validation, Test) split respectively.

| Dataset | Variates | Prediction Length | Dataset Size | Frequency | Information |
|---|---|---|---|---|---|
| ETTh1 | 7 | {96, 192, 336, 720} | (8545, 2881, 2881) | Hourly | Electricity |
| ETTh2 | 7 | {96, 192, 336, 720} | (8545, 2881, 2881) | Hourly | Electricity |
| ETTm1 | 7 | {96, 192, 336, 720} | (34465, 11521, 11521) | 15min | Electricity |
| ETTm2 | 7 | {96, 192, 336, 720} | (34465, 11521, 11521) | 15min | Electricity |
| Weather | 21 | {96, 192, 336, 720} | (36792, 5271, 10540) | 10min | Weather |
| Solar-Energy | 137 | {96, 192, 336, 720} | (36601, 5161, 10417) | 10min | Energy |
| ECL | 321 | {96, 192, 336, 720} | (18317, 2633, 5261) | Hourly | Electricity |
| Traffic | 862 | {96, 192, 336, 720} | (12185, 1757, 3509) | Hourly | Transportation |

We evaluate the multivariate long-term forecasting performance on 8 popular real-world datasets, including Weather, Traffic, ECL, Solar-energy and 4 ETT datasets (ETTh1, ETTh2, ETTm1, ETTm2). These datasets have been extensively utilized for benchmarking and cover many aspects of life.

The variate number, dataset size and sampling frequency of each dataset are summarized in Table 2 . We follow standard protocol [48] and split all datasets into training, validation and test set in chronological order by the ratio of 6:2:2 for the ETT datasets and 7:1:2 for the other datasets. And

training, validation and test sets are zero-mean normalized with the mean and standard deviation of training set. Each of above datasets only contains one continuous long time series, and we obtain samples by sliding window.

More introduction of the datasets are as follows:

1) **Weather**[1] contains 21 meteorological indicators of Germany in 2020.

2) **Traffic**[2] contains the road occupancy rates measured by 862 different sensors on San Francisco Bay area freeways in 2 years.

3) **ECL**(Electricity)[3] contains hourly electricity consumption of 321 clients from 2012 to 2014.

4) **ETT**(Electricity Transformer Temperature)[4] contains the data collected from two different electricity transformers (labeled as 1 and 2) with two different resolutions (denoted as m for 15 minutes and h for 1 hour) by 7 sensors. As a result, in total we have 4 ETT datasets: ETTh1, ETTh2, ETTm1, ETTm2.

5) **Solar**(Solar-Energy)[5] contains 137 time series representing the solar power production in Alabama state in 2006.

## C    Experiment Details

**Implementation Details**    Our method is trained with the L2 loss, using the ADAM [12] optimizer with an initial learning rate in $\{10^{-3}, 5 \times 10^{-4}, 10^{-4}\}$. The default training process is 30 epochs with proper early stopping (i.e., we set the maximum epoch as 30 and set the early stop patience as 3). The mean square error (MSE) and mean absolute error (MAE) are used as metrics. All the experiments are repeated 5 times with different seeds and the means of the metrics are reported as the final results. All the deep learning networks are implemented in PyTorch[32] and conducted on NVIDIA A100 40GB GPU.

**Selection Criterion for Backbone Models**    DecompNet is a model-agnostic framework and can be applied to any time series forecasters. To better prove our effectiveness, we adopt following representative forecasters as strong backbone models, including the Transformer-based models: PatchTST [30], iTransformer [18]; the MLP-based models: RLinear and RMLP [14] and the Convolution-based model: ModernTCN [23].

We choose above backbone models for following reasons: (i) the selected backbone models cover the mainstream types of time series forecasters, and the capability to enhance the performance of various types of models can prove our framework generality; (ii) the selected backbone models represent the state-of-the-art level in time series forecasting, and the capability to further enhance the performance of these state-of-the-art models can truly reflect our strong enhancement effects.

**Parameter Selection for Backbone Models**    For all backbone models, we follow the official codes in their original papers and search their model parameters from following searching space: number of layers $L$ from $\{1, 2, 3\}$, model dimension $D$ from $\{16, 32, 64, 128, 256\}$ and FFN expansion $\alpha$ from $\{1, 2, 4, 8\}$, which can fully unleash the performance potential of backbone models and provide a persuasive benchmark to truly reflect the enhancement effects of all enhancement frameworks.

We only search model parameters for the baseline backbone models. After the model parameters of a specific baseline backbone model have been determined, we directly apply our DecompNet framework into this baseline backbone model without additionally modifying its model parameters. Therefore, the model parameters in our method are the same as those in the baseline, which can ensure a fair comparision.

---

[1]https://www.bgc-jena.mpg.de/wetter/

[2]https://pems.dot.ca.gov/

[3]https://archive.ics.uci.edu/ml/datasets/ElectricityLoadDiagrams20112014

[4]https://github.com/zhouhaoyi/ETDataset

[5]https://www.nrel.gov/grid/solar-power-data.html

**Parameter Selection for Enhancement Frameworks**    By default, our DecompNet sets the moving average window size as 25 for time series decomposition during the decoupled pretraining stage. And we provide the parameter sensitivity study in Section 4.5. For other enhancement frameworks adopted for comparison, we follow their official codes with their recommended framework parameters in the original papers.

**Metric**    We adopt the mean square error (MSE) and mean absolute error (MAE) of multivariate time series forecasting as metrics.

$$\text{MSE} = \frac{1}{T} \sum_{i=0}^{T} (\widehat{\mathbf{Y}}_i - \mathbf{Y}_i)^2$$

$$\text{MAE} = \frac{1}{T} \sum_{i=0}^{T} \left| \widehat{\mathbf{Y}}_i - \mathbf{Y}_i \right|$$

where $\widehat{\mathbf{Y}}, \mathbf{Y} \in \mathbb{R}^{T \times M}$ are the $M$ variates prediction results of length $T$ and corresponding ground truth. $\widehat{\mathbf{Y}}_i$ means the $i$-th time point in the prediction result.

## D    More About Our Methods

**Advantages of Decoupled Training Strategy**    Our decoupled training strategy has following advantages: (i) *Better training results.* The goal of decomposition-based methods is to train two expert models specialized in the trend prediction task and seasonal prediction task, respectively. And our strategy directly trains the expert models on these two tasks, which can provide clearer supervision signals and bring better training results. For comparison, the previous joint training strategy trains the expert models on an indirect whole-series prediction task, which can only provide messily entangled supervision signals, limiting their performance (Figure 2 (b)). As a result, our strategy is more in line with the purpose of time series decomposition, therefore can fully unleash the performance potential of decomposition-based methods. (ii) *Faster convergence.* Since the ground truth $\mathbf{Y}$ is a raw time series entangled with various patterns, directly predicting the whole $\mathbf{Y}$ is more difficult [43, 20]. And our strategy can unravel $\mathbf{Y}$ into more predictable components and make the training process easier, contributing to faster convergence.

Please refer to Appendix E.1 and S.1 for more experimental verifications.

## E    More About Our Training Phase

### E.1    Training Efficiency

In this paper, we mainly focus on the inference efficiency for following reasons. Compared with the training phase which is usually performed on devices with substantial GPU resources, the inference phase is more likely to be conducted with limited computational budgets (e.g., deployed on resource-limited user devices). Therefore, focusing on and optimizing the inference cost can enable our method to be deployed in a wider range of real-world application scenarios (especially in resource-constrained environments) and thus increase the utility of our method. Under such consideration, we emphasize our excellent inference efficiency in this paper and highlight our inference-cost-free property as a key advantage that sets us apart from other competitors.

In addition to the excellent inference efficiency, **our method also enjoys nice training efficiency**. As stated in Section 3.3 and 3.4, the decoupled pretraining process for two expert models and the extra training process for fusion can converge quickly, **making the total training cost of our method comparable to the original model**.

We conduct experiments to verify our nice training efficiency. The experimental setups are as follows:

- We use PatchTST [30] as the backbone model. Results are recorded under input-384-predict-96 setting. For all training processes, the training setting is totally the same, where we set the maximum epoch as 30 and set the early stop patience as 3. And based on our observation,

the models can converge quickly in practice. So the actual training epochs consumed by a model are often less than the maximum number 30.

- For original model, we train it from scratch and report its total training epochs.

- For our method, we first pretrain the trend model and seasonal model respectively, then we conduct an extra training to fuse these two expert models. So the epochs of +*Ours* are recorded as (trend + seasonal + fuse = total).

- Due to the same structure of each model, the per epoch cost is similar. So the total training epochs can reflect the overall training cost.

We conduct experiments and report the training cost and model performance in Table 3. The results prove that:

- Although we pretrain two expert models during the decoupled pretraining process, the total training cost is not doubled. It is comparable or less than the original model.

- Although our method has an extra training process for fusion, it can converge quickly and bring very little training cost.

- As a result, **the total training epochs consumed by our method are comparable to the original model**.

- Considering the **comparable training cost**, the **totally same inference cost** and the **great performance improvement**, it can be proven that our method performs much better than the original model.

Table 3: Training cost comparison and model performance comparison. We use MSE to reflect the model performance and we use the total training epochs to reflect the overall training cost. A lower MSE and a less training cost indicate better forecasting performance and training efficiency. The best results are in **bold**.

| Dataset | ETTh1 | | ETTm1 | | Solar | | Traffic | |
|---|---|---|---|---|---|---|---|---|
| Metric | Epoch | MSE | Epoch | MSE | Epoch | MSE | Epoch | MSE |
| Original Backbone Model | 20 | 0.376 | **15** | 0.292 | 26 | 0.197 | **30** | 0.385 |
| **+Ours** | **(1+10+1=12)** | **0.361** | (4+12+3=19) | **0.283** | **(8+12+1=21)** | **0.172** | (10+18+3=31) | **0.350** |

## E.2 Ablation Study on Decoupled Pretraining Stage

We conduct ablation study to verify the necessity of our decoupled pretraining stage. As shown in Table 4, the removal of pretraining leads to performance degradation, validating the necessity of our decoupled pretraining stage. And this result is consistent with intuitive judgment: since the decoupled pretraining stage is a key process to inject the decomposition-related knowledge into models, it can not be removed.

Table 4: Ablation study on our decoupled pretraining stage. *Ours* indicates that $\mathbf{W}_s$ and $\mathbf{W}_t$ are firstly pretrained with decomposition-related knowledge before fusion. And *Without Pretraining* indicates that $\mathbf{W}_s$ and $\mathbf{W}_t$ are not pretrained with decomposition-related knowledge before fusion. They are just randomly initialized and directly used for fusion. Results are averaged from four prediction lengths $T \in \{96, 192, 336, 720\}$, using PatchTST [30] as the backbone model. A lower MSE or MAE indicates a better performance. The best results are in **bold**.

| Dataset | ETTh1 | | ETTm1 | | Solar | | Traffic | |
|---|---|---|---|---|---|---|---|---|
| Metric | MSE | MAE | MSE | MAE | MSE | MAE | MSE | MAE |
| **Ours** | **0.403** | **0.421** | **0.345** | **0.380** | **0.199** | **0.257** | **0.387** | **0.268** |
| Without Pretraining | 0.423 | 0.435 | 0.356 | 0.386 | 0.217 | 0.315 | 0.412 | 0.287 |

### E.3 Ablation Study on Learnable Fusion Factor $\lambda$

We conduct ablation study to verify the importance of our learnable fusion factor $\lambda$. As shown in Table 5, the removal of $\lambda$ degrades the performance, suggesting that it is vital to learn an optimal proportion $\lambda$ to fuse the seasonal and trend knowledge.

Table 5: Ablation study on learnable fusion factor $\lambda$. *Ours* indicates that we fuse $\mathbf{W}_s$ and $\mathbf{W}_t$ under the help of the learnable fusion factor $\lambda$, letting $\mathbf{W} = \mathbf{W}_s + \lambda \mathbf{W}_t$. And *Fusion Without $\lambda$* indicates that we fuse $\mathbf{W}_s$ and $\mathbf{W}_t$ without the help of the learnable fusion factor $\lambda$, only letting $\mathbf{W} = \mathbf{W}_s + \mathbf{W}_t$. Results are averaged from four prediction lengths $T \in \{96, 192, 336, 720\}$, using PatchTST [30] as the backbone model. A lower MSE or MAE indicates a better performance. The best results are in **bold**.

| Dataset | ETTh1 | | ETTm1 | | Solar | | Traffic | |
|---|---|---|---|---|---|---|---|---|
| Metric | MSE | MAE | MSE | MAE | MSE | MAE | MSE | MAE |
| **Ours** | **0.403** | **0.421** | **0.345** | **0.380** | **0.199** | **0.257** | **0.387** | **0.268** |
| Fusion Without $\lambda$ | 0.419 | 0.436 | 0.352 | 0.382 | 0.208 | 0.268 | 0.399 | 0.282 |

## F   More Related Works

### F.1   Other Procedure for Time Series Decomposition

As introduced in Section 2.1, the mainstream procedure of previous decomposition-based methods is to decompose the time series into seasonal and trend components and process them with two individual expert models respectively.

Apart from above mainstream procedure, there is also other type of procedure for decomposition-based methods, which is exemplified by Autoformer [41] and FEDformer [49]. They take time series decomposition as an inner block of the model structures and they decompose the time series at each layer, instead of at the beginning of the model. But this more sophisticated procedure is less straightforward and less performing, making it not the mainstream choice.

### F.2   Drawbacks of Explicit Decomposition

We denote the previous well-established procedures for decomposition-based methods as explicit decomposition, which meets several drawbacks:

(i) Previous methods need to explicitly decompose the input time series and use several expert models for inference, which brings heavier inference costs.

(ii) Previous methods advocate that the seasonal model and trend model should use different model structures, which makes it impossible to improve the efficiency by our Seasonal-Trend Re-parameterization.

(iii) Previous methods pay more attention on data processing, instead of model training. They prefer to design more sophisticated decomposition process (e.g., multi-kernel decomposition [37] or learnable decomposition [44]) to better decompose the input time series. But they ignore the importance of how to better train the two expert models. They simply use the sub-optimal joint training strategy to train the seasonal and trend models, making their performance potential not fully unleashed.

Due to above drawbacks, these previous explicit decomposition methods suffer from the performance and efficiency issues. And our implicit decomposition can handle these challenges from a brand new perspective, providing a better solution to apply decomposition in time series forecasting.

### F.3   Details of Explicit Decomposition Baselines in Section 4.3

MICN [37] adopts a sophisticated decomposition process (i.e., multi-kernel moving average decomposition) to better decompose the input time series. And it uses different model structures for two

expert models: Linear layer for trend part and multi-scale isometric convolution for seasonal part. And it is trained based on joint training strategy.

Leddam [44] adopts a sophisticated decomposition process (i.e., learnable decomposition) to better decompose the input time series. And it uses different model structures for two expert models: Linear layer for trend part and Dual Attention Module for seasonal part. And it is trained based on joint training strategy.

In Section 4.3, we modify MICN and Leddam with our implicit decomposition designs. In the first step, we compare the importance between data processing and model training. We replace their sophisticated decomposition processes (i.e., multi-kernel moving average decomposition in MICN and learnable decomposition in Leddam) with a simple single-kernel moving average decomposition. And we apply our decoupled training strategy to replace their original joint training strategy. In the second step, we re-examine the feasibility of using same model structure for two expert models. We adopt multi-scale isometric convolution for two expert models in MICN and adopt Dual Attention Module for two expert models in Leddam.

### F.4 Details of Enhancement Frameworks in Section 4.2

Normalization-based enhancement frameworks are developed based on RevIN [11]. RevIN proposes to first normalize the input time series with zero mean and unit standard deviation, and then add the mean and deviation back to the predicted outputs for de-normalization. RevIN puts an assumption that the input time series and the predicted outputs share the same statistical properties. Since this assumption may be at odds with reality, Dish-TS [9] further introduces an additional coefficient network to specifically learn the statistical properties for output series. As further improvement, SAN [20] explores statistical properties at a finer granularity (e.g., at the sliced level) and FAN [43] conducts normalization in frequency domain. Meanwhile, they also adopt a multi-stage and multi-target training strategy to unleash their performance potential, which is similar to our decoupled training strategy.

Prior-based enhancement frameworks can also improve the model performance, provided that the periodicity prior of a dataset is known in advance. Given the priori periodic length $P$ of a dataset, SparseTSF [16] proposes a cross-period sparse forecasting technique, which will downsample the original series into several subsequences based on the value of $P$ and apply a model with shared parameters to these subsequences for prediction. And CycleNet [15] utilizes an additional length-$P$ learnable matrix to specifically model the shared periodic patterns of the given dataset.

## G More Ablation Results

More ablation results are provided in Figure 8 and 9, which are using PatchTST [30] and RLinear [14] as the original backbone models, respectively.

## H Error Bar

We report the standard deviation of DecompNet performance under five runs with different random seeds in Table 6, using PatchTST [30] as the original backbone model. The results exhibit that the performance of DecompNet is stable.

## I Full Error Bar on More Backbone Models

The full results with full error bars of performance promotion obtained by our DecompNet framework are provided in Table 7. It is shown that the average performance improvement is roughly an order of magnitude larger than the random fluctuation from multiple runs (i.e., 0.0x vs 0.00x). Thus, the improvement in our main result (Table 1) is meaningful and not due to chance.

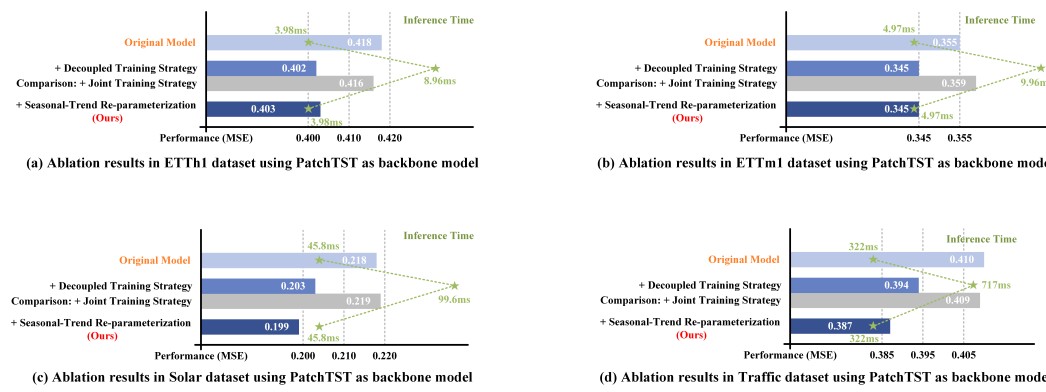

Figure 8: More ablation study results using PatchTST [30] as the original backbone model. From the top to the bottom, each row means one design we add on the original model to enhance it. We report the averaged MSE of four prediction lengths. The inference time is recorded under input-384-predict-96 setting per iteration. A lower MSE (a shorter blue bar) and a smaller inference time (a green star closer to the vertical axis) indicate better performance and efficiency.

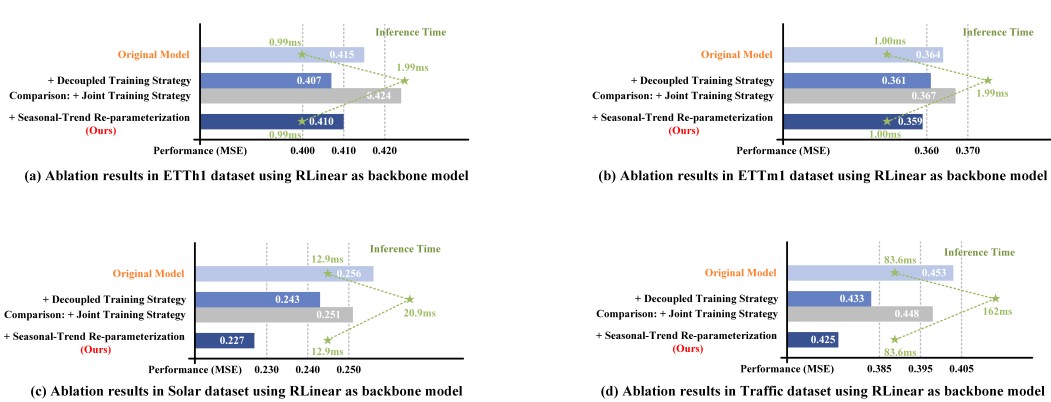

Figure 9: More ablation study results using RLinear [14] as the original backbone model. From the top to the bottom, each row means one design we add on the original model to enhance it. We report the averaged MSE of four prediction lengths. The inference time is recorded under input-384-predict-96 setting per iteration. A lower MSE (a shorter blue bar) and a smaller inference time (a green star closer to the vertical axis) indicate better performance and efficiency.

## J   How Robust is DecompNet to Non-stationary Test-time Inputs?

DecompNet can robustly handle non-stationary test-time inputs. And we compare our DecompNet with the popular RevIN [11] from the perspectives of mechanism guarantees and empirical evidence:

Table 6: Error bar.

| Dataset | ETTh1 | | ETTh2 | | ETTm1 | | ETTm2 | |
|---|---|---|---|---|---|---|---|---|
| Horizon | MSE | MAE | MSE | MAE | MSE | MAE | MSE | MAE |
| 96 | 0.361±0.000 | 0.390±0.001 | 0.277±0.001 | 0.337±0.001 | 0.283±0.001 | 0.337±0.001 | 0.166±0.000 | 0.254±0.000 |
| 192 | 0.391±0.001 | 0.410±0.001 | 0.351±0.002 | 0.386±0.001 | 0.328±0.001 | 0.368±0.001 | 0.222±0.001 | 0.293±0.001 |
| 336 | 0.418±0.003 | 0.428±0.002 | 0.371±0.002 | 0.399±0.002 | 0.365±0.002 | 0.392±0.002 | 0.272±0.001 | 0.325±0.001 |
| 720 | 0.442±0.004 | 0.455±0.003 | 0.386±0.003 | 0.425±0.002 | 0.405±0.002 | 0.421±0.002 | 0.366±0.002 | 0.385±0.002 |

| Dataset | Weather | | Solar-Energy | | ECL | | Traffic | |
|---|---|---|---|---|---|---|---|---|
| Horizon | MSE | MAE | MSE | MAE | MSE | MAE | MSE | MAE |
| 96 | 0.146±0.000 | 0.194±0.000 | 0.172±0.000 | 0.229±0.000 | 0.133±0.000 | 0.233±0.001 | 0.350±0.001 | 0.247±0.001 |
| 192 | 0.190±0.001 | 0.238±0.000 | 0.197±0.001 | 0.254±0.001 | 0.149±0.001 | 0.250±0.001 | 0.374±0.002 | 0.263±0.001 |
| 336 | 0.243±0.001 | 0.279±0.001 | 0.211±0.001 | 0.270±0.001 | 0.163±0.001 | 0.265±0.001 | 0.389±0.001 | 0.268±0.001 |
| 720 | 0.317±0.001 | 0.332±0.001 | 0.216±0.002 | 0.274±0.002 | 0.199±0.001 | 0.293±0.001 | 0.435±0.003 | 0.293±0.002 |

Table 7: Full results with full error bars of performance promotion obtained by our DecompNet framework. We conduct experiments in long-term forecasting tasks and adopt the mainstream state-of-the-art time series models as backbone models. The input length is fixed as 384 and the prediction lengths are in $T \in \{96, 192, 336, 720\}$. *Avg* means the average results from all four prediction lengths. A lower MSE or MAE indicates a better performance and the best results are in **bold**.

| Models | PatchTST | | | | iTransformer | | | | RLinear | | | | RMLP | | | | ModernTCN | | | |
|---|---|---|---|---|---|---|---|---|---|---|---|---|---|---|---|---|---|---|---|---|
| Settings | Original [30] | | +DecompNet (Ours) | | Original [18] | | +DecompNet (Ours) | | Original [14] | | +DecompNet (Ours) | | Original [14] | | +DecompNet (Ours) | | Original [23] | | +DecompNet (Ours) | |
| Metric | MSE | MAE | MSE | MAE | MSE | MAE | MSE | MAE | MSE | MAE | MSE | MAE | MSE | MAE | MSE | MAE | MSE | MAE | MSE | MAE |
| ETTh1 96 | 0.376±0.001 | 0.401±0.000 | 0.361±0.000 | 0.390±0.001 | 0.413±0.000 | 0.427±0.000 | 0.379±0.000 | 0.404±0.000 | 0.372±0.001 | 0.397±0.001 | 0.369±0.000 | 0.392±0.000 | 0.395±0.001 | 0.414±0.001 | 0.375±0.001 | 0.399±0.001 | 0.368±0.002 | 0.390±0.001 | 0.355±0.000 | 0.382±0.000 |
| ETTh1 192 | 0.410±0.001 | 0.421±0.001 | 0.391±0.001 | 0.410±0.001 | 0.449±0.001 | 0.450±0.001 | 0.413±0.000 | 0.425±0.001 | 0.411±0.001 | 0.421±0.001 | 0.403±0.000 | 0.412±0.000 | 0.439±0.001 | 0.443±0.001 | 0.410±0.001 | 0.419±0.001 | 0.399±0.001 | 0.410±0.000 | 0.385±0.001 | 0.404±0.000 |
| ETTh1 336 | 0.434±0.003 | 0.439±0.002 | 0.418±0.003 | 0.428±0.002 | 0.455±0.002 | 0.458±0.002 | 0.429±0.002 | 0.441±0.002 | 0.425±0.001 | 0.429±0.001 | 0.427±0.001 | 0.427±0.001 | 0.499±0.002 | 0.479±0.002 | 0.437±0.002 | 0.435±0.001 | 0.394±0.001 | 0.415±0.001 | 0.376±0.002 | 0.407±0.002 |
| ETTh1 720 | 0.451±0.003 | 0.471±0.002 | 0.442±0.004 | 0.455±0.003 | 0.528±0.002 | 0.521±0.002 | 0.446±0.002 | 0.469±0.002 | 0.450±0.001 | 0.465±0.001 | 0.441±0.001 | 0.456±0.001 | 0.595±0.003 | 0.541±0.003 | 0.469±0.002 | 0.467±0.002 | 0.452±0.003 | 0.461±0.003 | 0.433±0.008 | 0.450±0.002 |
| ETTh1 Avg | 0.418±0.002 | 0.433±0.001 | 0.403±0.002 | 0.421±0.002 | 0.461±0.001 | 0.464±0.001 | 0.417±0.001 | 0.435±0.001 | 0.415±0.001 | 0.428±0.001 | 0.410±0.001 | 0.422±0.001 | 0.482±0.002 | 0.469±0.002 | 0.423±0.002 | 0.430±0.001 | 0.403±0.002 | 0.419±0.002 | 0.387±0.002 | 0.411±0.000 |
| ETTh2 96 | 0.275±0.001 | 0.337±0.001 | 0.277±0.001 | 0.337±0.001 | 0.306±0.002 | 0.363±0.001 | 0.279±0.001 | 0.340±0.001 | 0.278±0.001 | 0.338±0.001 | 0.272±0.001 | 0.334±0.001 | 0.302±0.002 | 0.358±0.002 | 0.283±0.001 | 0.341±0.001 | 0.263±0.001 | 0.332±0.001 | 0.255±0.001 | 0.329±0.001 |
| ETTh2 192 | 0.354±0.001 | 0.388±0.001 | 0.351±0.002 | 0.386±0.001 | 0.366±0.001 | 0.402±0.001 | 0.347±0.001 | 0.385±0.001 | 0.356±0.001 | 0.393±0.001 | 0.337±0.001 | 0.376±0.001 | 0.350±0.002 | 0.387±0.002 | 0.340±0.002 | 0.380±0.002 | 0.321±0.001 | 0.376±0.001 | 0.320±0.001 | 0.378±0.001 |
| ETTh2 336 | 0.368±0.001 | 0.403±0.002 | 0.371±0.002 | 0.399±0.002 | 0.400±0.002 | 0.427±0.001 | 0.356±0.002 | 0.404±0.002 | 0.364±0.001 | 0.408±0.001 | 0.356±0.001 | 0.397±0.001 | 0.387±0.001 | 0.416±0.001 | 0.367±0.002 | 0.405±0.002 | 0.310±0.001 | 0.374±0.001 | 0.290±0.002 | 0.363±0.001 |
| ETTh2 720 | 0.398±0.001 | 0.435±0.001 | 0.386±0.003 | 0.425±0.002 | 0.435±0.003 | 0.454±0.002 | 0.392±0.002 | 0.427±0.002 | 0.364±0.001 | 0.447±0.001 | 0.386±0.002 | 0.426±0.002 | 0.416±0.004 | 0.443±0.003 | 0.396±0.002 | 0.433±0.002 | 0.417±0.004 | 0.447±0.002 | 0.382±0.002 | 0.422±0.002 |
| ETTh2 Avg | 0.349±0.001 | 0.391±0.001 | 0.346±0.002 | 0.387±0.002 | 0.377±0.001 | 0.412±0.001 | 0.344±0.002 | 0.389±0.002 | 0.354±0.001 | 0.397±0.001 | 0.338±0.001 | 0.383±0.001 | 0.364±0.002 | 0.401±0.001 | 0.347±0.002 | 0.390±0.002 | 0.328±0.002 | 0.382±0.001 | 0.312±0.000 | 0.373±0.001 |
| ETTm1 96 | 0.292±0.001 | 0.344±0.001 | 0.283±0.001 | 0.337±0.001 | 0.314±0.001 | 0.366±0.001 | 0.299±0.001 | 0.354±0.001 | 0.316±0.001 | 0.355±0.001 | 0.303±0.000 | 0.344±0.000 | 0.309±0.001 | 0.356±0.001 | 0.297±0.001 | 0.351±0.001 | 0.303±0.001 | 0.351±0.001 | 0.297±0.001 | 0.343±0.001 |
| ETTm1 192 | 0.340±0.001 | 0.374±0.001 | 0.328±0.001 | 0.368±0.001 | 0.353±0.001 | 0.389±0.001 | 0.336±0.001 | 0.373±0.001 | 0.337±0.001 | 0.366±0.001 | 0.336±0.000 | 0.364±0.000 | 0.344±0.001 | 0.380±0.001 | 0.337±0.001 | 0.377±0.001 | 0.352±0.001 | 0.379±0.001 | 0.337±0.001 | 0.367±0.001 |
| ETTm1 336 | 0.367±0.002 | 0.394±0.001 | 0.365±0.002 | 0.392±0.000 | 0.383±0.001 | 0.405±0.001 | 0.372±0.001 | 0.392±0.001 | 0.375±0.001 | 0.389±0.001 | 0.370±0.001 | 0.384±0.001 | 0.383±0.001 | 0.403±0.001 | 0.364±0.001 | 0.388±0.001 | 0.375±0.001 | 0.392±0.001 | 0.372±0.001 | 0.390±0.001 |
| ETTm1 720 | 0.422±0.003 | 0.427±0.003 | 0.405±0.002 | 0.421±0.002 | 0.437±0.002 | 0.435±0.002 | 0.424±0.002 | 0.422±0.000 | 0.426±0.001 | 0.415±0.001 | 0.426±0.001 | 0.415±0.000 | 0.447±0.002 | 0.437±0.001 | 0.421±0.001 | 0.415±0.001 | 0.429±0.003 | 0.421±0.001 | 0.410±0.002 | 0.416±0.002 |
| ETTm1 Avg | 0.355±0.001 | 0.385±0.001 | 0.345±0.002 | 0.380±0.001 | 0.372±0.001 | 0.399±0.001 | 0.358±0.001 | 0.385±0.001 | 0.364±0.001 | 0.381±0.001 | 0.359±0.001 | 0.377±0.000 | 0.371±0.001 | 0.394±0.001 | 0.355±0.001 | 0.383±0.001 | 0.365±0.002 | 0.386±0.001 | 0.354±0.001 | 0.379±0.001 |
| ETTm2 96 | 0.166±0.001 | 0.255±0.001 | 0.166±0.000 | 0.254±0.000 | 0.179±0.000 | 0.271±0.000 | 0.172±0.000 | 0.265±0.000 | 0.164±0.000 | 0.253±0.000 | 0.164±0.000 | 0.253±0.000 | 0.180±0.001 | 0.267±0.001 | 0.171±0.000 | 0.260±0.000 | 0.169±0.001 | 0.259±0.001 | 0.170±0.001 | 0.257±0.001 |
| ETTm2 192 | 0.228±0.001 | 0.299±0.001 | 0.222±0.001 | 0.293±0.001 | 0.243±0.000 | 0.313±0.000 | 0.227±0.001 | 0.298±0.000 | 0.227±0.000 | 0.297±0.000 | 0.218±0.000 | 0.289±0.000 | 0.247±0.001 | 0.308±0.001 | 0.229±0.001 | 0.296±0.001 | 0.223±0.001 | 0.296±0.001 | 0.222±0.000 | 0.292±0.001 |
| ETTm2 336 | 0.280±0.001 | 0.331±0.001 | 0.272±0.001 | 0.325±0.001 | 0.295±0.001 | 0.345±0.000 | 0.278±0.001 | 0.331±0.000 | 0.275±0.000 | 0.329±0.001 | 0.272±0.001 | 0.324±0.001 | 0.289±0.001 | 0.341±0.001 | 0.279±0.001 | 0.332±0.001 | 0.288±0.002 | 0.339±0.001 | 0.268±0.001 | 0.324±0.001 |
| ETTm2 720 | 0.373±0.002 | 0.389±0.001 | 0.366±0.002 | 0.385±0.002 | 0.377±0.002 | 0.397±0.002 | 0.371±0.001 | 0.388±0.001 | 0.368±0.001 | 0.386±0.001 | 0.359±0.002 | 0.382±0.001 | 0.368±0.003 | 0.387±0.003 | 0.358±0.002 | 0.382±0.002 | 0.365±0.002 | 0.384±0.001 | 0.357±0.001 | 0.383±0.001 |
| ETTm2 Avg | 0.262±0.001 | 0.319±0.001 | 0.257±0.001 | 0.314±0.001 | 0.274±0.001 | 0.332±0.001 | 0.262±0.001 | 0.321±0.000 | 0.259±0.000 | 0.316±0.001 | 0.255±0.001 | 0.312±0.001 | 0.271±0.002 | 0.326±0.001 | 0.259±0.001 | 0.318±0.001 | 0.261±0.002 | 0.320±0.001 | 0.254±0.001 | 0.314±0.001 |
| Weather 96 | 0.150±0.000 | 0.201±0.000 | 0.146±0.000 | 0.194±0.000 | 0.160±0.000 | 0.210±0.000 | 0.155±0.000 | 0.207±0.000 | 0.174±0.000 | 0.227±0.000 | 0.162±0.000 | 0.217±0.000 | 0.169±0.000 | 0.220±0.000 | 0.162±0.000 | 0.214±0.000 | 0.151±0.000 | 0.205±0.000 | 0.147±0.000 | 0.199±0.000 |
| Weather 192 | 0.194±0.000 | 0.244±0.001 | 0.190±0.001 | 0.238±0.000 | 0.203±0.001 | 0.251±0.001 | 0.198±0.001 | 0.247±0.001 | 0.216±0.000 | 0.261±0.000 | 0.206±0.000 | 0.256±0.000 | 0.212±0.000 | 0.259±0.000 | 0.207±0.001 | 0.254±0.001 | 0.197±0.002 | 0.248±0.001 | 0.195±0.000 | 0.243±0.000 |
| Weather 336 | 0.243±0.001 | 0.279±0.001 | 0.243±0.001 | 0.279±0.001 | 0.252±0.001 | 0.287±0.001 | 0.245±0.001 | 0.284±0.001 | 0.263±0.001 | 0.294±0.001 | 0.253±0.000 | 0.290±0.000 | 0.260±0.001 | 0.293±0.000 | 0.258±0.000 | 0.293±0.000 | 0.251±0.001 | 0.287±0.001 | 0.253±0.000 | 0.286±0.000 |
| Weather 720 | 0.319±0.000 | 0.333±0.000 | 0.317±0.001 | 0.332±0.000 | 0.325±0.000 | 0.339±0.000 | 0.317±0.000 | 0.333±0.001 | 0.329±0.001 | 0.340±0.001 | 0.320±0.001 | 0.336±0.001 | 0.333±0.001 | 0.345±0.001 | 0.327±0.000 | 0.340±0.000 | 0.332±0.000 | 0.340±0.000 | 0.328±0.001 | 0.337±0.001 |
| Weather Avg | 0.227±0.000 | 0.264±0.000 | 0.224±0.001 | 0.261±0.000 | 0.235±0.001 | 0.272±0.000 | 0.229±0.001 | 0.268±0.001 | 0.246±0.000 | 0.281±0.001 | 0.235±0.000 | 0.275±0.000 | 0.244±0.001 | 0.279±0.000 | 0.239±0.000 | 0.273±0.001 | 0.253±0.001 | 0.270±0.001 | 0.231±0.000 | 0.266±0.000 |
| Solar-Energy 96 | 0.197±0.001 | 0.291±0.001 | 0.172±0.000 | 0.229±0.000 | 0.187±0.002 | 0.246±0.002 | 0.185±0.002 | 0.245±0.002 | 0.223±0.001 | 0.297±0.002 | 0.201±0.001 | 0.257±0.001 | 0.215±0.002 | 0.269±0.002 | 0.196±0.001 | 0.234±0.001 | 0.194±0.001 | 0.261±0.001 | 0.177±0.001 | 0.232±0.000 |
| Solar-Energy 192 | 0.214±0.001 | 0.319±0.002 | 0.192±0.001 | 0.254±0.000 | 0.209±0.002 | 0.269±0.001 | 0.209±0.002 | 0.270±0.001 | 0.251±0.001 | 0.312±0.002 | 0.231±0.001 | 0.274±0.001 | 0.230±0.003 | 0.298±0.003 | 0.220±0.002 | 0.252±0.002 | 0.216±0.002 | 0.317±0.001 | 0.194±0.000 | 0.251±0.000 |
| Solar-Energy 336 | 0.255±0.001 | 0.330±0.001 | 0.211±0.001 | 0.270±0.001 | 0.222±0.001 | 0.281±0.001 | 0.207±0.002 | 0.258±0.001 | 0.265±0.002 | 0.338±0.002 | 0.238±0.001 | 0.282±0.001 | 0.238±0.002 | 0.304±0.003 | 0.221±0.001 | 0.254±0.001 | 0.230±0.000 | 0.321±0.000 | 0.215±0.002 | 0.275±0.001 |
| Solar-Energy 720 | 0.225±0.001 | 0.329±0.001 | 0.216±0.002 | 0.274±0.002 | 0.231±0.001 | 0.291±0.001 | 0.214±0.000 | 0.263±0.000 | 0.284±0.002 | 0.345±0.003 | 0.238±0.002 | 0.282±0.002 | 0.239±0.003 | 0.306±0.003 | 0.220±0.000 | 0.256±0.002 | 0.233±0.001 | 0.333±0.000 | 0.213±0.002 | 0.272±0.002 |
| Solar-Energy Avg | 0.218±0.001 | 0.317±0.001 | 0.199±0.001 | 0.257±0.001 | 0.212±0.002 | 0.272±0.001 | 0.204±0.002 | 0.259±0.001 | 0.256±0.002 | 0.323±0.002 | 0.227±0.001 | 0.274±0.001 | 0.231±0.003 | 0.294±0.003 | 0.214±0.002 | 0.249±0.002 | 0.218±0.001 | 0.306±0.001 | 0.200±0.001 | 0.258±0.001 |
| ECL 96 | 0.133±0.001 | 0.232±0.001 | 0.133±0.000 | 0.233±0.001 | 0.140±0.000 | 0.240±0.000 | 0.133±0.000 | 0.230±0.000 | 0.151±0.000 | 0.251±0.001 | 0.140±0.000 | 0.233±0.000 | 0.144±0.000 | 0.239±0.000 | 0.134±0.000 | 0.229±0.000 | 0.137±0.000 | 0.234±0.000 | 0.133±0.000 | 0.233±0.000 |
| ECL 192 | 0.148±0.001 | 0.246±0.001 | 0.149±0.001 | 0.250±0.001 | 0.158±0.000 | 0.258±0.000 | 0.153±0.000 | 0.250±0.000 | 0.165±0.000 | 0.262±0.001 | 0.153±0.000 | 0.245±0.000 | 0.158±0.000 | 0.251±0.000 | 0.149±0.001 | 0.245±0.001 | 0.156±0.001 | 0.254±0.001 | 0.147±0.000 | 0.248±0.000 |
| ECL 336 | 0.169±0.001 | 0.265±0.000 | 0.165±0.001 | 0.265±0.001 | 0.175±0.001 | 0.275±0.001 | 0.165±0.001 | 0.264±0.000 | 0.179±0.001 | 0.277±0.001 | 0.164±0.001 | 0.262±0.001 | 0.174±0.001 | 0.270±0.001 | 0.167±0.000 | 0.265±0.001 | 0.171±0.001 | 0.273±0.001 | 0.164±0.001 | 0.264±0.001 |
| ECL 720 | 0.202±0.001 | 0.295±0.001 | 0.199±0.001 | 0.293±0.001 | 0.215±0.002 | 0.310±0.002 | 0.219±0.001 | 0.313±0.001 | 0.218±0.001 | 0.308±0.001 | 0.208±0.001 | 0.295±0.001 | 0.213±0.001 | 0.301±0.001 | 0.207±0.001 | 0.295±0.001 | 0.209±0.003 | 0.303±0.003 | 0.192±0.000 | 0.285±0.000 |
| ECL Avg | 0.163±0.001 | 0.260±0.001 | 0.161±0.000 | 0.260±0.001 | 0.172±0.001 | 0.271±0.001 | 0.168±0.001 | 0.264±0.001 | 0.178±0.001 | 0.275±0.001 | 0.168±0.001 | 0.259±0.001 | 0.172±0.001 | 0.265±0.001 | 0.165±0.000 | 0.258±0.000 | 0.168±0.002 | 0.264±0.001 | 0.159±0.001 | 0.258±0.001 |
| Traffic 96 | 0.385±0.001 | 0.277±0.001 | 0.350±0.001 | 0.247±0.001 | 0.393±0.001 | 0.298±0.001 | 0.357±0.001 | 0.261±0.001 | 0.433±0.001 | 0.305±0.001 | 0.403±0.001 | 0.277±0.001 | 0.404±0.001 | 0.294±0.001 | 0.389±0.001 | 0.280±0.001 | 0.407±0.002 | 0.283±0.002 | 0.363±0.001 | 0.255±0.001 |
| Traffic 192 | 0.401±0.000 | 0.283±0.000 | 0.374±0.002 | 0.263±0.001 | 0.414±0.002 | 0.312±0.001 | 0.387±0.001 | 0.276±0.001 | 0.441±0.001 | 0.307±0.001 | 0.416±0.001 | 0.288±0.001 | 0.439±0.001 | 0.310±0.001 | 0.403±0.001 | 0.289±0.001 | 0.407±0.002 | 0.287±0.002 | 0.380±0.002 | 0.267±0.001 |
| Traffic 336 | 0.410±0.002 | 0.287±0.002 | 0.389±0.001 | 0.268±0.001 | 0.432±0.003 | 0.322±0.001 | 0.387±0.001 | 0.279±0.002 | 0.453±0.001 | 0.313±0.002 | 0.424±0.001 | 0.285±0.001 | 0.439±0.001 | 0.310±0.001 | 0.415±0.001 | 0.285±0.001 | 0.416±0.003 | 0.292±0.003 | 0.393±0.002 | 0.271±0.000 |
| Traffic 720 | 0.443±0.003 | 0.304±0.002 | 0.435±0.003 | 0.293±0.002 | 0.472±0.003 | 0.347±0.001 | 0.437±0.002 | 0.305±0.002 | 0.483±0.001 | 0.331±0.002 | 0.456±0.002 | 0.303±0.001 | 0.476±0.001 | 0.327±0.001 | 0.444±0.001 | 0.301±0.001 | 0.452±0.004 | 0.309±0.003 | 0.438±0.003 | 0.308±0.001 |
| Traffic Avg | 0.410±0.002 | 0.288±0.001 | 0.387±0.002 | 0.268±0.001 | 0.428±0.002 | 0.320±0.001 | 0.395±0.002 | 0.280±0.002 | 0.453±0.001 | 0.314±0.002 | 0.425±0.001 | 0.286±0.001 | 0.436±0.001 | 0.308±0.001 | 0.413±0.001 | 0.285±0.001 | 0.417±0.003 | 0.293±0.002 | 0.394±0.002 | 0.275±0.001 |

## J.1 Mechanism Guarantees

DecompNet has mechanism to handle non-stationary test-time inputs, but the mechanism is different from RevIN's:

- RevIN is a normalization-based method. It alleviates the non-stationary issue from the perspective of statistical measures.

- DecompNet is a decomposition-based method. It handles the non-stationary issue based on time series decomposition, since decomposition can unravel the entangled temporal patterns in raw time series and extract the stationary components [10, 34, 41].

## J.2 Empirical Evidence

To prove our superiority in handling non-stationary issues, we compare our performance with RevIN in the top-3 highly non-stationary datasets, which are ETTh2, Exchange and ILI [13, 2, 48, 19].

For ILI, the input length is 96 and the results are averaged from 4 prediction lengths $\{24, 36, 48, 60\}$. For other datasets, the input length is 384 and the results are averaged from 4 prediction lengths $\{96, 192, 336, 720\}$.

As shown in Table 8, DecompNet outperforms RevIN in all these highly non-stationary datasets, indicating its better ability in handling non-stationary issues.

## J.3 Conclusion

As verified on the currently available and publicly accessible datasets in Section 4.2 and in Appendix J.2, DecompNet performs the best in handling non-stationary issues and is even better than RevIN and other normalization-based enhancement frameworks. These results indicate that DecompNet can robustly handle non-stationary test-time inputs.

Table 8: Performance comparison with RevIN in the top-3 highly non-stationary datasets. The *Original* model equips with RevIN by default. And *Ours* replaces the RevIN with our DecompNet framework. So these results can provide a fair comparison between our DecompNet and RevIN in handling non-stationary issues.

| Models | PatchTST | | iTransformer | | RLinear | | RMLP | | ModernTCN | |
|---|---|---|---|---|---|---|---|---|---|---|
| Settings | Original [30] | +DecompNet (Ours) | Original [18] | +DecompNet (Ours) | Original [14] | +DecompNet (Ours) | Original [14] | +DecompNet (Ours) | Original [23] | +DecompNet (Ours) |
| Metric | MSE MAE | MSE MAE | MSE MAE | MSE MAE | MSE MAE | MSE MAE | MSE MAE | MSE MAE | MSE MAE | MSE MAE |
| ETTh2 | 0.349 0.391 | **0.346 0.387** | 0.377 0.412 | **0.344 0.389** | 0.354 0.397 | **0.338 0.383** | 0.364 0.401 | **0.347 0.390** | 0.328 0.382 | **0.312 0.373** |
| Exchange | 0.387 0.419 | **0.330 0.404** | 0.378 0.418 | **0.317 0.391** | 0.378 0.411 | **0.314 0.386** | 0.385 0.414 | **0.320 0.385** | 0.376 0.407 | **0.319 0.376** |
| ILI | 2.096 0.969 | **1.904 0.886** | 2.231 1.020 | **1.959 0.934** | 3.686 1.366 | **2.305 1.042** | 2.796 1.172 | **2.056 0.972** | 1.944 1.198 | **1.738 0.874** |

We are well aware that there are non-stationary issues that cannot be represented by the current datasets. In future work, we will keep monitoring the development in time series community and promptly validate the effectiveness of our methods on the latest and more challenging non-stationary datasets.

# K  DecompNet's Generalization to Other Forms of Decomposition like STL and Scalability to More Than Two Decomposition Components

## K.1  Preliminary Exploration on the Scalability and Generalization of DecompNet

We conduct a preliminary exploration on the scalability and generalization of DecompNet. The experiment settings are as follows:

- We integrate our DecompNet with STL and decompose the time series into three finer components (i.e., seasonal, trend and residual components).
- We introduce more fusion factors to fuse three expert models, letting $W = W_s + \lambda_1 W_t + \lambda_2 W_r$, where $W_s, W_t, W_r$ are the weight matrices from seasonal, trend and residual expert models respectively.
- We use RLinear [14] and PatchTST [30] as the backbone models.

As shown in Table 9, after integrating DecompNet with more advanced decomposition methods and scaling into more and finer decomposed components, we observe continuous performance improvement, which can prove the scalability and generalization of DecompNet.

Table 9: Preliminary exploration on the scalability and generalization of DecompNet.

| Dataset Metric | ETTh1 | ETTm1 | Solar | Traffic |
|---|---|---|---|---|
| | MSE MAE | MSE MAE | MSE MAE | MSE MAE |
| PatchTST + 2 Components | 0.403 0.421 | 0.345 0.380 | 0.199 0.257 | 0.387 0.268 |
| PatchTST + 3 Components | 0.394 0.407 | 0.338 0.370 | 0.193 0.247 | 0.380 0.261 |
| RLinear + 2 Components | 0.410 0.422 | 0.359 0.377 | 0.227 0.274 | 0.425 0.286 |
| RLinear + 3 Components | 0.399 0.410 | 0.349 0.367 | 0.213 0.265 | 0.412 0.277 |

## K.2  Discussion on Potential Technical Challenges during Scaling

We discuss some potential technical challenges as follows, including increased fusion complexity, parameter growth, or optimization difficulty.

Based on our preliminary exploration, we find that:

- Fusion complexity and optimization difficulty are not increased. Through directly generalizing the equation from $W = W_s + \lambda W_t$ to $W = W_s + \lambda_1 W_t + \lambda_2 W_r$, we can achieve the fusion of three expert models. And the training process is still stable and can converge quickly.
- Parameter growth: The number of parameters is only increased during training. When scaling to more decomposition components, we need to train more expert models and introduces more fusion factors. This will introduce more parameters, but will also bring better performance. During inference, all expert models are fused as one single model, so there is no additional inference costs.

### K.3 Future Work

Above preliminary experiments demonstrate the scalability and generalization of DecompNet. In future work, we will conduct more in-depth researches to further unleash its scaling potential and report more insights on optimization.

## L Comparison with Other Prominent Decomposition-based Models

We compare our method with DLinear [45], NHiTs [3] and N-BEATS [31]. As shown in Table 10, our method surpasses these prominent decomposition-based models, achieving an average MSE reduction of 11.1% compared to DLinear, 16.7% compared to NHiTs and 18.6% compared to N-BEATS, thus verifying our performance superiority.

Table 10: Comparison with other prominent decomposition-based models like DLinear, NHiTs and N-BEATS.

| Dataset | ETTh1 | | ETTm1 | | Solar | | Traffic | |
|---|---|---|---|---|---|---|---|---|
| Metric | MSE | MAE | MSE | MAE | MSE | MAE | MSE | MAE |
| DecompNet + PatchTST | 0.403 | 0.421 | 0.345 | 0.380 | 0.199 | 0.257 | 0.387 | 0.268 |
| DecompNet + RLinear | 0.410 | 0.422 | 0.359 | 0.377 | 0.227 | 0.274 | 0.425 | 0.286 |
| DLinear | 0.428 | 0.444 | 0.365 | 0.385 | 0.255 | 0.338 | 0.453 | 0.318 |
| NHiTs | 0.486 | 0.477 | 0.372 | 0.397 | 0.265 | 0.348 | 0.479 | 0.327 |
| N-BEATS | 0.500 | 0.486 | 0.385 | 0.404 | 0.266 | 0.345 | 0.487 | 0.331 |

## M Enhancing the Performance of Classic Models (e.g., Standard RNN)

We utilize our DecompNet to enhance the performance of classic models (e.g., standard RNN). As shown in Table 11, our method can also work on the classic model and greatly improve its performance, making it catch up with the state-of-the-art level in time series forecasting.

Table 11: Enhancing the Performance of Classic Models (e.g., Standard RNN)

| Dataset | ETTh1 | | ETTm1 | | Solar | | Traffic | |
|---|---|---|---|---|---|---|---|---|
| Metric | MSE | MAE | MSE | MAE | MSE | MAE | MSE | MAE |
| RNN | 1.114 | 0.794 | 1.130 | 0.779 | 0.913 | 0.498 | 1.043 | 0.486 |
| RNN + Ours | 0.448 | 0.447 | 0.372 | 0.404 | 0.246 | 0.288 | 0.474 | 0.309 |

## N Reproducibility Statement

### N.1 Code Release

Our method is introduced in detail with equations and figures in the main text. All the implementation details are included in the Appendix, including dataset descriptions, evaluation metrics, model configurations and experiment settings. Code is available at this repository: `https://github.com/luodhhh/DecompNet`.

### N.2 Code Related to the Implementation Details of Model Fusion Process

The implementation of our fusion process are provided in the following two files within our repository (i.e., "utils/seasonal_trend_reparam.py" and "utils/seasonal_trend_reparam_for_Linear.py"). Please refer to these two files for implementation details of our fusion process.

## O More Discussion on the Choice of Moving Average Window Size

In this paper, we use a fixed moving average window size of 25 for following reasons:

- Experimental Fairness: The primary motivation for adopting a fixed window size of 25 is to ensure strict compliance with mainstream protocols. Starting from the famous decomposition-based methods like Autoformer and DLinear, it becomes a widely adopted protocol to use a fixed moving average window size of 25 for the decomposition implementations. Consequently, our main experiments adhere to this convention to ensure a fair experimental setting.
- Simplicity: In our humble opinion, minimizing the numbers of tunable hyperparameter is critical for real-world applicability, as it reduces the implementation complexity and enhances the adaptability across various real-world scenarios. Since the window size of 25 can already provide sufficient performance while aligning with mainstream conventions, we prioritize simplicity and directly use the fixed moving average window size of 25 without further tuning this hyperparameter.

To better choose the appropriate moving average window size:

- For the setting in experimental scenarios: We advocate using a fixed moving average window size of 25 for the consideration of simplicity and experimental fairness.
- For users seeking to maximize performance in their own real-world application scenarios: Although the window size of 25 can already provide sufficient performance, we are well aware that other window sizes may bring even better performance. To fully push the performance limit in specific real-world applications, we encourage the users to explore other moving average window sizes. And the powerful technique like automatic period detection can serve as an effective guidance for their further parameter adjustments.

## P Ethics Statement and Broader Impact

### P.1 Ethics Statement

Our work only focuses on the time series forecasting problem, so there is no potential ethical risk.

### P.2 Impact on Future Research

In this paper, we pioneer the idea of implicit decomposition, which is a brand new idea to apply decomposition in time series forecasting and provides guidance to upgrade the time series decomposition into a model-agnostic enhancement framework. Based on this idea, we establish a new category of decomposition-based enhancement framework and propose DecompNet as its representative, which demonstrates compelling enhancement capability, framework generality, performance superiority and efficiency superiority, making the decomposition-based enhancement framework surpass the well-recognized normalization-based frameworks *for the first time*. We hope our exploration on implicit decomposition can provide some fresh perspectives and facilitate the future researches in both time series decomposition and time series enhancement frameworks.

### P.3 Broader Impact on Real-world Applications

As an advanced model-agnostic time series enhancement framework, our method can enhance the forecasting performance of the latest and state-of-the-art time series models, greatly pushing the performance limit of time series forecasting. Therefore, our proposed method makes it promising to tackle real-world forecasting problem, helping our society make better decisions and prevent risks in advance.

Meanwhile, our DecompNet can enhance the model performance without introducing any additional inference costs, enjoying totally the same inference costs as the original backbone model. This inference-cost-free property allows for a wider application of our method, especially in some real-world scenarios that with limited computational budgets.

Our paper mainly focuses on scientific research and has no obvious negative social impact.

## Q   Limitations and Future Work

In this paper, we mainly implement our idea of implicit decomposition based on the mainstream moving average decomposition and we only decompose the time series into seasonal and trend components. Combining implicit decomposition with more advanced decomposition implementations (e.g., Seasonal-Trend decomposition using Loess (STL) [5] and frequency domain decomposition [38]) and decomposing the time series into finer components (e.g., seasonal, trend and residual components) can further improve the performance. It will be our future work to conduct these further explorations and further push the performance limit of our method.

In this paper, our fusion mechanism relies on strict layer-wise alignment. The prerequisite for the fusion process is that the two expert models must have the same model architecture. And it will be our future work to further study on how to handle the architectural mismatches during fusion and study on the fusion of models with different architectures.

In this paper, we mainly verify the effectiveness of our method through extensive empirical results and comprehensive experimental evidence. It will be our future work to conduct deeper analysis on model fusion and provide better theoretical explanation.

## R   Full Results

Due to the space limitation of the main text, we place the full results of all experiments in the following subsections. And we also provide the showcases in Appendix S.

### R.1   Performance Promotion by DecompNet Framework

The full results of performance promotion obtained by our DecompNet framework are provided in Table 12.

### R.2   Compared with Explicit Decomposition

The full results of performance comparison between our implicit decomposition and the previous explicit decomposition are provided in Table 13.

### R.3   Compared with Other Enhancement Frameworks

The full results of performance comparison between our DecompNet and other enhancement frameworks are provided in Table 14 and 15, which adopt RLinear [14] and PatchTST [30] as backbone models, respectively.

Table 12: Full results of performance promotion obtained by our DecompNet framework. We conduct experiments in long-term forecasting tasks and adopt the mainstream state-of-the-art time series models as backbone models. The input length is fixed as 384 and the prediction lengths are in $T \in \{96, 192, 336, 720\}$. *Avg* means the average results from all four prediction lengths. A lower MSE or MAE indicates a better performance and the best results are in **bold**.

| Models | | PatchTST | | | iTransformer | | | RLinear | | | RMLP | | | ModernTCN | | |
|---|---|---|---|---|---|---|---|---|---|---|---|---|---|---|---|---|
| Settings | | Original [30] | | +DecompNet (Ours) | | Original [18] | | +DecompNet (Ours) | | Original [14] | | +DecompNet (Ours) | | Original [14] | | +DecompNet (Ours) | | Original [23] | | +DecompNet (Ours) | |
| Metric | | MSE | MAE | MSE | MAE | MSE | MAE | MSE | MAE | MSE | MAE | MSE | MAE | MSE | MAE | MSE | MAE | MSE | MAE | MSE | MAE |
| ETTh1 | 96 | 0.376 | 0.401 | **0.361** | **0.390** | 0.413 | 0.427 | **0.379** | **0.404** | 0.372 | 0.397 | **0.369** | **0.392** | 0.395 | 0.414 | **0.375** | **0.399** | 0.368 | 0.390 | **0.355** | **0.382** |
| | 192 | 0.410 | 0.421 | **0.391** | **0.410** | 0.449 | 0.450 | **0.413** | **0.425** | 0.411 | 0.421 | **0.403** | **0.412** | 0.439 | 0.443 | **0.410** | **0.419** | 0.399 | 0.410 | **0.385** | **0.404** |
| | 336 | 0.434 | 0.439 | **0.418** | **0.428** | 0.455 | 0.458 | **0.429** | **0.441** | 0.425 | 0.429 | **0.427** | **0.427** | 0.499 | 0.479 | **0.437** | **0.435** | 0.394 | 0.415 | **0.376** | **0.407** |
| | 720 | 0.451 | 0.471 | **0.442** | **0.455** | 0.528 | 0.521 | **0.446** | **0.469** | 0.450 | 0.465 | **0.441** | **0.456** | 0.595 | 0.541 | **0.469** | **0.467** | 0.452 | 0.461 | **0.433** | **0.450** |
| | Avg | 0.418 | 0.433 | **0.403** | **0.421** | 0.461 | 0.464 | **0.417** | **0.435** | 0.415 | 0.428 | **0.410** | **0.422** | 0.482 | 0.469 | **0.423** | **0.430** | 0.403 | 0.419 | **0.387** | **0.411** |
| ETTh2 | 96 | **0.275** | **0.337** | 0.277 | **0.337** | 0.306 | 0.363 | **0.279** | **0.340** | 0.278 | 0.338 | **0.272** | **0.334** | 0.302 | 0.358 | **0.283** | **0.341** | 0.263 | 0.332 | **0.255** | **0.329** |
| | 192 | 0.354 | 0.388 | **0.351** | **0.386** | 0.366 | 0.402 | **0.347** | **0.385** | 0.356 | 0.393 | **0.337** | **0.376** | 0.350 | 0.387 | **0.340** | **0.380** | 0.321 | 0.376 | **0.320** | **0.378** |
| | 336 | **0.368** | 0.403 | 0.371 | **0.399** | 0.400 | 0.427 | **0.356** | **0.404** | 0.364 | 0.408 | **0.356** | **0.397** | 0.387 | 0.416 | **0.367** | **0.405** | 0.310 | 0.374 | **0.290** | **0.363** |
| | 720 | 0.398 | 0.435 | **0.386** | **0.425** | 0.435 | 0.454 | **0.392** | **0.427** | 0.419 | 0.447 | **0.386** | **0.426** | 0.416 | 0.443 | **0.396** | **0.433** | 0.417 | 0.447 | **0.382** | **0.422** |
| | Avg | 0.349 | 0.391 | **0.346** | **0.387** | 0.377 | 0.412 | **0.344** | **0.389** | 0.354 | 0.397 | **0.338** | **0.383** | 0.364 | 0.401 | **0.347** | **0.390** | 0.328 | 0.382 | **0.312** | **0.373** |
| ETTm1 | 96 | 0.292 | 0.344 | **0.283** | **0.337** | 0.314 | 0.366 | **0.299** | **0.354** | 0.316 | 0.355 | **0.303** | **0.344** | 0.309 | 0.356 | **0.297** | **0.351** | 0.303 | 0.351 | **0.297** | **0.343** |
| | 192 | 0.340 | 0.374 | **0.328** | **0.368** | 0.353 | 0.389 | **0.336** | **0.373** | 0.337 | 0.366 | **0.336** | **0.364** | 0.344 | 0.380 | **0.337** | **0.377** | 0.352 | 0.379 | **0.337** | **0.367** |
| | 336 | 0.367 | 0.394 | **0.365** | **0.392** | 0.383 | 0.405 | **0.372** | **0.392** | 0.375 | 0.389 | **0.370** | **0.384** | 0.383 | 0.403 | **0.364** | **0.388** | 0.375 | 0.392 | **0.372** | **0.390** |
| | 720 | 0.422 | 0.427 | **0.405** | **0.421** | 0.437 | 0.435 | **0.424** | **0.422** | 0.426 | 0.415 | **0.426** | **0.415** | 0.447 | 0.437 | **0.421** | **0.415** | 0.429 | 0.421 | **0.410** | **0.416** |
| | Avg | 0.355 | 0.385 | **0.345** | **0.380** | 0.372 | 0.399 | **0.358** | **0.385** | 0.364 | 0.381 | **0.359** | **0.377** | 0.371 | 0.394 | **0.355** | **0.383** | 0.365 | 0.386 | **0.354** | **0.379** |
| ETTm2 | 96 | 0.166 | 0.255 | **0.166** | **0.254** | 0.179 | 0.271 | **0.172** | **0.265** | **0.164** | **0.253** | 0.164 | 0.253 | 0.180 | 0.267 | **0.171** | **0.260** | 0.169 | 0.259 | **0.170** | **0.257** |
| | 192 | 0.228 | 0.299 | **0.222** | **0.293** | 0.243 | 0.313 | **0.227** | **0.298** | 0.227 | 0.297 | **0.218** | **0.289** | 0.247 | 0.308 | **0.228** | **0.299** | 0.223 | 0.296 | **0.222** | **0.292** |
| | 336 | 0.280 | 0.331 | **0.272** | **0.325** | 0.295 | 0.345 | **0.278** | **0.331** | 0.275 | 0.329 | **0.272** | **0.324** | 0.289 | 0.341 | **0.279** | **0.332** | 0.288 | 0.339 | **0.268** | **0.324** |
| | 720 | 0.373 | 0.389 | **0.366** | **0.385** | 0.377 | 0.397 | **0.371** | **0.388** | 0.368 | 0.386 | **0.365** | **0.382** | 0.368 | 0.387 | **0.358** | **0.382** | 0.365 | 0.386 | **0.357** | **0.383** |
| | Avg | 0.262 | 0.319 | **0.257** | **0.314** | 0.274 | 0.332 | **0.262** | **0.321** | 0.259 | 0.316 | **0.255** | **0.312** | 0.271 | 0.326 | **0.259** | **0.318** | 0.261 | 0.320 | **0.254** | **0.314** |
| Weather | 96 | 0.150 | 0.201 | **0.146** | **0.194** | 0.160 | 0.210 | **0.155** | **0.207** | 0.174 | 0.227 | **0.162** | **0.217** | 0.169 | 0.220 | **0.162** | **0.214** | 0.151 | 0.205 | **0.147** | **0.199** |
| | 192 | 0.194 | 0.244 | **0.190** | **0.238** | 0.203 | 0.251 | **0.198** | **0.247** | 0.216 | 0.261 | **0.206** | **0.256** | 0.212 | 0.259 | **0.207** | **0.254** | 0.197 | 0.248 | **0.195** | **0.243** |
| | 336 | **0.243** | **0.279** | 0.243 | 0.279 | 0.252 | 0.287 | **0.245** | **0.284** | 0.263 | 0.294 | **0.253** | **0.290** | 0.260 | 0.293 | **0.258** | **0.293** | 0.251 | 0.287 | 0.253 | **0.286** |
| | 720 | 0.319 | 0.333 | **0.317** | **0.332** | 0.325 | 0.339 | **0.317** | **0.333** | 0.329 | 0.340 | **0.320** | **0.336** | 0.333 | 0.345 | **0.327** | **0.340** | 0.332 | 0.340 | **0.328** | **0.337** |
| | Avg | 0.227 | 0.264 | **0.224** | **0.261** | 0.235 | 0.272 | **0.229** | **0.268** | 0.246 | 0.281 | **0.235** | **0.275** | 0.244 | 0.279 | **0.239** | **0.275** | 0.233 | 0.270 | **0.231** | **0.266** |
| Solar-Energy | 96 | 0.197 | 0.291 | **0.172** | **0.229** | 0.187 | 0.246 | **0.185** | **0.245** | 0.223 | 0.297 | **0.201** | **0.257** | 0.215 | 0.269 | **0.196** | **0.234** | 0.194 | 0.261 | **0.177** | **0.232** |
| | 192 | 0.214 | 0.319 | **0.197** | **0.254** | 0.209 | 0.269 | **0.209** | 0.270 | 0.251 | 0.312 | **0.231** | **0.274** | 0.230 | 0.298 | **0.220** | **0.252** | 0.216 | 0.317 | **0.194** | **0.251** |
| | 336 | 0.235 | 0.328 | **0.211** | **0.270** | 0.222 | 0.281 | **0.207** | **0.258** | 0.265 | 0.338 | **0.238** | **0.282** | 0.238 | 0.304 | **0.221** | **0.254** | 0.230 | 0.321 | **0.215** | **0.275** |
| | 720 | 0.225 | 0.329 | **0.216** | **0.274** | 0.231 | 0.291 | **0.214** | **0.263** | 0.284 | 0.345 | **0.238** | **0.282** | 0.239 | 0.306 | **0.220** | **0.256** | 0.233 | 0.323 | **0.213** | **0.272** |
| | Avg | 0.218 | 0.317 | **0.199** | **0.257** | 0.212 | 0.272 | **0.204** | **0.259** | 0.256 | 0.323 | **0.227** | **0.274** | 0.231 | 0.294 | **0.214** | **0.249** | 0.218 | 0.306 | **0.200** | **0.258** |
| ECL | 96 | **0.133** | **0.232** | 0.133 | 0.233 | 0.140 | 0.240 | **0.133** | **0.230** | 0.151 | 0.251 | **0.140** | **0.233** | 0.144 | 0.239 | **0.134** | **0.229** | 0.137 | 0.234 | **0.133** | **0.233** |
| | 192 | **0.148** | **0.246** | 0.149 | 0.250 | 0.158 | 0.258 | **0.153** | **0.250** | 0.163 | 0.262 | **0.153** | **0.245** | 0.158 | 0.251 | **0.151** | **0.245** | 0.156 | 0.254 | **0.147** | **0.248** |
| | 336 | 0.169 | **0.265** | **0.163** | 0.265 | 0.175 | 0.275 | **0.165** | **0.264** | 0.179 | 0.277 | **0.170** | **0.262** | 0.174 | 0.270 | **0.167** | **0.261** | 0.168 | **0.265** | **0.164** | 0.266 |
| | 720 | 0.202 | 0.295 | **0.199** | **0.293** | 0.215 | 0.310 | 0.219 | 0.313 | 0.218 | 0.308 | **0.208** | **0.295** | 0.213 | 0.301 | **0.207** | **0.295** | 0.209 | 0.303 | **0.192** | **0.285** |
| | Avg | 0.163 | **0.260** | **0.161** | 0.260 | 0.172 | 0.271 | **0.168** | **0.264** | 0.178 | 0.275 | **0.168** | **0.259** | 0.172 | 0.265 | **0.165** | **0.258** | 0.168 | 0.264 | **0.159** | **0.258** |
| Traffic | 96 | 0.385 | 0.277 | **0.350** | **0.247** | 0.393 | 0.298 | **0.357** | **0.261** | 0.433 | 0.305 | **0.403** | **0.277** | 0.404 | 0.294 | **0.389** | **0.275** | 0.391 | 0.283 | **0.363** | **0.255** |
| | 192 | 0.401 | 0.283 | **0.374** | **0.263** | 0.414 | 0.312 | **0.387** | **0.276** | 0.441 | 0.307 | **0.416** | **0.280** | 0.423 | 0.302 | **0.403** | **0.280** | 0.407 | 0.287 | **0.380** | **0.267** |
| | 336 | 0.410 | 0.287 | **0.389** | **0.268** | 0.432 | 0.322 | **0.397** | **0.279** | 0.453 | 0.313 | **0.424** | **0.285** | 0.439 | 0.310 | **0.415** | **0.285** | 0.416 | 0.292 | **0.393** | **0.271** |
| | 720 | 0.443 | 0.304 | **0.435** | **0.293** | 0.472 | 0.347 | **0.437** | **0.305** | 0.483 | 0.331 | **0.456** | **0.303** | 0.476 | 0.327 | **0.444** | **0.301** | 0.452 | 0.309 | **0.438** | **0.308** |
| | Avg | 0.410 | 0.288 | **0.387** | **0.268** | 0.428 | 0.320 | **0.395** | **0.280** | 0.453 | 0.314 | **0.425** | **0.286** | 0.436 | 0.308 | **0.413** | **0.285** | 0.417 | 0.293 | **0.394** | **0.275** |

Table 13: Full performance comparison between our implicit decomposition and the previous explicit decomposition. We conduct experiments in long-term forecasting tasks. The input length is fixed as 384 and the prediction lengths are in $T \in \{96, 192, 336, 720\}$. And we also gradually modify the explicit decomposition methods with our implicit decomposition designs. *Avg* means the average results from all four prediction lengths. A lower MSE or MAE indicates a better performance. And *Total Avg* means the further average results from all eight datasets.

| Type | Implicit Decomposition (Ours) | | | | Explicit Decomposition | | | | | | | | | | | |
| Frameworks | DecompNet (Ours) | | | | MICN [37] | | | | | | Leddam [44] | | | | | |
| Settings | DecompNet +PatchTST | | DecompNet +RLinear | | Original [37] | | Use Decoupled Training +Our Design | | Use Same Structure +Our Design | | Original [44] | | Use Decoupled Training +Our Design | | Use Same Structure +Our Design | |
| Metric | MSE | MAE | MSE | MAE | MSE | MAE | MSE | MAE | MSE | MAE | MSE | MAE | MSE | MAE | MSE | MAE |
|---|---|---|---|---|---|---|---|---|---|---|---|---|---|---|---|---|
| ETTh1 96 | 0.361 | 0.390 | 0.369 | 0.392 | 0.408 | 0.434 | 0.396 | 0.417 | 0.400 | 0.420 | 0.381 | 0.403 | 0.375 | 0.396 | 0.373 | 0.394 |
| 192 | 0.391 | 0.410 | 0.403 | 0.412 | 0.421 | 0.441 | 0.412 | 0.431 | 0.413 | 0.429 | 0.419 | 0.427 | 0.411 | 0.424 | 0.413 | 0.423 |
| 336 | 0.418 | 0.428 | 0.427 | 0.427 | 0.461 | 0.470 | 0.440 | 0.447 | 0.445 | 0.450 | 0.439 | 0.443 | 0.429 | 0.436 | 0.432 | 0.438 |
| 720 | 0.442 | 0.455 | 0.441 | 0.456 | 0.695 | 0.629 | 0.515 | 0.489 | 0.514 | 0.485 | 0.463 | 0.474 | 0.440 | 0.463 | 0.441 | 0.464 |
| Avg | 0.403 | 0.421 | 0.410 | 0.422 | 0.496 | 0.494 | 0.441 | 0.446 | 0.443 | 0.446 | 0.426 | 0.437 | 0.414 | 0.430 | 0.415 | 0.430 |
| ETTh2 96 | 0.277 | 0.337 | 0.272 | 0.334 | 0.306 | 0.377 | 0.281 | 0.343 | 0.285 | 0.346 | 0.308 | 0.358 | 0.275 | 0.336 | 0.278 | 0.340 |
| 192 | 0.351 | 0.386 | 0.337 | 0.376 | 0.423 | 0.447 | 0.351 | 0.390 | 0.351 | 0.391 | 0.370 | 0.398 | 0.337 | 0.376 | 0.336 | 0.378 |
| 336 | 0.371 | 0.399 | 0.356 | 0.397 | 0.555 | 0.532 | 0.365 | 0.406 | 0.364 | 0.403 | 0.405 | 0.426 | 0.355 | 0.395 | 0.356 | 0.394 |
| 720 | 0.386 | 0.425 | 0.386 | 0.426 | 0.965 | 0.738 | 0.393 | 0.432 | 0.394 | 0.433 | 0.446 | 0.461 | 0.382 | 0.423 | 0.383 | 0.424 |
| Avg | 0.346 | 0.387 | 0.338 | 0.383 | 0.562 | 0.524 | 0.348 | 0.393 | 0.349 | 0.393 | 0.382 | 0.411 | 0.337 | 0.383 | 0.338 | 0.384 |
| ETTm1 96 | 0.283 | 0.337 | 0.303 | 0.344 | 0.321 | 0.368 | 0.304 | 0.350 | 0.305 | 0.352 | 0.317 | 0.362 | 0.301 | 0.347 | 0.298 | 0.345 |
| 192 | 0.328 | 0.368 | 0.336 | 0.364 | 0.360 | 0.392 | 0.344 | 0.374 | 0.346 | 0.372 | 0.357 | 0.386 | 0.338 | 0.370 | 0.336 | 0.372 |
| 336 | 0.365 | 0.392 | 0.370 | 0.384 | 0.429 | 0.432 | 0.379 | 0.395 | 0.376 | 0.393 | 0.389 | 0.407 | 0.371 | 0.388 | 0.372 | 0.390 |
| 720 | 0.405 | 0.421 | 0.426 | 0.415 | 0.474 | 0.459 | 0.431 | 0.423 | 0.434 | 0.427 | 0.442 | 0.435 | 0.424 | 0.418 | 0.420 | 0.414 |
| Avg | 0.345 | 0.380 | 0.359 | 0.377 | 0.396 | 0.413 | 0.365 | 0.386 | 0.365 | 0.386 | 0.376 | 0.398 | 0.359 | 0.381 | 0.357 | 0.380 |
| ETTm2 96 | 0.166 | 0.254 | 0.164 | 0.253 | 0.187 | 0.279 | 0.165 | 0.255 | 0.168 | 0.256 | 0.185 | 0.267 | 0.165 | 0.254 | 0.170 | 0.257 |
| 192 | 0.222 | 0.293 | 0.218 | 0.289 | 0.265 | 0.341 | 0.222 | 0.294 | 0.221 | 0.293 | 0.251 | 0.310 | 0.220 | 0.291 | 0.223 | 0.290 |
| 336 | 0.272 | 0.325 | 0.272 | 0.324 | 0.333 | 0.383 | 0.275 | 0.328 | 0.278 | 0.328 | 0.317 | 0.353 | 0.272 | 0.325 | 0.272 | 0.324 |
| 720 | 0.366 | 0.385 | 0.365 | 0.382 | 0.466 | 0.466 | 0.368 | 0.384 | 0.367 | 0.382 | 0.379 | 0.394 | 0.364 | 0.381 | 0.369 | 0.385 |
| Avg | 0.257 | 0.314 | 0.255 | 0.312 | 0.313 | 0.367 | 0.258 | 0.315 | 0.259 | 0.315 | 0.283 | 0.331 | 0.255 | 0.313 | 0.259 | 0.314 |
| Weather 96 | 0.146 | 0.194 | 0.162 | 0.217 | 0.178 | 0.246 | 0.173 | 0.226 | 0.159 | 0.215 | 0.156 | 0.206 | 0.152 | 0.205 | 0.149 | 0.201 |
| 192 | 0.190 | 0.238 | 0.206 | 0.256 | 0.228 | 0.292 | 0.218 | 0.263 | 0.209 | 0.257 | 0.199 | 0.247 | 0.196 | 0.241 | 0.191 | 0.239 |
| 336 | 0.243 | 0.279 | 0.253 | 0.290 | 0.319 | 0.372 | 0.264 | 0.296 | 0.265 | 0.297 | 0.249 | 0.286 | 0.242 | 0.279 | 0.238 | 0.275 |
| 720 | 0.317 | 0.332 | 0.320 | 0.336 | 0.377 | 0.408 | 0.332 | 0.344 | 0.336 | 0.345 | 0.333 | 0.339 | 0.320 | 0.332 | 0.311 | 0.324 |
| Avg | 0.224 | 0.261 | 0.235 | 0.275 | 0.276 | 0.330 | 0.247 | 0.282 | 0.242 | 0.279 | 0.234 | 0.270 | 0.228 | 0.264 | 0.222 | 0.260 |
| Solar-Energy 96 | 0.172 | 0.229 | 0.201 | 0.257 | 0.218 | 0.266 | 0.198 | 0.257 | 0.188 | 0.244 | 0.201 | 0.267 | 0.184 | 0.245 | 0.176 | 0.231 |
| 192 | 0.197 | 0.254 | 0.231 | 0.274 | 0.250 | 0.299 | 0.232 | 0.273 | 0.234 | 0.278 | 0.222 | 0.280 | 0.206 | 0.260 | 0.196 | 0.246 |
| 336 | 0.211 | 0.270 | 0.238 | 0.282 | 0.267 | 0.312 | 0.250 | 0.283 | 0.250 | 0.288 | 0.233 | 0.293 | 0.215 | 0.279 | 0.212 | 0.271 |
| 720 | 0.216 | 0.274 | 0.238 | 0.282 | 0.268 | 0.313 | 0.255 | 0.281 | 0.256 | 0.281 | 0.245 | 0.302 | 0.215 | 0.275 | 0.217 | 0.273 |
| Avg | 0.199 | 0.257 | 0.227 | 0.274 | 0.251 | 0.298 | 0.234 | 0.274 | 0.232 | 0.273 | 0.225 | 0.286 | 0.205 | 0.265 | 0.200 | 0.255 |
| ECL 96 | 0.133 | 0.233 | 0.140 | 0.233 | 0.162 | 0.275 | 0.155 | 0.259 | 0.156 | 0.257 | 0.137 | 0.236 | 0.136 | 0.231 | 0.131 | 0.228 |
| 192 | 0.149 | 0.250 | 0.153 | 0.245 | 0.177 | 0.291 | 0.169 | 0.270 | 0.168 | 0.269 | 0.160 | 0.253 | 0.151 | 0.249 | 0.150 | 0.247 |
| 336 | 0.163 | 0.265 | 0.170 | 0.262 | 0.195 | 0.309 | 0.183 | 0.286 | 0.183 | 0.288 | 0.169 | 0.266 | 0.166 | 0.264 | 0.165 | 0.265 |
| 720 | 0.199 | 0.293 | 0.208 | 0.295 | 0.215 | 0.326 | 0.210 | 0.312 | 0.215 | 0.319 | 0.205 | 0.300 | 0.198 | 0.291 | 0.200 | 0.297 |
| Avg | 0.161 | 0.260 | 0.168 | 0.259 | 0.187 | 0.300 | 0.179 | 0.282 | 0.181 | 0.283 | 0.168 | 0.264 | 0.163 | 0.259 | 0.162 | 0.259 |
| Traffic 96 | 0.350 | 0.247 | 0.403 | 0.277 | 0.499 | 0.318 | 0.466 | 0.294 | 0.466 | 0.291 | 0.391 | 0.286 | 0.365 | 0.262 | 0.367 | 0.265 |
| 192 | 0.374 | 0.263 | 0.416 | 0.280 | 0.500 | 0.315 | 0.480 | 0.302 | 0.478 | 0.300 | 0.421 | 0.297 | 0.396 | 0.277 | 0.400 | 0.281 |
| 336 | 0.389 | 0.268 | 0.424 | 0.285 | 0.517 | 0.321 | 0.490 | 0.307 | 0.487 | 0.305 | 0.432 | 0.302 | 0.405 | 0.279 | 0.402 | 0.276 |
| 720 | 0.435 | 0.293 | 0.456 | 0.303 | 0.549 | 0.332 | 0.521 | 0.319 | 0.516 | 0.317 | 0.464 | 0.314 | 0.440 | 0.302 | 0.446 | 0.306 |
| Avg | 0.387 | 0.268 | 0.425 | 0.286 | 0.516 | 0.322 | 0.489 | 0.306 | 0.487 | 0.303 | 0.427 | 0.300 | 0.402 | 0.280 | 0.404 | 0.282 |
| Total Avg | 0.290 | 0.319 | 0.302 | 0.324 | 0.375 | 0.381 | 0.320 | 0.336 | 0.320 | 0.335 | 0.315 | 0.337 | 0.295 | 0.322 | 0.295 | 0.321 |

Table 14: Full performance comparison between our DecompNet and other enhancement frameworks. We conduct experiments in long-term forecasting tasks and adopt RLinear [14] as the backbone model. The original backbone model equips with RevIN [11] by default. And we replace RevIN with other time series enhancement frameworks for comparisons. The input length is fixed as 384 and the prediction lengths are in $T \in \{96, 192, 336, 720\}$. *Avg* means the average results from all four prediction lengths. A lower MSE or MAE indicates a better performance. The best results are in **bold** and the second best are underlined. And *Total Avg* means the further average results from all eight datasets.

| | | Decomposition-based | | Prior-based | | | | Normalization-based | | | | | | Original Backbone Model | |
|---|---|---|---|---|---|---|---|---|---|---|---|---|---|---|---|
| Time Series Enhancement Frameworks | | +DecompNet (**Ours**) | | +CycleNet [15] | | +SpraseTSF [16] | | +FAN [43] | | +SAN [20] | | +Dish-TS [9] | | +RevIN [11] | |
| Metric | | MSE | MAE | MSE | MAE | MSE | MAE | MSE | MAE | MSE | MAE | MSE | MAE | MSE | MAE |
| ETTh1 | 96 | **0.369** | **0.392** | 0.374 | 0.396 | 0.402 | 0.417 | 0.395 | 0.419 | 0.381 | 0.398 | 0.419 | 0.433 | 0.372 | 0.397 |
| | 192 | **0.403** | **0.412** | 0.428 | 0.434 | 0.416 | 0.439 | 0.446 | 0.455 | 0.416 | 0.419 | 0.447 | 0.451 | 0.411 | 0.421 |
| | 336 | 0.427 | 0.427 | 0.448 | 0.448 | **0.422** | **0.420** | 0.490 | 0.485 | 0.432 | 0.432 | 0.473 | 0.473 | 0.425 | 0.429 |
| | 720 | 0.441 | 0.456 | 0.462 | 0.474 | **0.432** | **0.431** | 0.560 | 0.544 | 0.448 | 0.461 | 0.512 | 0.522 | 0.450 | 0.465 |
| | Avg | **0.410** | **0.422** | 0.428 | 0.438 | 0.418 | 0.427 | 0.473 | 0.476 | 0.419 | 0.428 | 0.463 | 0.470 | 0.415 | 0.428 |
| ETTh2 | 96 | **0.272** | **0.334** | 0.299 | 0.355 | 0.303 | 0.351 | 0.318 | 0.382 | 0.282 | 0.342 | 0.343 | 0.398 | 0.278 | 0.338 |
| | 192 | **0.337** | **0.376** | 0.351 | 0.392 | 0.379 | 0.397 | 0.398 | 0.434 | 0.343 | 0.383 | 0.449 | 0.462 | 0.356 | 0.393 |
| | 336 | **0.356** | **0.397** | 0.369 | 0.411 | 0.381 | 0.410 | 0.454 | 0.465 | 0.366 | 0.406 | 0.564 | 0.526 | 0.364 | 0.408 |
| | 720 | **0.386** | 0.426 | 0.405 | 0.439 | 0.390 | **0.424** | 0.623 | 0.572 | 0.394 | 0.435 | 0.748 | 0.619 | 0.419 | 0.447 |
| | Avg | **0.338** | **0.383** | 0.356 | 0.399 | 0.363 | 0.396 | 0.448 | 0.463 | 0.346 | 0.392 | 0.526 | 0.501 | 0.354 | 0.397 |
| ETTm1 | 96 | 0.303 | **0.344** | 0.312 | 0.354 | 0.311 | 0.353 | 0.301 | 0.350 | **0.297** | 0.347 | 0.312 | 0.354 | 0.316 | 0.355 |
| | 192 | 0.336 | **0.364** | 0.346 | 0.374 | 0.346 | 0.373 | 0.341 | 0.375 | **0.334** | 0.366 | 0.345 | 0.375 | 0.337 | 0.366 |
| | 336 | 0.370 | 0.384 | 0.378 | 0.392 | 0.377 | 0.391 | 0.376 | 0.397 | **0.367** | **0.383** | 0.379 | 0.395 | 0.375 | 0.389 |
| | 720 | 0.426 | **0.415** | 0.433 | 0.421 | 0.424 | 0.416 | 0.438 | 0.436 | **0.415** | 0.419 | 0.441 | 0.436 | 0.426 | **0.415** |
| | Avg | 0.359 | **0.377** | 0.367 | 0.385 | 0.365 | 0.383 | 0.364 | 0.390 | **0.353** | 0.379 | 0.369 | 0.390 | 0.364 | 0.381 |
| ETTm2 | 96 | **0.164** | **0.253** | 0.175 | 0.260 | 0.180 | 0.268 | 0.177 | 0.266 | 0.177 | 0.272 | 0.178 | 0.275 | **0.164** | **0.253** |
| | 192 | **0.218** | **0.289** | 0.223 | 0.296 | 0.231 | 0.301 | 0.276 | 0.344 | 0.224 | 0.301 | 0.250 | 0.331 | 0.227 | 0.297 |
| | 336 | **0.272** | **0.324** | 0.277 | 0.330 | 0.288 | 0.335 | 0.324 | 0.371 | 0.276 | 0.337 | 0.328 | 0.385 | 0.275 | 0.329 |
| | 720 | 0.365 | **0.382** | 0.372 | 0.391 | 0.371 | 0.386 | 0.420 | 0.434 | **0.362** | 0.391 | 0.450 | 0.456 | 0.368 | 0.386 |
| | Avg | **0.255** | **0.312** | 0.262 | 0.319 | 0.268 | 0.323 | 0.299 | 0.354 | 0.260 | 0.325 | 0.302 | 0.362 | 0.259 | 0.316 |
| Weather | 96 | 0.162 | 0.217 | 0.166 | 0.220 | 0.174 | 0.225 | 0.156 | **0.209** | **0.152** | 0.211 | 0.176 | 0.235 | 0.174 | 0.227 |
| | 192 | 0.206 | 0.256 | 0.213 | 0.260 | 0.216 | 0.260 | 0.202 | 0.259 | **0.196** | **0.253** | 0.217 | 0.273 | 0.216 | 0.261 |
| | 336 | 0.253 | **0.290** | 0.259 | 0.292 | 0.265 | 0.297 | 0.255 | 0.297 | **0.246** | 0.294 | 0.262 | 0.311 | 0.263 | 0.294 |
| | 720 | **0.320** | **0.336** | 0.326 | 0.339 | 0.332 | 0.343 | **0.320** | 0.352 | 0.321 | 0.346 | 0.325 | 0.362 | 0.329 | 0.340 |
| | Avg | 0.235 | **0.275** | 0.241 | 0.278 | 0.247 | 0.281 | 0.233 | 0.279 | **0.229** | 0.276 | 0.245 | 0.295 | 0.246 | 0.281 |
| Solar-Energy | 96 | 0.201 | 0.257 | 0.212 | 0.267 | 0.227 | 0.267 | **0.197** | **0.248** | 0.198 | 0.253 | 0.224 | 0.294 | 0.223 | 0.297 |
| | 192 | 0.231 | 0.274 | 0.240 | 0.293 | 0.259 | 0.287 | **0.229** | **0.270** | 0.234 | 0.273 | 0.270 | 0.312 | 0.251 | 0.312 |
| | 336 | **0.238** | 0.282 | 0.255 | 0.302 | 0.276 | 0.297 | **0.240** | 0.280 | 0.243 | 0.287 | 0.264 | 0.323 | 0.265 | 0.338 |
| | 720 | **0.238** | 0.282 | 0.257 | 0.301 | 0.279 | 0.293 | **0.238** | **0.277** | 0.241 | 0.281 | 0.266 | 0.326 | 0.284 | 0.345 |
| | Avg | 0.227 | 0.274 | 0.241 | 0.291 | 0.260 | 0.286 | **0.226** | **0.269** | 0.229 | 0.274 | 0.256 | 0.314 | 0.256 | 0.323 |
| ECL | 96 | 0.140 | **0.233** | **0.139** | **0.233** | 0.147 | 0.241 | 0.140 | 0.239 | 0.140 | 0.234 | 0.160 | 0.266 | 0.151 | 0.251 |
| | 192 | **0.153** | **0.245** | 0.154 | 0.247 | 0.157 | 0.250 | 0.156 | 0.254 | 0.155 | 0.247 | 0.172 | 0.276 | 0.163 | 0.262 |
| | 336 | **0.170** | **0.262** | 0.170 | 0.264 | 0.174 | 0.268 | 0.171 | 0.271 | 0.173 | 0.267 | 0.187 | 0.291 | 0.179 | 0.277 |
| | 720 | 0.208 | **0.295** | 0.208 | 0.297 | 0.211 | 0.298 | 0.207 | 0.305 | **0.202** | 0.296 | 0.221 | 0.323 | 0.218 | 0.308 |
| | Avg | **0.168** | **0.259** | **0.168** | 0.260 | 0.172 | 0.264 | 0.169 | 0.267 | **0.168** | 0.261 | 0.185 | 0.289 | 0.178 | 0.275 |
| Traffic | 96 | **0.403** | **0.277** | 0.413 | 0.287 | 0.425 | 0.283 | 0.416 | 0.290 | 0.414 | 0.288 | 0.438 | 0.308 | 0.433 | 0.305 |
| | 192 | **0.416** | **0.280** | 0.421 | 0.290 | 0.436 | 0.292 | 0.430 | 0.304 | 0.427 | 0.298 | 0.450 | 0.311 | 0.441 | 0.307 |
| | 336 | **0.424** | **0.285** | 0.432 | 0.297 | 0.465 | 0.310 | 0.443 | 0.311 | 0.441 | 0.305 | 0.462 | 0.317 | 0.453 | 0.313 |
| | 720 | **0.456** | **0.303** | 0.459 | 0.306 | 0.497 | 0.330 | 0.483 | 0.336 | 0.476 | 0.320 | 0.489 | 0.338 | 0.483 | 0.331 |
| | Avg | **0.425** | **0.286** | 0.431 | 0.295 | 0.456 | 0.304 | 0.443 | 0.310 | 0.440 | 0.303 | 0.460 | 0.319 | 0.453 | 0.314 |
| 1st Count | | **23** | **29** | 2 | 1 | 2 | 3 | 6 | 5 | 12 | 2 | 0 | 0 | 1 | 2 |
| Total Avg | | **0.302** | **0.324** | 0.312 | 0.333 | 0.319 | 0.333 | 0.332 | 0.351 | 0.306 | 0.330 | 0.351 | 0.368 | 0.316 | 0.339 |

Table 15: Full performance comparison between our DecompNet and other enhancement frameworks. We conduct experiments in long-term forecasting tasks and adopt PatchTST [30] as the backbone model. The original backbone model equips with RevIN [11] by default. And we replace RevIN with other time series enhancement frameworks for comparisons. The input length is fixed as 384 and the prediction lengths are in $T \in \{96, 192, 336, 720\}$. *Avg* means the average results from all four prediction lengths. A lower MSE or MAE indicates a better performance. The best results are in **bold** and the second best are underlined. And *Total Avg* means the further average results from all eight datasets.

| | | Decomposition-based +DecompNet (Ours) | | Prior-based +CycleNet [15] | | +SpraseTSF [16] | | Normalization-based +FAN [43] | | +SAN [20] | | +Dish-TS [9] | | Original Backbone Model +RevIN [11] | |
|---|---|---|---|---|---|---|---|---|---|---|---|---|---|---|---|
| | Metric | MSE | MAE | MSE | MAE | MSE | MAE | MSE | MAE | MSE | MAE | MSE | MAE | MSE | MAE |
| ETTh1 | 96 | **0.361** | **0.390** | 0.382 | 0.407 | 0.368 | 0.393 | 0.418 | 0.430 | 0.410 | 0.418 | 0.405 | 0.422 | 0.376 | 0.401 |
| | 192 | **0.391** | **0.410** | 0.419 | 0.428 | 0.398 | **0.410** | 0.447 | 0.450 | 0.473 | 0.452 | 0.447 | 0.456 | 0.410 | 0.421 |
| | 336 | **0.418** | **0.428** | 0.453 | 0.456 | 0.434 | 0.437 | 0.490 | 0.476 | 0.568 | 0.502 | 0.481 | 0.481 | 0.434 | 0.439 |
| | 720 | **0.442** | **0.455** | 0.468 | 0.482 | 0.487 | 0.486 | 0.557 | 0.530 | 0.578 | 0.530 | 0.579 | 0.535 | 0.451 | 0.471 |
| | Avg | **0.403** | **0.421** | 0.431 | 0.443 | 0.422 | 0.432 | 0.478 | 0.472 | 0.507 | 0.476 | 0.478 | 0.474 | 0.418 | 0.433 |
| ETTh2 | 96 | 0.277 | **0.337** | 0.277 | 0.348 | 0.311 | 0.357 | 0.300 | 0.365 | 0.319 | 0.366 | 0.389 | 0.426 | **0.275** | **0.337** |
| | 192 | **0.351** | 0.386 | 0.355 | **0.383** | 0.374 | 0.405 | 0.387 | 0.423 | 0.396 | 0.427 | 0.496 | 0.468 | 0.354 | 0.388 |
| | 336 | 0.371 | **0.399** | 0.381 | 0.411 | 0.375 | 0.409 | 0.453 | 0.507 | 0.414 | 0.436 | 0.507 | 0.496 | **0.368** | 0.403 |
| | 720 | **0.386** | **0.425** | 0.396 | 0.431 | 0.435 | 0.471 | 0.578 | 0.529 | 0.451 | 0.475 | 0.706 | 0.576 | 0.398 | 0.435 |
| | Avg | **0.346** | **0.387** | 0.352 | 0.393 | 0.374 | 0.411 | 0.430 | 0.456 | 0.395 | 0.426 | 0.525 | 0.492 | 0.349 | 0.391 |
| ETTm1 | 96 | **0.283** | **0.337** | 0.289 | 0.342 | 0.309 | 0.363 | 0.303 | 0.351 | 0.289 | 0.342 | 0.305 | 0.360 | 0.292 | 0.344 |
| | 192 | **0.328** | **0.368** | 0.342 | 0.371 | 0.340 | 0.379 | 0.341 | 0.374 | 0.334 | 0.370 | 0.351 | 0.383 | 0.340 | 0.374 |
| | 336 | **0.365** | 0.392 | **0.365** | **0.389** | 0.395 | 0.411 | 0.385 | 0.400 | 0.368 | 0.395 | 0.385 | 0.410 | 0.367 | 0.394 |
| | 720 | **0.405** | **0.421** | 0.428 | 0.427 | 0.429 | 0.434 | 0.440 | 0.436 | 0.424 | 0.433 | 0.431 | 0.437 | 0.422 | 0.427 |
| | Avg | **0.345** | **0.380** | 0.356 | 0.382 | 0.368 | 0.397 | 0.367 | 0.390 | 0.354 | 0.385 | 0.368 | 0.398 | 0.355 | 0.385 |
| ETTm2 | 96 | **0.166** | **0.254** | 0.177 | 0.265 | 0.187 | 0.280 | 0.171 | 0.266 | 0.170 | 0.265 | 0.172 | 0.264 | **0.166** | 0.255 |
| | 192 | **0.222** | **0.293** | 0.234 | 0.307 | 0.242 | 0.316 | 0.238 | 0.316 | 0.239 | 0.314 | 0.236 | 0.312 | 0.228 | 0.299 |
| | 336 | **0.272** | **0.325** | 0.279 | 0.332 | 0.291 | 0.341 | 0.305 | 0.365 | 0.377 | 0.390 | 0.304 | 0.352 | 0.280 | 0.331 |
| | 720 | **0.366** | **0.385** | 0.367 | 0.395 | 0.367 | 0.387 | 0.414 | 0.435 | 0.550 | 0.475 | 0.420 | 0.437 | 0.373 | 0.389 |
| | Avg | **0.257** | **0.314** | 0.264 | 0.325 | 0.272 | 0.331 | 0.282 | 0.346 | 0.334 | 0.361 | 0.283 | 0.341 | 0.262 | 0.319 |
| Weather | 96 | 0.146 | **0.194** | **0.145** | 0.199 | 0.168 | 0.226 | 0.154 | 0.211 | 0.150 | 0.208 | 0.150 | 0.207 | 0.150 | 0.201 |
| | 192 | 0.190 | **0.238** | **0.188** | 0.240 | 0.208 | 0.261 | 0.199 | 0.254 | 0.194 | 0.251 | 0.195 | 0.256 | 0.194 | 0.244 |
| | 336 | **0.243** | **0.279** | 0.247 | 0.286 | 0.258 | 0.299 | 0.249 | 0.295 | **0.243** | 0.293 | 0.248 | 0.304 | **0.243** | **0.279** |
| | 720 | 0.317 | **0.332** | **0.316** | 0.333 | 0.331 | 0.353 | 0.317 | 0.346 | 0.322 | 0.344 | 0.324 | 0.370 | 0.319 | 0.333 |
| | Avg | **0.224** | **0.261** | **0.224** | 0.265 | 0.241 | 0.285 | 0.230 | 0.277 | 0.227 | 0.274 | 0.229 | 0.284 | 0.227 | 0.264 |
| Solar-Energy | 96 | **0.172** | **0.229** | 0.198 | 0.266 | 0.196 | 0.273 | 0.187 | 0.243 | 0.183 | 0.240 | 0.180 | 0.241 | 0.197 | 0.291 |
| | 192 | **0.197** | **0.254** | 0.271 | 0.320 | 0.216 | 0.286 | 0.201 | 0.273 | 0.207 | 0.265 | 0.202 | 0.256 | 0.214 | 0.319 |
| | 336 | **0.211** | **0.270** | 0.243 | 0.306 | 0.232 | 0.298 | 0.223 | 0.288 | 0.223 | 0.278 | 0.218 | 0.276 | 0.235 | 0.328 |
| | 720 | **0.216** | **0.274** | 0.240 | 0.293 | 0.234 | 0.296 | 0.225 | 0.290 | 0.232 | 0.285 | 0.219 | 0.276 | 0.225 | 0.329 |
| | Avg | **0.199** | **0.257** | 0.238 | 0.296 | 0.220 | 0.288 | 0.209 | 0.274 | 0.211 | 0.267 | 0.205 | 0.262 | 0.218 | 0.317 |
| ECL | 96 | **0.133** | 0.233 | 0.139 | 0.239 | 0.138 | 0.233 | 0.137 | 0.235 | **0.133** | **0.231** | 0.136 | 0.234 | **0.133** | 0.232 |
| | 192 | 0.149 | 0.250 | 0.157 | 0.256 | 0.161 | 0.264 | 0.152 | 0.250 | **0.148** | 0.248 | 0.151 | **0.246** | **0.148** | **0.246** |
| | 336 | **0.163** | 0.265 | 0.171 | 0.267 | 0.171 | 0.272 | 0.168 | 0.268 | 0.165 | 0.267 | 0.166 | 0.266 | 0.169 | **0.265** |
| | 720 | 0.199 | 0.293 | 0.210 | 0.300 | 0.210 | 0.309 | 0.202 | 0.300 | **0.197** | 0.294 | 0.199 | 0.295 | 0.202 | 0.295 |
| | Avg | **0.161** | 0.260 | 0.169 | 0.266 | 0.170 | 0.270 | 0.165 | 0.263 | **0.161** | **0.260** | 0.163 | **0.260** | 0.163 | **0.260** |
| Traffic | 96 | **0.350** | **0.247** | 0.411 | 0.295 | 0.396 | 0.274 | 0.405 | 0.284 | 0.408 | 0.282 | 0.394 | 0.277 | 0.385 | 0.277 |
| | 192 | **0.374** | **0.263** | 0.422 | 0.301 | 0.410 | 0.281 | 0.423 | 0.294 | 0.417 | 0.287 | 0.408 | 0.282 | 0.401 | 0.283 |
| | 336 | **0.389** | **0.268** | 0.432 | 0.309 | 0.421 | 0.287 | 0.437 | 0.302 | 0.436 | 0.298 | 0.423 | 0.291 | 0.410 | 0.287 |
| | 720 | **0.435** | **0.293** | 0.458 | 0.319 | 0.442 | 0.303 | 0.476 | 0.321 | 0.464 | 0.313 | 0.452 | 0.309 | 0.443 | 0.304 |
| | Avg | **0.387** | **0.268** | 0.431 | 0.306 | 0.417 | 0.286 | 0.435 | 0.300 | 0.431 | 0.295 | 0.419 | 0.290 | 0.410 | 0.288 |
| 1st Count | | **33** | **36** | 5 | 2 | 0 | 1 | 0 | 0 | 5 | 2 | 0 | 2 | 6 | 5 |
| Total Avg | | **0.290** | **0.319** | 0.308 | 0.335 | 0.311 | 0.338 | 0.325 | 0.347 | 0.328 | 0.343 | 0.334 | 0.350 | 0.300 | 0.332 |

Backbone Model: PatchTST

# S Showcases

## S.1 Intuitive Comparison between Decoupled Training Strategy and Joint Training Strategy

We provide the intuitive comparison between our decoupled training strategy and previous joint training strategy. We visualize the performance of two expert models in seasonal prediction task and trend prediction task, respectively. The results are in Figure 10 and 11. When trained with our decoupled training strategy, the trend expert model can perform much better in trend prediction task. And the seasonal expert model can also better handle the details in seasonal prediction task.

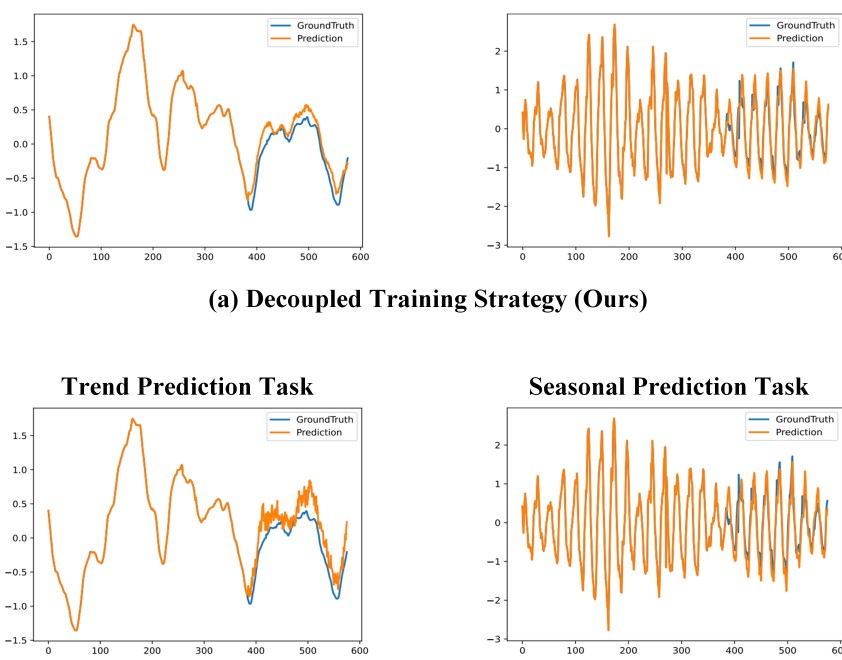

Figure 10: Intuitive comparison between our decoupled training strategy and previous joint training strategy. We visualize the performance of two expert models in seasonal prediction task and trend prediction task, respectively. We conduct experiment in ECL dataset under input-384-predict-192 setting, using RLinear [14] as the backbone model. The blue lines stand for the ground truth and the orange lines stand for predicted values.

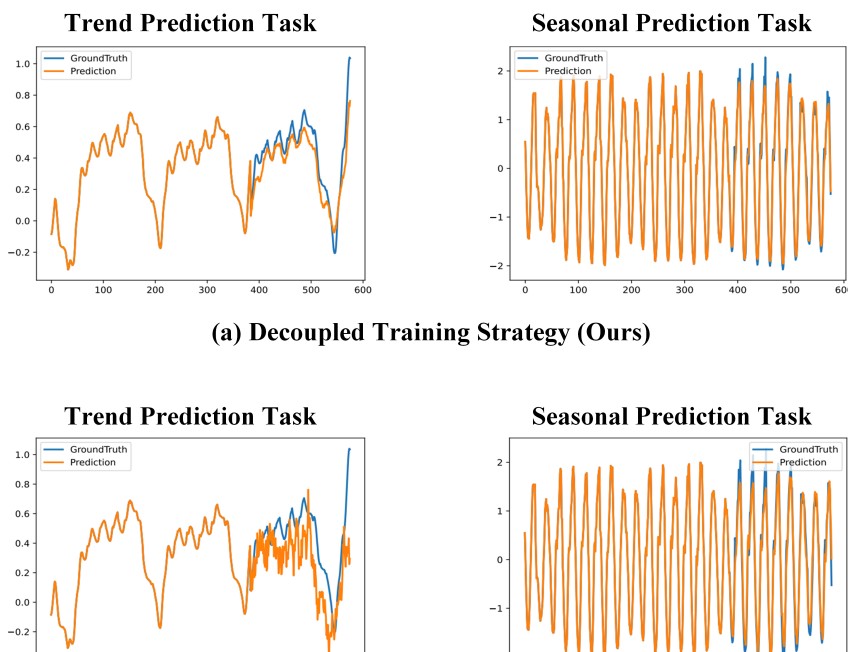

Figure 11: Intuitive comparison between our decoupled training strategy and previous joint training strategy. We visualize the performance of two expert models in seasonal prediction task and trend prediction task, respectively. We conduct experiment in Traffic dataset under input-384-predict-192 setting, using RLinear [14] as the backbone model. The blue lines stand for the ground truth and the orange lines stand for predicted values.

## S.2 Intuitive Comparison among Different Enhancement Frameworks

To provide an intuitive comparison among different enhancement frameworks, we provide showcases to the long-term forecasting tasks. The results are in Figure 12. Among the various enhancement frameworks, our DecompNet predicts the most precise future series variations and exhibits superior performance.

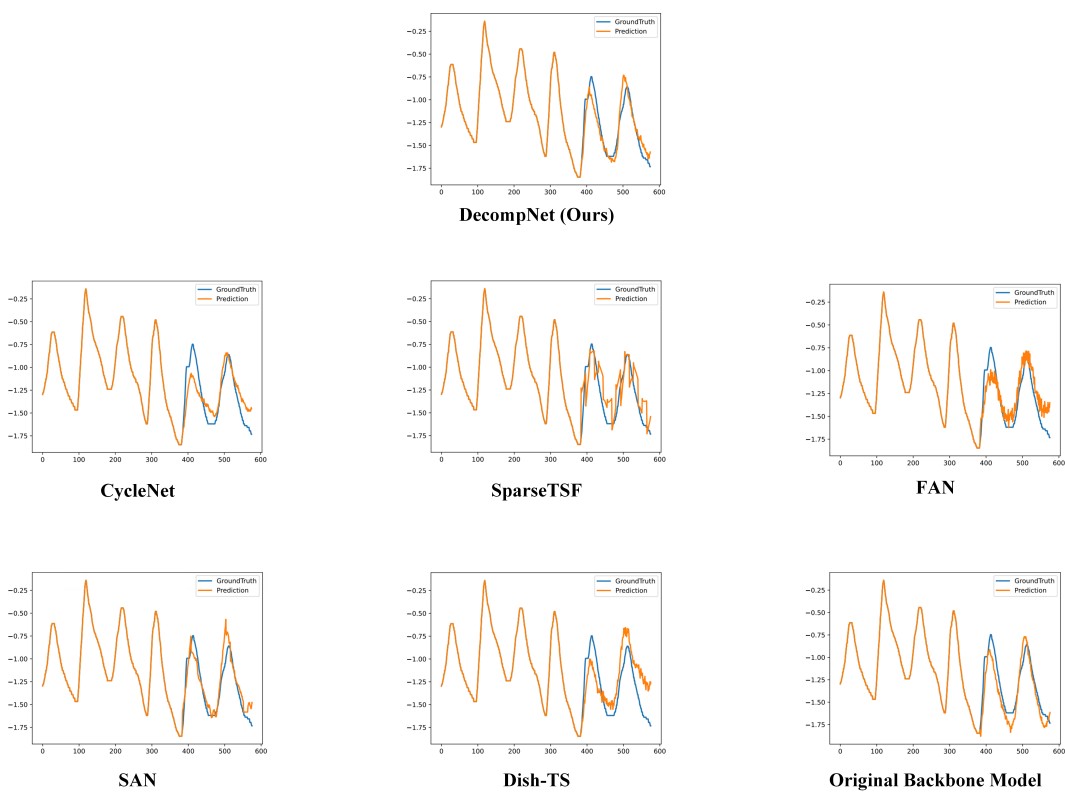

Figure 12: Visualization of ETTm2 predictions by different enhancement frameworks under the input-384-predict-192 setting. We adopt PatchTST [30] as the backbone model for this visualization. The blue lines stand for the ground truth and the orange lines stand for predicted values.

