# OpenReview forum: "DecompNet: Enhancing Time Series Forecasting Models with Implicit Decomposition"
_NeurIPS.cc/2025/Conference — NeurIPS 2025 poster_

### Official Review · Reviewer_CJUx · 2025-06-21

**Clarity:** 2
**Significance:** 3
**Originality:** 2
**Rating:** 4
**Confidence:** 4

**Summary:**

This paper proposes a novel time series forecasting method centered on trend-seasonality re-parametrization, introducing a new concept referred to as "implicit decomposition." The proposed approach demonstrates consistent improvements across various baselines.

**Questions:**

1. Is the lambda fusion parameter learned end-to-end during model training as a network weight, or is it tuned as a hyperparameter? If the latter, what is the range of values considered?

**Ethical Concerns:**

["NO or VERY MINOR ethics concerns only"]

**Final Justification:**

I recommend increasing my score to a 4 (borderline accept) based on the authors response to my comments.

I am happy that the authors have added error bars for experiments across random seeds. However, I would like to see their statement, "The average performance improvement is roughly an order of magnitude larger than the random fluctuation from multiple runs", supported with statistical tests and to valid the use of the term “significant” when referring to performance improvements. Additionally, I would like to see metrics like FLOPs and wall-clock time to convey computational cost and training time, respectively in addition to current metric (training epochs).

**Limitations:**

Yes

**Quality:**

3

**Strengths And Weaknesses:**

Strengths:
- The proposed model shows strong performance gains across a wide range of datasets and baselines, as reflected in the presented tables and figures. Table 7 in the appendix nicely summarizes the model's improvements.

- The figures are clear and convey the findings well. Figure 4 provides an effective visual comparison of benchmarks for multiple factors, including loss, inference time, and varying input lengths.

- Forecast examples included in Appendix N.1 and N.2 help showcase the performance improvement.

- An ablation study on each component of the method is provided to showcase the combined improvement of the proposed techniques.

Weaknesses:
1. The reference to trend-seasonality re-parameterization as “implicit decomposition” is potentially misleading. In practice, the model performs explicit decomposition during training using a moving average filter. Implicit decomposition is assumed to occur during inference through the re-parameterized model; however, there is no clear evidence or analysis confirming that this actually happens. This behavior seems to be inferred from the empirical forecasting results.

2. Many of the claims throughout the paper use subjective or vague language where quantitative evidence is not explicitly provided to verify them. The reader must try to parse the claims from the provided figures and tables. I recommend that the authors revise these statements and explicitly tie each claim to specific metrics results. Some examples include:
   - Line 241: “by a large margin”
   - Lines 261–262: “DecompNet also significantly surpasses...”
   - Line 366: “enjoying better performance and having totally the same inference costs”

3. The authors report the standard deviation of DecompNet performance under five runs with different random seeds in Table 6 in the Appendix using PatchTST, but do not do the same for the other benchmarks. The average results along with standard deviations are not included in the main tables. With results across random seeds for both the baseline and the proposed method (for each model), one can also consider adding statistical significance tests to confirm whether improvements are meaningful or due to chance.

4. I understand the focus on inference cost for deployment practicality. However, including training cost information is still valuable to ensure the method incurs a reasonable expense. Training cost details are not explicitly conveyed in the main paper, though metrics like the number of epochs are used in the appendix. Including a couple of sentences on this in the main paper with explicit metrics would be helpful. Also, metrics like FLOPs and wall-clock time are generally used to convey computational cost and training time, respectively.

5. For many key details, the reader is directed to refer to the appendix. For example, explanations of benchmarks like MICN and Leddam, details on the average window size for the proposed method, and references to model training details in Appendix C. Including a summary sentence on these topics before redirecting readers to the appendix would be helpful.

Other:
1. The trend-seasonality re-parameterization, which appears to be one of the main contributions of the paper, is not introduced until page 3. This core idea should be presented earlier, as it seems to be the main contribution and may be both clearer and more familiar to readers than the concept of implicit decomposition. The notion of implicit decomposition could then be introduced in relation to this concept.

2. It would strengthen the paper to compare the proposed model and its variants against other prominent decomposition-based models such as DLinear, NHiTs, and N-BEATS. This comparison would better contextualize the method’s performance and practical relevance.

3. Section G is helpful and addresses the effect of moving window size. This could be referenced in the main text to assure readers that this aspect has been considered and studied.

4. Including some forecast examples in the main paper would be very helpful to demonstrate the performance improvement of the proposed approach.

---

> ### Author Rebuttal · Authors · 2025-07-31
>
> Many thanks to Reviewer CJUx for the detailed comments.
>
> > W1-1: About the relationship between implicit decomposition and Seasonal-Trend Re-parameterization
>
> We'd like to clarify that we do not refer Seasonal-Trend Re-parameterization as implicit decomposition.
> + Implicit decomposition is **a brand new idea** we pioneered in this paper, which aims to improve a model's performance by decomposition-related knowledge during inference but without actually decomposing the input time series.
> + Seasonal-Trend Re-parameterization (along with decoupled training strategy) are **two key designs we proposed to achieve the goal of implicit decomposition**.
>
> > W1-2: About the naming of implicit decomposition
>
> To eliminate potential misleading, we provide the reasons for the naming of implicit decomposition as follows:
> + Previous decomposition-based methods hold a belief that: if we want to utilize the decomposition-related knowledge to improve a model's performance, we **must explicitly and actually** decompose the input time series during inference.
> + While in this paper, we find it possible to utilize the decomposition-related knowledge to improve a model's performance **without actually** decomposing the input time series during inference.
> + This is a brand-new finding that challenges the commonly held belief in previous decomposition-based methods. To distinguish it from the previous methods, we name it as **implicit decomposition**.
>
> > W1-3: The model performs explicit decomposition during training
> + Thanks for highlighting this. The concept of explicit decomposition and implicit decomposition in this paper are used to distinguish whether a method needs to actually decompose the input time series **during inference**.
> + Both previous explicit decomposition method and our implicit decomposition method need to explicitly decompose the data during training. **But the main difference lies in the inference stage**, our implicit decomposition can avoid explicitly decomposing the input time series during inference, achieving inference-cost-free performance enhancement.
>
> > W1-4: Evidence and analysis
> + Experimental evidence: Implicit decomposition aims to improve a model's performance by decomposition-related knowledge during inference but without actually decomposing the input time series. And as shown in our main results in **Table 1 in Section 4.1**, this kind of performance improvement really happens, which can serve as the experimental evidence for implicit decomposition.
> + Theoretical analysis: Since rigorous theoretical analysis for a model-fusion-based method like implicit decomposition remains an open challenge beyond current methodological capabilities—even leading works in this nascent field (e.g., [1-4]) rely primarily on empirical validation, we provide **comprehensive experimental evidence** to demonstrate our efficacy and will leave the theoretical explanation in future work.
>
> [1] Qwen3 Embedding: Advancing Text Embedding and Reranking Through Foundation Models
>
> [2] Language Models are Super Mario: Absorbing Abilities from Homologous Models as a Free Lunch
>
> [3] Model Merging in Pre-training of Large Language Models
>
> [4] Multimodal Pathway: Improve Transformers with Irrelevant Data from Other Modalities
>
> > Other1-1: Seasonal-Trend Re-parameterization should be presented earlier.
>
> Thanks for the recognition of our Seasonal-Trend Re-parameterization. Following your valuable suggestion, we will rephrase the sentence in **line 66** as "we propose a powerful decomposition-based enhancement framework called DecompNet, which features two key designs: Seasonal-Trend Re-parameterization and decoupled training strategy" in our final version.
>
> > Other1-2: About contributions.
>
> Beyond the proposal a novel technique like Seasonal-Trend Re-parameterization, **the introduction of the brand-new idea of implicit decomposition is also an important contribution** of our paper, which can provide some **novel insights** to time series community:
> + Based on the idea of implicit decomposition, we are **the first** to propose a decomposition-based enhancement framework that surpasses the well-recognized normalization-based frameworks. And we are **the first** to propose a inference-cost-free enhancement framework. **These unprecedented findings can encourage further exploration on more types and more efficient enhancement frameworks**. (Quantitative evidence is provided in **Section 4.1 and 4.2**.)
> + Implicit decomposition re-examines and challenges the commonly held belief in previous decomposition-based methods, which can **bring some fresh perspectives** and provide a better solution to the classic research topic of **time series decomposition**. (Quantitative evidence is provided in **Section 4.3**.)
>
> > Other1-3: Clarification on why we introduce implicit decomposition at first.
>
> Thanks for your great suggestions about the paper writing. While we do improve the other parts, we hope we can maintain to introduce implicit decomposition at first for following reasons:
> + Implicit decomposition represents the core theme of this paper—improving a model's performance by decomposition-related knowledge but without actually decomposing the input time series during inference. Therefore, introducing implicit decomposition at first can help readers quickly grasp the research objectives and the key contributions of this paper.
> + The idea of implicit decomposition can provide novel insights to time series community. Thus, we introduce implicit decomposition and share these insights with readers at first, which can highlight our contributions and novelty.
> + From a technical perspective, implicit decomposition represents the overall design idea, while Seasonal-Trend Re-parameterization is a specific technical detail. In our humble opinion, introducing the overall design idea first, followed by the specific technical details, will help to better clarify our method.
>
> > W2: About the claims and quantitative evidence
> + We'd like to clarify that our claims are not using subjective or vague language. All of our claims can be verified by the experimental results in our figures and tables.
> + Following your valuable suggestion, we will revise these statements in our final version and explicitly tie each claim to specific metrics results as follows:
>     + Line 241: DecompNet consistently enhances the performance of all backbone models by a large margin. For instance, on Solar, ETTh1, and Traffic datasets, the average MSE performance improvements are rather significant: 8.0%, 6.4% and 6.1% respectively.
>     + Line 261–262: DecompNet also significantly surpasses the latest prior-based methods. For instance, when using PatchTST as the backbone model, DecompNet achieves the best performance with an average MSE of 0.290, obtaining an average of 6.3% MSE reduction than the latest prior-based methods in all datasets.
>     + Line 366: enjoying better performance and having totally the same inference costs (e.g., in traffic dataset, it can boost the performance from 0.410 to 0.387 under the same inference time 322ms.)
>
> > W3: Main result (Table 1) with full error bar
>
> We report the **Table 1** with full error bar as follow. The average performance improvement is **roughly an order of magnitude larger** than the random fluctuation from multiple runs (0.0x vs 0.00x). Thus, the improvement in our main result is meaningful and not due to chance.
>
> |MSE/MAE|PatchTST|PatchTST+Ours|iTransformer|iTransformer+Ours|RLinear|RLinear+Ours|RMLP|RMLP+Ours|ModernTCN|ModernTCN+Ours|
> |:---:|:---:|:---:|:---:|:---:|:---:|:---:|:---:|:---:|:---:|:---:|
> |ETTh1|0.418±0.002/0.433±0.001|0.403±0.002/0.421±0.002|0.461±0.001/0.464±0.001|0.417±0.001/0.435±0.001|0.415±0.001/0.428±0.001|0.410±0.001/0.422±0.001|0.482±0.002/0.469±0.002|0.423±0.002/0.430±0.001|0.403±0.002/0.419±0.002|0.387±0.002/0.411±0.001|
> |ETTh2|0.349±0.001/0.391±0.001|0.346±0.002/0.387±0.002|0.377±0.001/0.412±0.001|0.344±0.002/0.389±0.002|0.354±0.001/0.397±0.001|0.338±0.001/0.383±0.001|0.364±0.002/0.401±0.001|0.347±0.002/0.390±0.002|0.328±0.002/0.382±0.001|0.312±0.002/0.373±0.001|
> |ETTm1|0.355±0.002/0.385±0.002|0.345±0.002/0.380±0.002|0.372±0.001/0.399±0.001|0.358±0.001/0.385±0.001|0.364±0.001/0.381±0.001|0.359±0.001/0.377±0.000|0.371±0.001/0.394±0.001|0.355±0.001/0.383±0.001|0.365±0.002/0.386±0.001|0.354±0.001/0.379±0.001|
> |ETTm2|0.262±0.001/0.319±0.001|0.257±0.001/0.314±0.001|0.274±0.001/0.332±0.001|0.262±0.001/0.321±0.000|0.259±0.000/0.316±0.001|0.255±0.001/0.312±0.001|0.271±0.002/0.326±0.001|0.259±0.001/0.318±0.001|0.261±0.002/0.320±0.001|0.254±0.001/0.314±0.001|
> |Weather|0.227±0.000/0.264±0.000|0.224±0.001/0.261±0.001|0.235±0.001/0.272±0.001|0.229±0.001/0.268±0.001|0.246±0.000/0.281±0.001|0.235±0.000/0.275±0.000|0.244±0.001/0.279±0.000|0.239±0.000/0.275±0.000|0.233±0.001/0.270±0.001|0.231±0.000/0.266±0.000|
> |Solar|0.218±0.001/0.317±0.001|0.199±0.001/0.257±0.001|0.212±0.002/0.272±0.001|0.204±0.002/0.259±0.001|0.256±0.002/0.323±0.002|0.227±0.001/0.274±0.001|0.231±0.003/0.294±0.003|0.214±0.002/0.249±0.002|0.218±0.001/0.306±0.001|0.200±0.001/0.258±0.001|
> |ECL|0.163±0.001/0.260±0.001|0.161±0.001/0.260±0.001|0.172±0.001/0.271±0.001|0.168±0.001/0.264±0.001|0.178±0.001/0.275±0.001|0.168±0.001/0.259±0.001|0.172±0.001/0.265±0.001|0.165±0.000/0.258±0.000|0.168±0.002/0.264±0.001|0.159±0.001/0.258±0.001|
> |Traffic|0.410±0.002/0.288±0.001|0.387±0.002/0.268±0.001|0.428±0.002/0.320±0.001|0.395±0.002/0.280±0.002|0.453±0.001/0.314±0.002|0.425±0.001/0.286±0.001|0.436±0.001/0.308±0.001|0.413±0.001/0.285±0.001|0.417±0.003/0.293±0.002|0.394±0.002/0.275±0.001|
> |Average|0.300±0.001/0.332±0.001|0.290±0.002/0.319±0.001|0.316±0.001/0.343±0.001|0.297±0.001/0.325±0.001|0.316±0.001/0.339±0.001|0.302±0.001/0.324±0.001|0.321±0.002/0.342±0.001|0.302±0.001/0.324±0.001|0.299±0.002/0.330±0.001|0.286±0.001/0.317±0.001|
>
> > Due to space limitation, the responses to W4, W5, Other2, Other3, Other4, Q1 are placed **at the bottom of the Response to Reviewer 5VM5**

---

> ### Author Response · Authors · 2025-08-01
> **A copy of the responses to W4, W5, Other2, Other3, Other4, Q1 (which are originally provided at the bottom of the Response to Reviewer 5VM5)**
>
> Dear Reviewer CJUx,
>
> Due to the space limitation, we have originally placed our responses to W4, W5, Other2, Other3, Other4, Q1 **at the bottom of the Response to Reviewer 5VM5** during the rebuttal period.
>
> To facilitate your review of our responses, **we provide a copy of these responses here**. We hope it can better help to address your concern.
>
> > W4 of Reviewer CJUx: Convey the training cost details (Appendix E.1) in the main text.
>
> Thanks for your suggestion. We will move **Appendix E.1** to the main text in our final version and highlight our nice training efficiency.
>
> > W5 of Reviewer CJUx: Include a summary sentence before redirecting readers to the appendix.
> + For the two baselines MICN and Leddam, we will rephrase **line 309-311** as: "We compare our DecompNet with some latest and representative explicit decomposition methods, i.e., MICN  and Leddam. Please refer to **Appendix F.3** for a detailed introduction of their model structures and data processing pipelines".
> + For moving average window size, we will add a new sentence in **line 234**: "We set the moving average window size as 25 for time series decomposition during the decoupled pretraining stage. And we provide the parameter sensitivity study in **Appendix G**".
> + For training details, we will add a new sentence in **line 234**: "Our method is trained with L2 loss, using ADAM optimizer with an initial learning rate in {$10^{-3}, 5\times 10^{-4}, 10^{-4}$}. The default training process is 30 epochs with early stopping. More implementation details are in **Appendix C**".
>
> > Other2 of Reviewer CJUx: Comparison with other prominent decomposition-based models like DLinear, NHiTs and N-BEATS
> + Following your valuable suggestion, we conduct these comparisons. As shown in following table, our method surpasses these prominent decomposition-based models, achieving an average MSE reduction of 11.1% compared to DLinear, 16.7% compared to NHiTs and 18.6% compared to N-BEATS, thus verifying our performance superiority.
> + Please also see **Section 4.3** for our comparison with two more competitive decomposition-based models (i.e., MICN and Leddam).
>
> |MSE/MAE|DecompNet+PatchTST|DecompNet+RLinear|DLinear|NHiTs|N-BEATS|
> |:---:|:---:|:---:|:---:|:---:|:---:|
> |ETTh1|0.403/0.421|0.410/0.422|0.428/0.444|0.486/0.477|0.500/0.486|
> |ETTm1|0.345/0.380|0.359/0.377|0.365/0.385|0.372/0.397|0.385/0.404|
> |Solar|0.199/0.257|0.227/0.274|0.255/0.338|0.265/0.348|0.266/0.345|
> |Traffic|0.387/0.268|0.425/0.286|0.453/0.318|0.479/0.327|0.487/0.331|
>
> > Other3 of Reviewer CJUx: Appendix G could be referenced in the main text.
>
> Thanks for your reminder. We will move **Appendix G** to the main text in our final version and highlight our robustness to moving average window size.
>
> > Other4 of Reviewer CJUx: Include some forecasting examples in the main text.
> + Considering the large space occupation of visualization figures and the page limitation of the main text, we place the forecasting examples in **Appendix N**.
> + We will add a sentence in **Section 4 line 228** to direct readers to **Appendix N** for forecasting examples.
>
> > Q1 of Reviewer CJUx: Is $\lambda$ learned during training, or is it tuned as a hyperparameter?
> + Thanks for highlighting this. As mentioned in **line 198**, $\lambda$ is learned during training.
> + To ensure visibility, we will bold this sentence in our final version.

---

### Official Review · Reviewer_5VM5 · 2025-06-30

**Clarity:** 2
**Significance:** 3
**Originality:** 3
**Rating:** 3
**Confidence:** 4

**Summary:**

The authors propose training the seasonal module and trend module separately, and then fusing them into any time series model. However, the details of the model are not clearly explained, and treating different time series models merely as simple linear combinations is not appropriate.

**Questions:**

see weaknesses

**Ethical Concerns:**

["NO or VERY MINOR ethics concerns only"]

**Final Justification:**

The author’s explanation effectively addresses w1, w3, and w4, and I truly appreciate the clarity in their responses. However, while I acknowledge the author's perspective on w1, I still maintain my own viewpoint regarding it. As for w5, although the model framework diagram is helpful, it could benefit from further optimization as it currently appears to be somewhat simplistic.

**Limitations:**

yes

**Paper Formatting Concerns:**

No formatting concerns

**Quality:**

2

**Strengths And Weaknesses:**

Strengths

The authors  pretrain two copies of it on the decomposed seasonal and trend data, helping them grasp  the decomposition-related knowledge and making them to be the seasonal and trend expert models.  Then we fuse these two expert models back as one single model, merging all of their decomposition related knowledge into the fused model. In this way, author make it to inject the decomposition related knowledge into a single model (i.e., the fused model).

weaknesses:
1. What are the network architectures of Model X and Model Y?
2. Wt and Ws cannot be fused using Equation 10. On one hand, Xs and Xt are derived under a non-stationary mode. On the other hand, for Equation 10, the second equality does not hold—there are no parameters $\lambda$ that can make the two sides equal.
3. In Fusion of complicated models module, the explanation lacks a detailed discussion on the fusion of the two, particularly  for models like patchTST and iTransformer, which do not fall under the category of simple linear models.
4. In the introduction, the authors mention the implementation of inference costs; however, in the experiments, there is no detailed analysis of the inference costs after incorporating DecompNet.
5. The DecompNet section lacks an overall model architecture diagram.
6. Including specific experimental results in the introduction and abstract would further enhance the clarity and impact.

---

> ### Author Rebuttal · Authors · 2025-07-31
>
> Many thanks to Reviewer 5VM5 for the detailed review. However, it seems that some comments in the "Summary" part of the review **stem from a misunderstanding of our work**.
>
> To eliminate potential misunderstanding, we'd like to clarify and explain our work as follows:
> + DecompNet is a model-agnostic enhancement framework and it can be applied to any time series models. When using our DecompNet to enhance a given time series model, it has following steps:
>     + When given a model to be enhanced, we first pretrain two copies of it on the decomposed seasonal and trend data, making these two copies to become the seasonal and trend expert models.
>     + Then we fuse these two expert models (i.e., the seasonal expert model and the trend expert model) back as one single model, merging all of their decomposition-related knowledge into the fused model. In this way, we make it to inject the decomposition-related knowledge into a single model (i.e., the fused model).
>     + And this fused model can be seen as an enhanced version of the original given model.
>
> > Responses to some comments in the "Summary" part
> + Response to comment "The authors propose training the seasonal module and trend module separately, and then fusing them into any time series model":
>     + Our method is not "fuse two separate seasonal module and trend module into any time series model so as to enhance that time series model".
>     + Our method is "when given any time series model to be enhanced, we first pretrain two copies of the given model to be the seasonal and trend expert models. Then we fuse these two expert models as one single model. And this fused model can be seen as an enhanced version of the original given model".
> + Response to comment "the details of the model are not clearly explained":
>     + We'd like to clarify that our proposed DecompNet is not a specific model. It is a model-agnostic enhancement framework.
>     + And the details of DecompNet are clearly introduced in **Section 3** and summarized in **Section 1 line 50-59**.
> + Response to comment "treating different time series models merely as simple linear combinations":
>     + We do not treat different time series models merely as simple linear combinations.
>     + To ensure we address your concern most effectively, could you please specify which sentences in our paper lead to this comment?
>
> > W1: What are the network architectures of Model X and Model Y?
> + We suppose this question may be "What are the network architectures of seasonal model and trend model", because we do not mention "Model X" and "Model Y" in this paper.
> + Seasonal model and trend model are **two copies of the original model to be enhanced**. So they have the same network architecture as the original model.
> + For "what is the network architecture of the original model": Since DecompNet is a model-agnostic enhancement framework, the original model can be any time series models. And specifically in our main experiments, we use PatchTST, iTransformer, RLinear, RMLP and ModernTCN as the original models.
>
> > W2-1: About fusion process: Can $W_{s}$ and $W_{t}$ be fused using Equation 10?
>
> Yes. We'd like to highlight that **$W_{s}$ and $W_{t}$ can be fused**. And **we have verified this through adequate experimental evidence**.
> + The consistent forecasting improvements across a wide range of models and datasets in **Section 4.1** and our ablations in **Section 4.4** can verify the effectiveness of our fusion process.
> + Similar researches on model fusion in other fields of Natural Language Processing (NLP) [1-3] and Computer Vision (CV) [4] can also help to guarantees the feasibility of such fusion process.
>
> [1] Qwen3 Embedding: Advancing Text Embedding and Reranking Through Foundation Models
>
> [2] Language Models are Super Mario: Absorbing Abilities from Homologous Models as a Free Lunch
>
> [3] Model Merging in Pre-training of Large Language Models
>
> [4] Multimodal Pathway: Improve Transformers with Irrelevant Data from Other Modalities
>
> > W2-2: About Equation 10: $X_{s}$ and $X_{t}$ are derived under a non-stationary mode.
>
> As defined and mentioned in **line 121, 157, 181**, $X_{s}$ and $X_{t}$ are two mathematical symbols to denote the decomposed seasonal and trend components. **They are not derived**.
>
> > W2-3: About Equation 10: Can we make Equation 10 hold? Is there a fusion factor $\lambda$ that can make the two sides equal?
>
> Yes. As mentioned in **line 197-198**, we introduce an extra training process to learn the optimal fusion factors $\lambda$ to make Equation 10 hold. And **our experimental results in Section 4.1 and our ablations in Section 4.4, Appendix E.3 can verify that such optimal fusion factors $\lambda$ really exist**.
>
> > W3: Lack of discussion on the fusion of the models like patchTST and iTransformer.
> + Thanks for highlighting this. PatchTST and iTransformer fall under the category of Transformer-based models. And the fusion of Transformer-based models is explicitly addressed in **line 219-222 (Section 3.4 Paragraph "Fusion of complicated models")**, where we can achieve the fusion of two structurally identical Transformer-based models by fusing each of the corresponding linear layers.
>
> > W4: Lack of analysis of the inference costs.
> + Thanks for highlighting this. We have provided a detailed analysis of the inference costs in **line 279-288 (Section 4.2 Paragraph "Efficiency Comparison")** and provided the experimental evidence in **Figure 4 (c)**.
> + The results show that our method has totally the same inference speed and memory usage as the original model, demonstrating that our method can greatly improve the forecasting performance and bring completely no additional inference costs.
>
> > W5: The DecompNet section (Section 3) lacks an overall model architecture diagram.
> + Thanks for highlight this. We have provided the diagram to demonstrate the overall workflow of DecompNet in **Figure 1 (b)**. And we have included the sentence "Figure 1 (b) shows the overall workflow of DecompNet" in **Section 3 line 126** as a reference to this diagram.
> + And we'd like to claim that DecompNet is not a specific model architecture but a model-agnostic enhancement framework.
>
> > W6: Including specific experimental results in the introduction and abstract would further enhance the clarity and impact.
> + We very much appreciate your great suggestions about the paper writing. But considering the the page limitation of the main text, we hope we can maintain to arrange the presentation of experimental results as follows:
>     + In **Abstract (line 9-15) and Introduction (line 59-63, 67-72)**, we **briefly summarize our experimental performance** to give readers a preliminary understanding of our performance superiority.
>     + In **Section 4**, we provide **specific experimental results and detailed performance analysis** to further enhance the clarity and impact.
> ---
> **Due to space limitation, we place some responses to Reviewer CJUx here.**
> > W4 of Reviewer CJUx: Convey the training cost details (Appendix E.1) in the main text.
>
> Thanks for your suggestion. We will move **Appendix E.1** to the main text in our final version and highlight our nice training efficiency.
>
> > W5 of Reviewer CJUx: Include a summary sentence before redirecting readers to the appendix.
> + For the two baselines MICN and Leddam, we will rephrase **line 309-311** as: "We compare our DecompNet with some latest and representative explicit decomposition methods, i.e., MICN  and Leddam. Please refer to **Appendix F.3** for a detailed introduction of their model structures and data processing pipelines".
> + For moving average window size, we will add a new sentence in **line 234**: "We set the moving average window size as 25 for time series decomposition during the decoupled pretraining stage. And we provide the parameter sensitivity study in **Appendix G**".
> + For training details, we will add a new sentence in **line 234**: "Our method is trained with L2 loss, using ADAM optimizer with an initial learning rate in {$10^{-3}, 5\times 10^{-4}, 10^{-4}$}. The default training process is 30 epochs with early stopping. More implementation details are in **Appendix C**".
>
> > Other2 of Reviewer CJUx: Comparison with other prominent decomposition-based models like DLinear, NHiTs and N-BEATS
> + Following your valuable suggestion, we conduct these comparisons. As shown in following table, our method surpasses these prominent decomposition-based models, achieving an average MSE reduction of 11.1% compared to DLinear, 16.7% compared to NHiTs and 18.6% compared to N-BEATS, thus verifying our performance superiority.
> + Please also see **Section 4.3** for our comparison with two more competitive decomposition-based models (i.e., MICN and Leddam).
>
> |MSE/MAE|DecompNet+PatchTST|DecompNet+RLinear|DLinear|NHiTs|N-BEATS|
> |:---:|:---:|:---:|:---:|:---:|:---:|
> |ETTh1|0.403/0.421|0.410/0.422|0.428/0.444|0.486/0.477|0.500/0.486|
> |ETTm1|0.345/0.380|0.359/0.377|0.365/0.385|0.372/0.397|0.385/0.404|
> |Solar|0.199/0.257|0.227/0.274|0.255/0.338|0.265/0.348|0.266/0.345|
> |Traffic|0.387/0.268|0.425/0.286|0.453/0.318|0.479/0.327|0.487/0.331|
>
> > Other3 of Reviewer CJUx: Appendix G could be referenced in the main text.
>
> Thanks for your reminder. We will move **Appendix G** to the main text in our final version and highlight our robustness to moving average window size.
>
> > Other4 of Reviewer CJUx: Include some forecasting examples in the main text.
> + Considering the large space occupation of visualization figures and the page limitation of the main text, we place the forecasting examples in **Appendix N**.
> + We will add a sentence in **Section 4 line 228** to direct readers to **Appendix N** for forecasting examples.
>
> > Q1 of Reviewer CJUx: Is $\lambda$ learned during training, or is it tuned as a hyperparameter?
> + Thanks for highlighting this. As mentioned in **line 198**, $\lambda$ is learned during training.
> + To ensure visibility, we will bold this sentence in our final version.

---

> ### Author Response · Authors · 2025-08-05
>
> We would like to thank Reviewer 5VM5 again for providing the valuable review and insightful suggestions. Your constructive suggestions are very helpful for us to improve the paper into a better shape.
>
> And we would also like to thank you for raising the score and recommending our paper!
>
> If you have any further questions or concerns, please feel free to let us know. We'd be very happy to answer any further questions.

---

### Official Review · Reviewer_f7pT · 2025-07-01

**Clarity:** 3
**Significance:** 4
**Originality:** 3
**Rating:** 5
**Confidence:** 3

**Summary:**

This paper proposes DecompNet, a model-agnostic framework that enhances time series forecasting models via implicit decomposition. During training, two expert models are trained separately on the seasonal and trend components of the input series. These expert models are later fused into a single model using a learnable fusion factor, enabling inference on raw time series without requiring explicit decomposition at test time. Extensive experiments across 8 benchmark datasets and multiple backbone models (e.g., PatchTST, RLinear) show consistent improvements in accuracy. The authors also claim that DecompNet outperforms prior decomposition methods and normalization-based frameworks like RevIN, without increasing inference cost. The main contributions are: (1) introducing implicit decomposition, (2) proposing a decoupled-and-fusion training scheme, and (3) demonstrating consistent gains across model types and datasets

**Questions:**

1.	Are the fusion weights (λ factors) shared across all layers, or learned independently per layer?
Clarifying this would help assess the flexibility and generalization capability of the fusion strategy. A more detailed explanation—ideally supported by a schematic or pseudocode—would improve transparency and reproducibility.
2.	How does DecompNet handle architectural mismatches during fusion?
If the expert models have different depths or contain non-matching layers (e.g., normalization, residual connections), how is the fusion performed? Does the fusion mechanism rely on strict layer-wise alignment, and how are such inconsistencies resolved?
3.	How robust is DecompNet to non-stationary test-time inputs?
Without any mechanism to adjust for distribution shifts at inference time, how does the model perform when the input time series exhibits level shifts, changing trends, or variance fluctuations not seen during training? Is there any empirical evidence comparing its performance with methods like RevIN that explicitly account for such shifts?
4.	How do the authors plan to address the scalability of DecompNet to more than two decomposition components?
The paper acknowledges as a limitation that the current implementation only supports seasonal and trend components and suggests future integration with methods like STL or frequency-domain decomposition. Could the authors elaborate on what specific technical challenges they foresee in extending DecompNet to handle additional components (e.g., residuals or noise)? For example, how might fusion complexity, training stability, or parameter efficiency be affected?

**Ethical Concerns:**

["NO or VERY MINOR ethics concerns only"]

**Final Justification:**

The rebuttal provided clear answers to my concerns regarding the fusion mechanism, including the prerequisite of identical architectures and the independent learning of fusion weights across layers. Additional experiments comparing DecompNet with RevIN on highly non-stationary datasets, as well as the preliminary extension to three-component decomposition using STL, convincingly support the robustness and scalability of the method. While the method primarily builds on existing concepts, the integration of implicit decomposition with structural re-parameterization is practical and novel within the time series forecasting context. Some limitations remain, such as the reliance on identical architectures for fusion, but these are acknowledged by the authors and clearly outlined as directions for future work. In light of these clarifications and additions, I find the contribution solid and impactful, and have accordingly increased my score.

**Limitations:**

Yes, the authors acknowledge a key limitation of their current implementation: it only supports decomposition into two components—seasonal and trend—and does not yet incorporate more advanced methods such as STL or frequency-domain decomposition. This limitation is clearly stated in Appendix L, along with a plan for future work. However, it would strengthen the paper if the authors also discussed the potential technical challenges involved in extending the framework to handle more than two components (e.g., residuals or noise), such as increased fusion complexity, parameter growth, or optimization difficulty. Addressing these considerations would provide a clearer picture of the method’s scalability.

**Paper Formatting Concerns:**

No major formatting issues.

**Quality:**

3

**Strengths And Weaknesses:**

Strengths:
- Practical Innovation: Injects decomposition-related knowledge via a learnable fusion step, eliminating the inference overhead of prior decomposition-based frameworks.
- Model-Agnostic Compatibility: Enhances diverse forecasting architectures (e.g., Transformer, MLP, CNN) without modifying their design.
- Strong Empirical Rigor: Evaluates across 8 datasets and multiple baselines, with ablations and multiple seeds confirming consistency.
- Efficiency + Effectiveness: Achieves notable performance gains without any added inference cost—which is rare among enhancement frameworks. The decomposition is done implicitly during training, so the final model runs as fast and light as the original, making it highly practical for real-world deployment.

Weaknesses:
- Limited Technical Novelty: The framework builds on existing ideas such as structural re-parameterization and training expert models on decomposed components. As a result, it may be seen more as a practical engineering extension rather than a fundamentally novel algorithmic breakthrough. The core idea – train separate models for trend and seasonal components and then merge – is clever but not entirely unprecedented.
- Unclear Fusion Implementation: Although the paper outlines a high-level strategy for fusing expert models by aligning corresponding linear or convolutional layers, it does not clarify how architectural mismatches are handled, such as differences in model depth, or the presence of normalization layers and residual connections. Furthermore, it remains unclear whether the fusion weights (e.g., λ factors) are shared across all layers or learned independently for each layer. These omissions limit clarity and may hinder reproducibility.
- Lack of Robustness to Test-Time Shifts: The method assumes stationary test distributions and currently has no mechanism to adapt to distribution shifts at inference (such as sudden level changes or variance shifts in the input data). Unlike normalization-based methods like RevIN or more adaptive frameworks, DecompNet does not explicitly address these forms of non-stationarity. This could make the approach vulnerable when the training and testing data distributions differ significantly.

---

> ### Author Rebuttal · Authors · 2025-07-31
>
> Many thanks to Reviewer f7pT for the insightful comments.
>
> > W1: Technical Novelty
>
> We'd like to highlight our difference from previous works as follows:
> + Although the idea of training expert models on decomposed components already exists, **we innovatively propose a better training strategy**, which can fully unleash the performance potential of decomposition-based methods.
>     + Previous methods mainly adopt a joint training strategy, which can only provide messily entangled supervision signals and limit the performance of expert models.
>     + In this paper, we propose a novel decoupled training strategy. By decoupling the whole-series prediction task into a seasonal prediction task and a trend prediction task, our strategy can provide more direct and clearer supervision signals for expert models, fully unleashing their performance potential.
> + Although the technique of structural re-parameterization has been utilized in other domain, we **innovatively integrate it with time series decomposition**.
>     + Our Seasonal-Trend Re-parameterization is **a novel variant** of structural re-parameterization **specially designed for implicit decomposition**, helping to achieve **inferece-cost-free** forecasting performance enhancement.
> + With the help of the above two innovations, we **successfully pioneer and verify the idea of implicit decomposition**, which can provide some **novel insights** to time series community:
>     + We are **the first** to propose a decomposition-based enhancement framework that surpasses the well-recognized normalization-based frameworks. And we are **the first** to propose a inference-cost-free enhancement framework. **These unprecedented findings can encourage further exploration on more types and more efficient enhancement frameworks**.
>     + As a **brand new idea**, our implicit decomposition can **bring some fresh perspectives** and provide a better solution to the classic research topic of **time series decomposition**.
> + We hope the above clarification helps to distinguish our method from previous works and clarify our contributions, thus highlighting our novelty.
>
> > W2-1, Q2: How does DecompNet handle architectural mismatches during fusion?
>
> >> Does the fusion mechanism rely on strict layer-wise alignment?
> + Yes, it relies on strict layer-wise alignment. The prerequisite for the fusion process is that **the two expert models must have the same model architecture**.
>
> >> We will not encounter the situation of architectural mismatches in this paper.
> + In this paper, **the two expert models in our fusion process definitely have the same architecture**, since they are two copies of the same original model.
> + As a result, **we will not encounter the situation of architectural mismatches during fusion** and we do not need to handle this situation specifically.
>
> >> Future work
> + We fully acknowledge the reviewer's interest in handling the architectural mismatches during model fusion. However, such topic remains an open challenge beyond current methodological capabilities—even leading works in this nascent field (e.g., [1-4]) mainly focus on the fusion of models with same architecture.
> + Aligning with state-of-the-art practice, our method mainly focuses on the fusion of models with same architecture. And our method can avoid the situation of architectural mismatches during fusion.
> + It will be our future work to further study on how to handle the architectural mismatches during fusion.
>
> [1] Qwen3 Embedding: Advancing Text Embedding and Reranking Through Foundation Models
>
> [2] Language Models are Super Mario: Absorbing Abilities from Homologous Models as a Free Lunch
>
> [3] Model Merging in Pre-training of Large Language Models
>
> [4] Multimodal Pathway: Improve Transformers with Irrelevant Data from Other Modalities
>
> > W2-2, Q1-1: Whether λ factors are shared across all layers or learned independently for each layer.
> + λ factors are learned independently for each layer.
>
> > W2-3, Q1-2: About clarity and reproducibility
> + Thanks for the suggestion.
> + To enhance clarity, we will emphasize the same model architecture as a prerequisite for model fusion in **Section 3.4**. And we will also clarify that λ factors are learned independently for each layer.
> + To ensure reproducibility, we guarantee to make the code public upon paper acceptance. And we will add a pseudocode related to model fusion process in the appendix for the final version.
>
> > W3, Q3: How robust is DecompNet to non-stationary test-time inputs? About mechanism guarantees. Empirical evidence comparing DecompNet with RevIN in non-stationary issues.
> + DecompNet can robustly handle non-stationary test-time inputs. And based on our experimental results, it performs better than RevIN.
> + Mechanism Guarantees: DecompNet **has mechanism** to handle non-stationary test-time inputs, but the mechanism is different from RevIN's:
>     + RevIN is a normalization-based method. It alleviates the non-stationary issue **from the perspective of statistical measures**.
>     + DecompNet is a decomposition-based method. It handles the non-stationary issue **based on time series decomposition**, since decomposition can unravel the entangled temporal patterns in raw time series and and **extract the stationary components** [5-7].
> + Empirical Evidence: To prove our superiority in handling non-stationary issues, we compare our performance with RevIN in the top-3 highly non-stationary datasets, which are ETTh2, Exchange and ILI[8].
>     + For ILI, the input length is 96 and the results are averaged from 4 prediction lengths {24,36,48,60}. For other datasets, the input length is 384 and the results are averaged from 4 prediction lengths {96,192,336,720}.
>     + In following results, the *Original* model equips with RevIN by default. And *Ours* replaces the RevIN with our DecompNet framework. **So these results can provide a fair comparison between our DecompNet and RevIN in handling non-stationary issues**.
>     + It is shown that DecompNet outperforms RevIN in all these highly non-stationary datasets, indicating its better ability in handling non-stationary issues.
> + Conclusion:
>     + As verified on the currently available and publicly accessible datasets, DecompNet performs the best in handling non-stationary issues and is even better than RevIN.
>     + We are well aware that there are non-stationary issues that cannot be represented by the current datasets. In future work, we will keep monitoring the development in time series community and promptly validate the effectiveness of our methods on the latest and more challenging non-stationary datasets.
>
> |ETTh2(MSE/MAE)|Original|Ours|
> |:---:|:---:|:---:|
> |PatchTST|0.349/0.391|0.346/0.387|
> |iTransformer|0.377/0.412|0.344/0.389|
> |RLinear|0.354/0.397|0.338/0.383|
> |RMLP|0.364/0.401|0.347/0.390|
> |ModernTCN|0.328/0.382|0.312/0.373|
>
> |Exchange(MSE/MAE)|Original|Ours|
> |:---:|:---:|:---:|
> |PatchTST|0.387/0.419|0.330/0.404|
> |iTransformer|0.378/0.418|0.317/0.391|
> |RLinear|0.378/0.411|0.314/0.386|
> |RMLP|0.385/0.414|0.320/0.385|
> |ModernTCN|0.376/0.407|0.319/0.376|
>
> |ILI(MSE/MAE)|Original|Ours|
> |:---:|:---:|:---:|
> |PatchTST|2.096/0.969|1.904/0.886|
> |iTransformer|2.231/1.020|1.959/0.934|
> |RLinear|3.686/1.366|2.305/1.042|
> |RMLP|2.796/1.172|2.056/0.972|
> |ModernTCN|1.944/1.198|1.738/0.874|
>
> [5] Forecasting: principles and practice
>
> [6] Forecasting: theory and practice.
>
> [7] Autoformer: Decomposition Transformers with Auto-Correlation for Long-Term Series Forecasting
>
> [8] Non-stationary Transformers: Exploring the Stationarity in Time Series Forecasting
>
> > Q4, Limitation: The scalability of DecompNet to more than two decomposition components. Integration with STL.
>
> >> Experimental results from our preliminary exploration
> + Following your valuable suggestion, we conduct a preliminary exploration on the scalability of DecompNet.
> + The experiment settings are as follows.
>     + We integrate our DecompNet with STL and decompose the time series into three components (i.e., seasonal, trend and residual components).
>     + We introduce more fusion factors to fuse three expert models, letting $W = W_{s} + \lambda_{1} W_{t} + \lambda_{2} W_{r}$, where $W_{s}, W_{t}, W_{r}$ are the weight matrices from seasonal, trend and residual expert models respectively.
>     + We use RLinear and PatchTST as the backbone models.
> + As shown in the following table, we observe continuous performance improvement after scaling into more and finer components, which can prove the scalability of DecompNet.
>
> |MSE/MAE|PatchTST + 2 Components|PatchTST + 3 Components|RLinear + 2 Components|RLinear + 3 Components|
> |:---:|:---:|:---:|:---:|:---:|
> |ETTh1|0.403/0.421|0.394/0.407|0.410/0.422|0.399/0.410|
> |ETTm1|0.345/0.380|0.338/0.370|0.359/0.377|0.349/0.367|
> |Solar|0.199/0.257|0.193/0.247|0.227/0.274|0.213/0.265|
> |Traffic|0.387/0.268|0.380/0.261|0.425/0.286|0.412/0.277|
>
> >> About potential technical challenges like increased fusion complexity, parameter growth, or optimization difficulty.
>
> Based on our preliminary exploration, we find that:
> + Fusion complexity and optimization difficulty are not increased. Through directly generalizing the equation from $W = W_{s} + \lambda W_{t}$ to $W = W_{s} + \lambda_{1} W_{t} + \lambda_{2} W_{r}$, we can achieve the fusion of three expert models. And the training process is still stable and can converge quickly.
> + Parameter growth:
>     + The number of parameters is only increased during training. When scaling to more decomposition components, we need to train more expert models and introduces more fusion factors. This will introduce more parameters, but will also bring better performance.
>     + During inference, all expert models are fused as one single model, so there is no additional inference costs.
>
> >> Future work
>
> Above preliminary experiments demonstrate the scalability of DecompNet. In future work, we will conduct more in-depth researches to further unleash its scaling potential and report more insights on optimization.

---

> > ### Comment · Reviewer_f7pT · 2025-08-07
> >
> > Thank you for your thorough and well-organized rebuttal.
> > Your clarifications regarding the fusion mechanism—particularly the assumption of strict architectural alignment and the independent learning of layer-wise fusion weights—were helpful in addressing my concerns about clarity and reproducibility.
> > I also appreciate the additional experiments comparing DecompNet with RevIN on highly non-stationary datasets, as well as your preliminary exploration of multi-component decomposition using STL. These results reinforce the robustness and scalability of your proposed framework.
> > While the core components are grounded in known techniques, your integration of implicit decomposition with a re-parameterization-based enhancement strategy presents a novel and practical perspective for time series forecasting.
> > In light of the new evidence and clarifications, I have raised my score and look forward to the potential impact of your work on future developments in decomposition-based forecasting.

---

> > > ### Author Response · Authors · 2025-08-07
> > >
> > > We would like to thank Reviewer f7pT again for providing the valuable review and insightful suggestions. Your constructive suggestions are very helpful for us to improve the paper into a better shape.
> > >
> > > And we would also like to thank you for raising the score and recommending our paper!

---

### Official Review · Reviewer_USxs · 2025-07-02

**Clarity:** 3
**Significance:** 3
**Originality:** 3
**Rating:** 4
**Confidence:** 3

**Summary:**

This paper introduces a framework called DecompNet that implements implicit decomposition to improve time-series forecasting performance without added inference cost. The key idea is to train two copies of a base forecasting model on the decomposed seasonal and trend components of the data (so each becomes an expert in one component) and then fuse them into a single model via a novel Seasonal-Trend re-parameterization technique. During this fusion, the two expert networks’ parameters are merged (with a learned scaling factor) into one set of weights, yielding an enhanced model that can directly ingest raw time series and still leverage decomposition-related knowledge.

**Questions:**

I would consider increasing my score if most of the following points are clearly addressed. Please also refer to the weaknesses section.

- How sensitive is DecompNet’s performance to the choice of decomposition hyperparameters? For instance, the moving average window size (25) was fixed. Did you try other window lengths or methods like STL, and if so, how did they affect results?
- In the fusion stage with the learnable factor $\lambda$, do you allow the weights $W_s$ and $W_t$ of the expert models to update as well, or is only $\lambda$ learned? How stable is this training process? Does $\lambda$ reliably converge to a meaningful value (e.g. close to 1 or something data-dependent), and have you observed any instances where the fused model underperformed the explicit two-model ensemble?
- Implicit decomposition is demonstrated with two components (trend and seasonal). How would the framework extend to decomposing into more than two components (say seasonal/trend/residual)? Would you envision introducing multiple fusion factors (e.g. $\lambda_1, \lambda_2$) to fuse three expert models, or some iterative fusion process?
- In existing explicit decomposition approaches, researchers sometimes choose heterogeneous model types for seasonal vs trend components (anticipating different dynamics). Your approach instead uses the same base architecture for both, which has the huge benefit of enabling weight fusion. Did you experiment with using different architectures for the two expert models without fusion (just to see performance)? If not fused, would, say, a CNN for seasonality and an RNN for trend perform better, or did you find that using the same architecture with decoupled training already captures both aspects well?
- Could you provide more details on the training overhead introduced by DecompNet? For example, how much longer (in terms of epochs or wall-clock time) does it take to train a model with DecompNet compared to training the base model normally?

**Ethical Concerns:**

["NO or VERY MINOR ethics concerns only"]

**Final Justification:**

After reviewing the rebuttal and discussion, my overall evaluation (4) remains unchanged. The authors provided useful clarifications on training overhead, robustness to decomposition parameters, and generalization to alternative decomposition methods. However, several of my key questions, particularly those concerning the principled justification of design choices and a deeper theoretical understanding of the fusion mechanism, were not fully addressed. While the approach is technically solid and has practical relevance, the limited resolution of these points prevents me from raising my score. However, I appreciate the overall novelty and scope of the work and the effort that the authors put into the rebuttal. I therefore maintain my positive rating.

**Limitations:**

The authors do address some limitations of their work, albeit briefly in the appendix. They acknowledge that their implementation of implicit decomposition is currently tied to the classic moving-average decomposition with a fixed seasonal/trend split. They note that extending the framework to use more advanced decomposition techniques (such as STL or frequency-domain methods ) and to decompose time series into finer components (e.g., adding a residual/noise component beyond just trend and seasonal) could further improve performance.

**Paper Formatting Concerns:**

Figures are too small to read easily.

**Quality:**

3

**Strengths And Weaknesses:**

Strengths:

- The paper introduces a new approach to time-series forecasting by injecting decomposition knowledge via training, rather than performing decomposition at inference. This idea of implicit decomposition is highly novel and addresses a gap, as it transforms decomposition into a model-agnostic enhancement that incurs no runtime cost.
- DecompNet delivers consistent t forecasting improvements across a wide range of models and datasets.
- The paper is well-written and logically organized.

Weaknesses:

- DecompNet’s benefits come at the cost of a more complex training pipeline. The method requires training two separate model instances on decomposed data and then performing an extra fusion fine-tuning stage. This means the total training time and computational resources are higher than for training a single model once. The authors do argue that the extra stages converge quickly, but the paper does not quantify the overhead.
- The framework relies on a specific decomposition technique (moving average with fixed window, yielding one seasonal and one trend series). This choice might not be optimal for all data – for example, a window of 25 might not align with the true seasonality in certain datasets, or some time series might have multiple seasonalities or a large residual component. The current method does not adapt the decomposition to data characteristics (e.g., no automatic period detection).
-  While the paper’s approach is empirically validated, it lacks a deeper theoretical explanation for why the fused model can perform as well as two separate models, especially given the non-linear nature of deep networks. The derivation in Equation (10) is a helpful intuition for linear layers, but it doesn’t formally guarantee performance preservation for complex networks – instead the authors rely on a training-based solution. The success of the fusion is demonstrated experimentally, but the paper does not analyze scenarios where it might fail or whether any bounds/guarantees exist.
- The study focuses on one particular way of decomposing (seasonal/trend via moving average). It does not explore whether the implicit decomposition idea generalizes to other forms of decomposition or other non-stationarity issues (e.g., piecewise trends, irregular seasonal patterns).
-

---

> ### Author Rebuttal · Authors · 2025-07-31
>
> Many thanks to Reviewer USxs for the insightful review.
>
> > W1, Q5: Quantify the training costs (in terms of epochs).
> + Thanks for highlighting this. We have reported the quantitative comparison results in **Table 3 in Appendix E.1**. The results show that **the total training epochs consumed by our method are comparable or sometimes less than the original model**.
> + To ensure visibility and highlight our nice training efficiency, we will move **Appendix E.1** to the main text in our final version.
>
> > W2-1, Q1-1: Parameter sensitivity to the choice of moving average window size.
> + As shown in our parameter sensitivity study in **Figure 7 in Appendix G**, our method is very robust to the choice of moving average window size:
>     + Our method maintains consistent performance across window sizes spanning an order of magnitude (7 to 169).
>     + Using the default value of 25 can provide consistently strong performance across various datasets.
> + To ensure visibility and highlight our robustness to moving average window size, we will move **Appendix G** to the main text in our final version.
>
> > W2-2: Why don't we adapt the decomposition to data characteristics?
> + As mentioned above, our method is very robust to window size and can achieve consistently strong performance across various datasets just using the default value of 25. This property **eliminates the need to spend extra effort analyzing the dataset in advance or adjusting hyperparameters based on the characteristics of the dataset**.
> + Since we aim to propose a **prior-free** enhancement framework (i.e., a method that does not require any prior analysis of a dataset), we do not analyze the characteristics of the dataset in advance to determine the hyperparameters.
>
> > W3-1: Analysis on model fusion
>
> >> Theoretical explanation:
> + We sincerely thank the reviewer for highlighting the importance of theoretical explanation. However, rigorous theoretical explanation for model fusion remains an open challenge beyond current methodological capabilities—even leading works in this nascent field (e.g., [1-4]) rely primarily on empirical validation.
> + In this paper, we mainly verify the effectiveness of our method through **extensive empirical results and comprehensive experimental evidence**, which is consistent with state-of-the-art practice.
> + It will be our future work to conduct deeper analysis on model fusion and provide better theoretical explanation.
>
> >> Guarantees.
> + The extensive experimental results in our paper can firmly verify the effectiveness of our fusion process.
> + Similar researches on model fusion in the fields of Natural Language Processing (NLP) [1-3] and Computer Vision (CV) [4] can also help to guarantees the feasibility of such fusion process.
>
> >> Bounds:
> + Since deriving rigorous theoretical bounds remains challenging, we expand our experimental analysis to address this concern empirically.
> + We consider the following boundary cases to test the performance bounds of our method:
>     + Enhancing the performance of state-of-the-art models. The results are provided in **Table 1 in Section 4.1**.
>     + Enhancing the performance of classic models (e.g., standard RNN). The results are provided in **following table**.
>     + Handling highly non-stationary data. Please refer to the **Response to W3, Q3 of Reviewer f7pT** for detailed results.
> + Above boundary cases cover the extreme scenarios in time series domain. And the results show that our method can work well for each case, which can ensure the reliability of our method in most application scenarios.
>
> |MSE/MAE|RNN|+Ours|
> |:---:|:---:|:---:|
> |ETTh1|1.114/0.794|0.448/0.447|
> |ETTm1|1.130/0.779|0.372/0.404|
> |Solar|0.913/0.498|0.246/0.288|
> |Traffic|1.043/0.486|0.474/0.309|
>
> > W4-1, Q1-2, Q3, Limitation: Can our method generalize to other decomposition method like STL? Can our method extent to more and finer decomposition components?
> + Following your valuable suggestion, we integrate our DecompNet with STL and decompose the time series into three finer components (i.e., seasonal, trend and residual components). The experiment settings are as follows.
>     + We introduce more fusion factors to fuse three expert models (i.e., $W = W_{s}+\lambda_{1} W_{t}+\lambda_{2} W_{r}$).
>     + We use RLinear and PatchTST as the backbone models.
> + As shown in following table, after integrating DecompNet with more advanced decomposition methods and decomposing the time series into more components, we observe continuous performance improvement, which can prove our ability to further generalize and extend.
> + It will be our future work to further explore the potential of DecompNet in generalization and extension.
>
> |MSE/MAE|PatchTST + 2 Components|PatchTST + 3 Components|RLinear + 2 Components|RLinear + 3 Components|
> |:---:|:---:|:---:|:---:|:---:|
> |ETTh1|0.403/0.421|0.394/0.407|0.410/0.422|0.399/0.410|
> |ETTm1|0.345/0.380|0.338/0.370|0.359/0.377|0.349/0.367|
> |Solar|0.199/0.257|0.193/0.247|0.227/0.274|0.213/0.265|
> |Traffic|0.387/0.268|0.380/0.261|0.425/0.286|0.412/0.277|
>
> > W4-2: Can our method generalize to other non-stationary issues?
> + In this paper, we compare our DecompNet with other enhancement frameworks on many public non-stationary datasets. These results verify that DecompNet performs the best in handling non-stationary issues and outperforms all other enhancement frameworks, which demonstrates the strong capabilities of our method in handling non-stationary issues.
> + We are well aware that there are non-stationary issues that cannot be represented by the current datasets. In future work, we will keep monitoring the development in time series community and promptly validate the effectiveness of our method on the latest and more challenging non-stationary datasets.
>
> > Q2-1: Do you allow the weights $W_s$ and $W_t$ to update as well?
> + Yes. As stated in **line 202-203**, $W_{s}$ and $W_{t}$ are also updated with $\lambda$.
> + Thanks for highlighting this. We will bold this sentence in our final version to ensure visibility.
>
> > Q2-2: How stable is the training process for $\lambda$?
> + As shown in **Appendix E.1**, the training process for $\lambda$ is very stable and it can converge quickly with in 3 epochs in most cases.
>
> > Q2-3: What value do $\lambda$ converge to?
> + $\lambda$ can reliably converge to a meaningful value. This value is data-dependent and this value is usually not equal to 1.
> + As shown in **Appendix E.3**, the fusion of $W_{s}+\lambda W_{t}$ consistently performs better than $W_{s}+W_{t}$, which indicates that the final learned $\lambda$ is usually not equal to 1. Instead, it is learned to be an optimal value data-dependently.
>
> > Q2-4, W3-2: Scenarios where the fusion process might fail. Can our fusion process guarantee the performance preservation for complex networks? Have you observed any instances where the fused model underperformed the explicit two-model ensemble?
> + As shown in our ablation study in **Section 4.4 and Appendix H**, the fused model outperforms the explicit two-model ensemble in most cases.
> + The only exception case is in ETTh1, where the MSE sightly increase after fusion. When using PatchTST as the backbone, the MSE increases by 0.001 (0.402-->0.403) (**Figure 8 (a) in Appendix H**). And when using RLinear as the backbone, the MSE increases by 0.003 (0.407-->0.410) (**Figure 9 (a) in Appendix H**). In this exceptional case, although the fused model exhibits inferior performance, **the performance difference is very small**.
> + These results empirically validate that our fusion process can guarantee performance preservation for both complex networks (e.g., PathcTST) and simple networks (e.g., RLinear).
>
> > Q4: "using different architectures for two expert models" or "using the same architecture for two expert models"
>
> >> Did you experiment with using different architectures for the two expert models without fusion (just to see performance after decoupled training)?
> + Yes. In **Section 4.3**, we conduct experiments with two popular heterogeneous models, i.e., MICN and Leddam. MICN uses linear layers for trend part and uses CNN for seasonal part. Leddam uses linear layers for trend part and uses attention for seasonal part.
> + Our decoupled training strategy can also enhance the performance of heterogeneous models. Compared with previous joint training strategy, our strategy reduces the average MSE of MICN from 0.375 to 0.320, and reduces the average MSE of Leddam from 0.315 to 0.295 (**Figure 5 (b) in Section 4.3**).
>
> >> Did you find that using the same architecture with decoupled training already captures both aspects well?
> + Yes. After using our decoupled training strategy, we find that adopting the same architecture for two expert models can achieve the same performance as heterogeneous models (**Figure 5 (b) in Section 4.3**).
> + We attribute this finding to our decoupled training strategy, since it can better train the two expert models and fully realize the strong representation capabilities within neural networks, making them able to capture the meaningful seasonal and trend representations even with same model structure (**line 337-339 in Section 4.3**).
>
> >> Experimental evidence for Q4. This table is a copy of the result from **Figure 5 (b) in Section 4.3**
>
> |Average MSE in eight datasets|heterogeneous models with joint training|heterogeneous models with our decoupled training|same model structure with our decoupled training|
> |:---:|:---:|:---:|:---:|
> |MICN|0.375|0.320|0.320|
> |Leddam|0.315|0.295|0.295|
>
> > Concern about figure size
>
> Thanks for your reminder. We will increase the size of key figures to ensure they can be read easily.
>
> > Reference
>
> [1] Qwen3 Embedding: Advancing Text Embedding and Reranking Through Foundation Models
>
> [2] Language Models are Super Mario: Absorbing Abilities from Homologous Models as a Free Lunch
>
> [3] Model Merging in Pre-training of Large Language Models
>
> [4] Multimodal Pathway: Improve Transformers with Irrelevant Data from Other Modalities

---

> ### Author Response · Authors · 2025-08-01
> **A copy of the results for "W3-1: Analysis on model fusion—bounds" and more discussion about this concern**
>
> Dear Reviewer USxs,
>
> Due to the space limitation, in our original responses to **W3-1: Analysis on model fusion—bounds**, we refer some results from the **Response to W3, Q3 of Reviewer f7pT**.
>
> To facilitate your review of our responses, **we provide a copy of these results here**. And we further provide more discussion about this concern. We hope it can better help to address your concern.
>
> > W3-1: Analysis on model fusion
>
> >> Bounds:
> + Since deriving rigorous theoretical bounds remains challenging, we expand our experimental analysis to address this concern empirically.
> + We consider the following boundary cases to test the performance bounds of our method:
>     + Boundary case 1: Enhancing the performance of state-of-the-art models. The results are as follows, which verify that our method can enhance the performance of a wide range of state-of-the-art models. Even though these models have already achieved the previous state-of-the-art performance by themselves, our method can still further improve their performance.
>
>     |MSE/MAE|PatchTST|PatchTST+Ours|iTtransformer|iTtransformer+Ours|RLinear|RLinear+Ours|RMLP|RMLP+Ours|ModernTCN|ModernTCN+Ours|
>     |:---:|:---:|:---:|:---:|:---:|:---:|:---:|:---:|:---:|:---:|:---:|
>     |ETTh1|0.418/0.433|0.403/0.421|0.461/0.464|0.417/0.435|0.415/0.428|0.410/0.422|0.482/0.469|0.423/0.430|0.403/0.419|0.387/0.411|
>     |ETTm1|0.355/0.385|0.345/0.380|0.372/0.399|0.358/0.385|0.364/0.381|0.359/0.377|0.371/0.394|0.355/0.383|0.365/0.386|0.354/0.379|
>     |Solar|0.218/0.317|0.199/0.257|0.212/0.272|0.204/0.259|0.256/0.323|0.227/0.274|0.231/0.294|0.214/0.249|0.218/0.306|0.200/0.258|
>     |Traffic|0.410/0.288|0.387/0.268|0.428/0.320|0.395/0.280|0.453/0.314|0.425/0.286|0.436/0.308|0.413/0.285|0.417/0.293|0.394/0.275|
>
>     + Boundary case 2: Enhancing the performance of classic models (e.g., standard RNN). The results are as follows, which verify that our method can also work on the classic model and greatly improve its performance, making it catch up with the state-of-the-art level in time series forecasting.
>
>     |MSE/MAE|RNN|RNN+Ours|
>     |:---:|:---:|:---:|
>     |ETTh1|1.114/0.794|0.448/0.447|
>     |ETTm1|1.130/0.779|0.372/0.404|
>     |Solar|0.913/0.498|0.246/0.288|
>     |Traffic|1.043/0.486|0.474/0.309|
>
>     + Boundary case 3: Handling highly non-stationary data. We conduct experiments on the top-3 highly non-stationary datasets, which are ETTh2, Exchange and ILI. The following results show that our method brings consistent performance improvement across these highly non-stationary datasets, which demonstrate the strong capabilities of our method in handling non-stationary issues.
>
>     |ILI(MSE/MAE)|Original|+Ours|
>     |:---:|:---:|:---:|
>     |PatchTST|2.096/0.969|1.904/0.886|
>     |iTransformer|2.231/1.020|1.959/0.934|
>     |RLinear|3.686/1.366|2.305/1.042|
>     |RMLP|2.796/1.172|2.056/0.972|
>     |ModernTCN|1.944/1.198|1.738/0.874|
>
>     |Exchange(MSE/MAE)|Original|+Ours|
>     |:---:|:---:|:---:|
>     |PatchTST|0.387/0.419|0.330/0.404|
>     |iTransformer|0.378/0.418|0.317/0.391|
>     |RLinear|0.378/0.411|0.314/0.386|
>     |RMLP|0.385/0.414|0.320/0.385|
>     |ModernTCN|0.376/0.407|0.319/0.376|
>
>     ETTh2(MSE/MAE)|Original|+Ours|
>     |:---:|:---:|:---:|
>     |PatchTST|0.349/0.391|0.346/0.387|
>     |iTransformer|0.377/0.412|0.344/0.389|
>     |RLinear|0.354/0.397|0.338/0.383|
>     |RMLP|0.364/0.401|0.347/0.390|
>     |ModernTCN|0.328/0.382|0.312/0.373|
>
> + Above boundary cases cover the extreme scenarios in time series domain. And the results show that our method can work well for each case, which can ensure the reliability of our method in most application scenarios.

---

> > ### Comment · Reviewer_USxs · 2025-08-03
> >
> > Thank you to the reviewers for the time and effort they dedicated to addressing my concerns.
> >
> > I still find the choice of a fixed context window size of 25 unclear. While I understand that multiple values were tested during development, it's not evident why 25 is the appropriate choice across all datasets or forecasting tasks. I would have appreciated a more principled justification or an analysis of sensitivity to this hyperparameter.
> >
> > After reviewing the author response and considering the perspectives of the other reviewers, I have decided to maintain my original score.

---

> > > ### Author Response · Authors · 2025-08-04
> > >
> > > We would like to thank Reviewer USxs again for providing the valuable review and insightful suggestions. Your constructive suggestions are very helpful for us to improve the paper into a better shape.
> > >
> > > And we would also like to thank you for recognizing and recommending our paper!
> > >
> > > > About further comments on moving average window size:
> > >
> > > Thanks a lot for your valuable further comments. We would like to provide more elaborations as follows. We hope it can better help to address your concerns.
> > >
> > > >> Why do we use a fixed moving average window size of 25?
> > >
> > > + We'd like to explain the reasons for using a fixed moving average window size of 25 as follows:
> > >     + **Experimental Fairness**: The primary motivation for adopting a fixed window size of 25 is to **ensure strict compliance with mainstream protocols**. Starting from the famous decomposition-based methods like Autoformer and DLinear, **it becomes a widely adopted protocol** to use a fixed moving average window size of 25 for the decomposition implementations. Consequently, our main experiments **adhere to this convention** to ensure a fair experimental setting.
> > >     + **Simplicity**: In our humble opinion, **minimizing the numbers of tunable hyperparameter** is critical for real-world applicability, as it reduces the implementation complexity and enhances the adaptability across various real-world scenarios. Since the window size of 25 can **already provide sufficient performance** while **aligning with mainstream conventions**, we **prioritize simplicity and directly use the fixed moving average window size of 25** without further tuning this hyperparameter.
> > >
> > > >> Why 25 is the appropriate choice across all datasets or forecasting tasks?
> > > + Thanks for providing valuable suggestions and highlighting this point.
> > >     + In this paper, we mainly focus on **the experimental scenarios**. To ensure **simplicity and experimental fairness**, we advocate using a fixed moving average window size of 25, strictly following the convention in the mainstream protocol.
> > >         + Therefore, the value of 25 is an appropriate choice in **experimental scenarios**.
> > >     + We fully agree with the reviewer's suggestion to "adapt the decomposition to data characteristics", which is of great significance when applying our method to specific **real-world application scenarios**.
> > >         + For **real-world application scenarios**, although the window size of 25 can already provide sufficient performance, **there may be more appropriate window sizes that can provide even better performance**. And the reviewer's suggestion of "adapting the decomposition-related hyperparameter according to the data characteristics" can help us to find these more appropriate window sizes.
> > > + To better clarify how to choose the appropriate moving average window size, we will include following sentences in our final version:
> > >     + For **the setting in experimental scenarios**: We advocate using a fixed moving average window size of 25 for the consideration of **simplicity and experimental fairness**.
> > >     + For users seeking to maximize performance **in their own real-world application scenarios**: Although the window size of 25 can already provide sufficient performance, **we are well aware that other window sizes may bring even better performance**. To fully push the performance limit in specific real-world applications, we will encourage the users to explore other moving average window sizes. And the powerful technique recommended by the reviewer like automatic period detection can serve as an effective guidance for their further parameter adjustments.
> > >
> > > Sincerely thanks again for your timely help and kind dedication.

---

### Decision · Program_Chairs · 2025-09-17

**Decision:**

Accept (poster)

**Comment:**

This paper introduces DecompNet, which performs implicit time series decomposition to enhance time-series forecasting without additional inference costs. In the training phase, two expert models are trained on the seasonal and trend components of the input time series, respectively. The two experts are fused into a single model through seasonal-trend re-parameterization with a learnable scaling factor.

Initially, this paper received mixed recommendations. Major concerns raised by reviewers include the additional costs for training two experts, lack of evaluation on the inference costs, the dependency on a specific decomposition technique, lack of discussion or theoretical explanation on the success of the fusion and the implicit decomposition during inference, the generalization to other forms of decomposition and non-stationarity, the validation of some technical choices, limited technical novelty, unclear fusion implementation, lack of quantitative evidence regarding some claims, lack of standard deviations or statistical significance tests, lack of comparisons with other decomposition-based models, and presentation issues.

The rebuttal addressed most of these concerns, but several limitations regarding the principled justification of design choices, a deeper theoretical understanding of the fusion mechanism, the reliance on identical architectures for fusion, the lack of statistical tests, and the lack of metrics such as FLOPs and wall-clock time on computational cost remain. The reviewers have provided very detailed feedback. The authors are encouraged to follow their comments closely and include all additional experiments in the camera-ready version. Especially, the authors should (1) include error bars, (2) not use the term “significant” if there is no statistical test conducted, as suggested by the reviewer CJUx.